# Strictly local one-dimensional topological quantum error correction with symmetry-constrained cellular automata

**Nicolai Lang⋆ and Hans Peter Büchler**

Institute for Theoretical Physics III and Center for Integrated Quantum Science and Technology, University of Stuttgart, 70550 Stuttgart, Germany

⋆ nicolai@itp3.uni-stuttgart.de

## Abstract

Active quantum error correction on topological codes is one of the most promising routes to long-term qubit storage. In view of future applications, the scalability of the used decoding algorithms in physical implementations is crucial. In this work, we focus on the one-dimensional Majorana chain and construct a strictly local decoder based on a self-dual cellular automaton. We study numerically and analytically its performance and exploit these results to contrive a scalable decoder with exponentially growing decoherence times in the presence of noise. Our results pave the way for scalable and modular designs of actively corrected one-dimensional topological quantum memories.



# 1 Introduction

Storing quantum information in a noisy, classical environment is essential for scalable quantum computation and communication [1]. Kick-started by Shor's 9-qubit code [2], quantum error correction comes to the rescue: Logical qubits are stored in virtual subsystems [3] that decouple from typical environmental perturbations and allow for error detection and correction [4,5]. Quantum error correction codes come in two flavors: The conventional ones (e.g., Shor's code) have no physical interpretation and are treated as abstract entities, isolated from the underlying computational architecture (much like *classical* error correction codes). *Topological* quantum codes, in contrast, are tied to the real world in that they are realized as ground state manifolds of local Hamiltonians and thereby inherit the geometry of their environment. Familiar examples are the Majorana chain (a *p*-wave superconductor) in one [6,7] and the toric code in two spatial dimensions [8,9], both of which have seen experimental progress in the last years, see e.g. [10–13] and references therein. In this manuscript, we are interested in such topological codes in one dimension and present a method to stabilize them using only strictly local resources.

    Topological codes allow, in principle, for two modes of operation: Taking their realization as ground states seriously entails the intriguing concept of *self-correction* where errors appear as excitations that are energetically suppressed by the parent Hamiltonian [14–16]. In contrast, *active* error correction adopts the algorithmic scheme of conventional codes, i.e., an external decoder is fed with measured syndromes and computes compatible corrections. The fragility of low-dimensional topological order to thermal excitations [17–19], and the so-far unsettled quest for realizable self-correcting codes [20], makes *active* error correction on topological codes one of the most promising routes to long-term qubit coherence [21–23]. As realizable quantum architectures loom on the horizon [24], convenient abstractions face the intricacies of reality: Can active error correction be implemented efficiently? How can it be scaled up when it is cast into hardware? Since space and time constraints can rule out implementations of otherwise promising algorithms, it is a crucial question whether and how topological quantum codes can be stabilized by manifestly local decoders. For the toric code, this has been tackled with a completely local but hierarchical decoder in [25] (inspired

by [26]), with translationally invariant cellular automata [27,28], with a modular setup of simple units connected by noisy links in [29], and with optimized versions of minimum-weight perfect matching [30–33]. Prolonging the lifetime of certain stabilizer codes by local unitary operations (instead of full-fledged error correction) may be a viable alternative [34]. However, rigorous results on the performance of decoders with strict space and time constraints are scarce.

In this work, we focus on the simplest case of a one-dimensional topological quantum code, defined by the ground state space of the Majorana chain [6], and remodel a known (classical) cellular automaton [35,36] to contrive a convenient, strictly local quantum decoder. We prove that both the probability for successful decoding and the time required to do so scales favorably with the chain length, surpassing conventional *global* decoding schemes. For realistic error rates, this allows for the stabilization of logical qubits in the presence of continuous (uncorrelated) noise using shallow, translationally invariant circuits with local wiring only. This paves the way for scalable and modular on-chip realizations of actively corrected topological quantum memories based on one-dimensional *p*-wave superconductors. In the following, we provide a detailed outline of the methods and approaches used to derive these results:

In Subsec. 2.1 we start with a description of the quantum code defined by the degenerate ground state space of the Majorana chain, where dephasing is topologically suppressed and depolarizing errors are forbidden by fermionic parity superselection (which can be violated in real setups due to quasiparticle poisoning [37,38]). This paradigmatic model exemplifies topological quantum error correction and relates to the familiar toric code via Jordan-Wigner transformation in the degenerate case of a $L \times 1$ square lattice with open boundaries. The syndromes of the Majorana chain quantum code (MCQC) are fermionic quasiparticles flanking strings of parity-preserving errors. Maximum-likelihood decoding therefore requires pairing quasiparticles with minimum-length error strings; this scheme is known as *minimum-weight perfect matching* (MWPM) [39] for the toric code and reduces in one dimension to simple *majority voting*, the decoding scheme used for classical repetition codes. In Subsec. 2.2, we review the known result that applying majority voting at a fixed rate to the MCQC leads to an exponentially growing lifetime of the encoded logical qubit with the chain length $L$. This is true for continuous, uncorrelated (Bernoulli) noise on the physical qubits with arbitrary on-site error probability $p_0^{\times}$—except for the singular, completely mixing channel with $p_0^{\times} = \frac{1}{2}$; there is no non-trivial error threshold, in contrast to "true" two-dimensional MWPM for the toric code [30,40]. However, global majority voting violates locality as it requires *space* for each logic gate and *time* for communication between them. This raises the question whether this extraordinary robustness of majority voting survives in realistic setups. In Subsec. 2.3 we argue that low-level decoders of quantum memories must be realized *in hardware* and close to the coherent subsystem (here the Majorana chain) to allow for modularity and scalability, both in the number of chains and their length. Then, collecting the syndromes of an extended chain in a central processing unit, and distributing corrections afterwards requires *time*—which scales with the system size $L$. We demonstrate that this important feature of global majority voting precludes its application at a fixed rate for $L \to \infty$, and thereby spoils the favorable scaling of decay times.

This line of thought motivates our search for a manifestly *local* decoder of the MCQC, taking finite communication speed and spatial extent seriously. Then, locality implies that restrictions on the time granted for decoding translates into restrictions on the syndromes that can influence a local correction. We derive a generic upper bound on the success probability for decoding the MCQC with local decoders and discuss implications for the scaling of the decoding time with the chain length.

After setting the scene (and sketching what we *can* expect and what we *cannot*), we aim for a feasible local decoder of the MCQC. To this end, Subsec. 3.1 introduces the concept of *cellular*

*automata* (CA) as well-developed prime example for physically realistic local computation. The natural invariance of local CA rules in space narrows down the choice of local decoders but allows for implementations that can be scaled up easily. While CAs naturally operate on classical bits, the physical qubits of the MCQC are not accessible—only the syndromes can be measured without perturbing the state (we call this the "quantum handicap"). We argue that only CAs featuring a particular symmetry (called *self-duality*) can be employed as MCQC-decoders.

To decode the MCQC by means of a CA, implementing a *global* majority vote by *local* rules seems a good approach. This task is known as *density classification problem* [41, 42] and has been shown to be unsolvable for binary CAs in any dimension [43]. In Subsec. 3.2 we review some of the results on *approximate* density classifiers which could provide viable replacements for perfect majority voting if error rates are small (i.e., away from $p_0^{\chi} = \frac{1}{2}$). We present two binary CAs that are known to perform well on density classification, one of which (called TLV) is self-dual; it can be rewritten in a form that complies with the "quantum handicap": It naturally takes syndromes as input and produces correction operations as output. Before we can explore the performance of TLV as MCQC-decoder, the question of boundary conditions has to be addressed. It is common to place CAs on finite chains with *periodic* boundaries. In Subsec. 3.3 we point out that this is not compatible with *locality* of classical computations on the one hand and the necessity of a *stretched* quantum chain on the other. Hence a modification of TLV at the boundaries is required (denoted by $\overline{\text{TLV}}$). We demonstrate that for MCQC-decoding, *mirrored* boundary conditions are the way to go: The CA operates in a cavity-like geometry to pair quasiparticles with partners in the edge modes of the MCQC.

In Subsec. 4.1 we start our analysis of $\overline{\text{TLV}}$ with a numerical evaluation of its decoding capabilities. Sampling uncorrelated Bernoulli random configurations with on-site error probability $p_0^{\chi}$ and subsequent evolution with $\overline{\text{TLV}}$ allows us to gauge the possible downsides of performing only *approximate* majority voting. Despite the existence of periodic cycles that cannot be decoded, numerics suggests that for $p_0^{\chi} < \frac{1}{2}$ only an exponentially (in $L$) small fraction of error patterns fails to be corrected successfully. Moreover, the typical time needed to rotate an error-afflicted instance of the MCQC back into the codespace grows sublinearly with the chain length $L$ (in contrast to *global* majority voting). To substantiate these claims, we apply the concept of *sparse errors* to the particular case of $\overline{\text{TLV}}$. In Subsec. 4.2 we derive a central statement of this work: The probability to decode a length-$L$ MCQC successfully with $\overline{\text{TLV}}$ after $t \propto L^{\kappa}$ time steps (with $\kappa > 0$ arbitrary) tends to 1 exponentially fast for $L \to \infty$ and small but finite error probabilities $p_0^{\chi} \ll \frac{1}{2}$. This provides us with a much simpler and faster decoder than global majority voting (the decoding time of which scales linearly with $L$), and especially implies that for these error probabilities the "expensive" global nature of majority voting is not required for efficient decoding.

In the remainder, we shift our focus from the *decoding* of static error patterns to the *protection* of the MCQC in the presence of continuous (Bernoulli) noise. In Subsec. 5.1 we realize this scenario by applying the local rules of $\overline{\text{TLV}}$ and on-site errors with probability $p_0^{\chi}$ alternately. We demonstrate numerically that $\overline{\text{TLV}}$ *cannot* cope with such perturbations of its evolution in that the lifetime of the logical qubit grows only subexponentially with the chain length. In the light of known results on the behavior of one-dimensional CAs, this is unfortunate but not surprising: Simple, one-dimensional CAs subject to noise are expected to be ergodic; this is known as the *positive rates conjecture* [44]—it is well-established that this conjecture is incorrect, but the only known counterexample is extraordinary complex [26, 45, 46], and we cannot expect that our setup is a simpler one.

If one abandons strictly one-dimensional decoders (with circuit complexity $\sim L$), there is a rather generic solution to this problem: Any given decoder can be employed to counter continuous noise by repetitive applications with a fixed rate. If the time required to decode a fixed

error pattern grows with the code size $L$, so does the number of required instances running in parallel to prevent errors from accumulating. Thus the additional hardware overhead due to continuous noise correlates with the decoding time for fixed error patterns. In Subsec. 5.2 we follow this idea and stack copies of $\overline{\text{TLV}}$ in the second dimension perpendicular to the quantum chain. The *depth* of this classical circuit quantifies the hardware overhead required for the retention of the logical qubit in the presence of noise; as it directly relates to the decoding time of $\overline{\text{TLV}}$, it grows sublinearly with the chain length, so that shallow circuits suffice for reasonably low error rates. Indeed, the complexity of these circuits scales with $L^{1+\kappa}$ for $0 < \kappa < 1$, in contrast to the typical $L^2$-scaling of global majority voting.

## 2  1D topological quantum codes

We start this section with a description of the Majorana chain and thereby review the realization of a topological quantum code as degenerate ground state space of a local Hamiltonian. In particular, we revisit the procedure of quantum error correction using syndrome measurements and demonstrate that it reduces to global majority voting in this particular case. This decoding scheme features an exponentially growing lifetime of the encoded logical qubit with the chain length $L$. However, global majority voting violates locality as it relies on the evaluation of a function of spatially distributed syndrome measurements. But the time required for collecting the syndromes of an extended chain in a central processing unit (and distributing corrections afterwards) scales with the system size $L$. We conclude by demonstrating that taking into account this processing time eliminates the exponential scaling of the qubit lifetime. This sets the stage for the construction and study of a strictly local, inherently scalable replacement for global majority voting.

### 2.1  Majorana chain

The simplest example of a topological quantum error correction code in one dimension is given by the degenerate ground state manifold of the paradigmatic Majorana chain [6], see Fig. 1 (a). The Hamiltonian for an open chain of $L$ spinless fermions $c_i$ reads

$$H_{\text{MC}} = \sum_{i=1}^{L-1} \left( w_i\, c_i^\dagger c_{i+1} + \Delta_i\, c_i c_{i+1} + \text{h.c.} \right) + \sum_{i=1}^{L} \mu_i \left( c_i^\dagger c_i - \frac{1}{2} \right), \tag{1}$$

where $c_i, c_i^\dagger$ denote fermionic annihilation and creation operators, $w_i$ is the tunneling amplitude, $\Delta_i$ the superconducting gap parameter, and $\mu_i$ denotes the chemical potential. At the "sweet spot", $\mu_i = 0$ and $w_i = -\Delta_i = 1$, the Hamiltonian takes the form

$$H_{\text{MC}} = i\sum_{j=1}^{L-1} \gamma_{2j}\gamma_{2j+1} = -\sum_{j=1}^{L-1} S_j, \tag{2}$$

with Majorana fermions $\gamma_{2j} \equiv i(c_j^\dagger - c_j)$ and $\gamma_{2j-1} \equiv c_j^\dagger + c_j$, $\gamma_j^\dagger = \gamma_j$ and $\{\gamma_i, \gamma_j\} = 2\delta_{ij}$. The operators $S_j = -i\gamma_{2j}\gamma_{2j+1} = (-1)^{\tilde{c}_j^\dagger \tilde{c}_j}$ measure the parity of the localized quasiparticle modes $\tilde{c}_j^\dagger$ above the superconducting condensate and play the role of a stabilizer known from quantum information theory [7, 47]: $[S_i, S_j] = 0$, $S_j^\dagger = S_j$, and $S_j^2 = \mathbb{1}$. Let $\mathscr{S} = \langle\{S_1, \ldots, S_{L-1}\}\rangle$ be the (Abelian) stabilizer group. The codespace $\mathscr{C} \equiv \{|\Psi\rangle \in \mathscr{H} \,|\, \mathscr{S}\,|\Psi\rangle = |\Psi\rangle\}$ is the $\mathscr{S}$-invariant subspace and encodes a single logical qubit, $\dim \mathscr{C} = 2$; this space is equivalent to the degenerate ground state space of Eq. (2). This observation can be understood as follows: Since the edge Majorana modes $\gamma_L \equiv \gamma_1$ and $\gamma_R \equiv \gamma_{2L}$ are missing in Hamiltonian (2), one

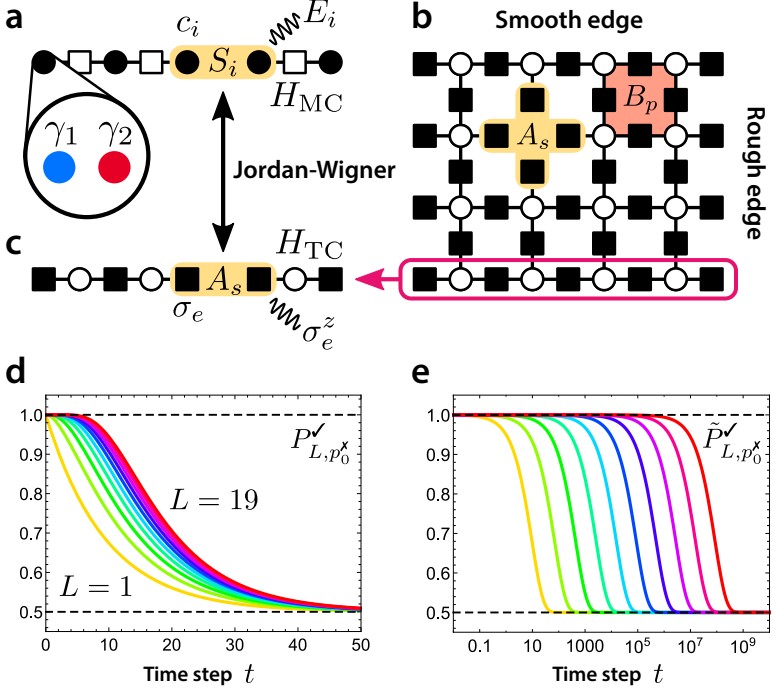

Figure 1: *1D quantum memories and global majority voting.* **a** The Majorana chain with fermions $c_i$ on sites, composed of Majorana fermions $\gamma_j$, parity symmetric errors $E_i$ and syndrome measurements $S_i$. **b** The toric code on a planar square lattice with rough and smooth edges. Stabilizers $A_s$ ($B_p$) act on spins on edges adjacent to sites $s$ (faces $p$). **c** The degenerate $L \times 1$ toric code is the Jordan-Wigner transform of the Majorana chain. Thus it features the same error syndromes and correction schemes, namely majority voting. **d** The probability $P^{\checkmark}_{L,p_0^{\times}}$ of correctly decoding an ensemble of $L$ bits by global majority voting under continuous noise $p_0^{\times} = 0.05$ as a function of time $t$ for different system sizes $L = 1, \ldots, 19$ without any stabilizing correction performed. **e** The same ($\tilde{P}^{\checkmark}_{L,p_0^{\times}}$) for a stabilized system with global majority voting and active correction after each time step. Mind the logarithmic $t$-axis.

finds that $\Sigma^z = S_{\text{edges}} \equiv -i\gamma_L\gamma_R$ acts on the ground state space $\mathscr{C}$ as $\left[\Sigma^z, S_j\right] = 0$. Furthermore, it allows for the definition of a convenient basis of the code space, namely

$$\Sigma^z \left|\pm 1\right\rangle = \pm \left|\pm 1\right\rangle. \tag{3}$$

Flipping the encoded qubit is possible via the edge modes $\Sigma^x \equiv \gamma_L$ and $\Sigma^y \equiv \gamma_R$, e.g.,

$$\Sigma^x \left|\pm 1\right\rangle = \left|\mp 1\right\rangle \tag{4}$$

without violating any stabilizer constraint, $\left[\Sigma^{x,y}, S_j\right] = 0$. The operators $\Sigma^\alpha$ characterize the logical qubit completely as they realize the Pauli algebra $\left[\Sigma^\alpha, \Sigma^\beta\right] = 2i\varepsilon_{\alpha\beta\gamma}\Sigma^\gamma$ on $\mathscr{C}$.

Crucial for realizing a quantum memory is its resilience against depolarizing and dephasing noise. The Majorana chain fights these types differently: Depolarizing (bit-flip) noise cannot be suppressed by the Hamiltonian since the logical operators $\Sigma^x$ and $\Sigma^y$ are perfectly local in any embedding of the open chain and energetically not penalized by the Hamiltonian in Eq. (2). However, in terms of the fermions it is $\Sigma^x = c_1^\dagger + c_1$ and $\Sigma^y = i(c_L^\dagger - c_L)$—operators which break the fermionic parity symmetry of the superconducting Hamiltonian. In superconducting systems, fermionic parity is considered a natural symmetry that can be enforced to high precision because fermions are created by breaking cooper pairs (though it *can* be violated by quasiparticle poisoning [37, 38]). In that sense, the fermionic nature of the physical

realization is exploited to suppress depolarizing errors. Strictly speaking, this is just symmetry protection.

In contrast, dephasing noise operates as $\Sigma^z \propto \gamma_L \gamma_R$ on the chain which is a non-local operator that cannot be induced directly by a noisy environment respecting locality. Indeed, the generic form of environmental noise that is both local and parity-symmetric has the form $E_j = -i\gamma_{2j-1}\gamma_{2j}$ (note that pairs shifted by a single site act trivially on the code space). Since $\{E_i, S_j\} = 0$ if and only if $j = i$ or $j = i-1$ (otherwise $[E_i, S_j] = 0$), a single error $E_i$ is flanked by a pair of syndromes or *charges* with $S_i = -1 = S_{i-1}$. From the condensed matter point of view, this accounts for breaking a cooper pair and lifting the localized quasiparticles above the superconducting gap. Subsequent errors can move and/or create additional quasiparticle pairs that can, as time goes on, traverse the macroscopic chain in a noise-driven, diffusive process. Once a pair of charges traverses the whole chain, it is described by

$$\prod_{j=1}^{L} E_j = -i\gamma_L \left[ \prod_{i=1}^{L-1} S_i \right] \gamma_R = \Sigma^z , \tag{5}$$

where we used $S_i = \mathbb{1}$ on the code space. Thus dephasing noise on the logical qubit is only possible if quasiparticles travel freely through the system. Unfortunately, save for the energy gap which penalizes the *creation* of charges, there is no cost for *moving* them. This deconfinement renders the Hamiltonian theory unstable at finite temperatures (this is related to the fact that there is no phase transition for the one-dimensional classical Ising model).

To protect the logical qubit from dephasing, active error correction must be employed. Assume the system is initialized in state $|\Psi\rangle \in \mathscr{C}$ and subsequently has been affected by the error $E(\mathbf{x}) = \prod_j E_j^{x_j}$, encoded by the binary vector $\mathbf{x} = (x_1, \ldots, x_L) \in \mathbb{Z}_2^L$. Because $E_i^2 = \mathbb{1}$, applying the same error twice cancels the latter and removes all syndromes,

$$E(\mathbf{x})E(\mathbf{x})|\Psi\rangle = E(\mathbf{x} \oplus \mathbf{x})|\Psi\rangle = E(\mathbf{0})|\Psi\rangle = |\Psi\rangle , \tag{6}$$

rotating the system's state back into the code space $\mathscr{C}$. Here, $\oplus$ denotes the (element-wise) modulo-2 addition. To infer $\mathbf{x}$, the local stabilizers $\{S_1, \ldots, S_{L-1}\}$ are measured periodically to yield a binary syndrome pattern $\mathbf{s} = (s_{1+\frac{1}{2}}, \ldots, s_{L-1+\frac{1}{2}}) \in \mathbb{Z}_2^{L-1}$ (with $S_j = (-1)^{s_{j+\frac{1}{2}}}$) that indicates the boundaries of $E(\mathbf{x})$. (Recall that only projective measurements that leave $\mathscr{C}$ invariant do not destroy the logical qubit—these are exactly the stabilizer generators.) In terms of binary vectors, this reads

$$\mathbf{s} = \partial \mathbf{x} \quad \text{with} \quad (\partial \mathbf{x})_{i+\frac{1}{2}} \equiv x_i \oplus x_{i+1} . \tag{7}$$

The index shift by $\frac{1}{2}$ for syndromes is purely formal to distinguish them from error patterns $x_i$. Inferring $\mathbf{x}$ from $\mathbf{s}$ is complicated by the fact that $\partial \mathbf{x} = s = \partial \mathbf{x}^c$, where $(\mathbf{x}^c)_i = x_i \oplus 1$ is the element-wise binary complement. The decoding problem is therefore not unique as complementary error strings share the same syndrome. If $\mathbf{x}^c$ is chosen to specify the correction, one has

$$E(\mathbf{x}^c)E(\mathbf{x})|\Psi\rangle = E(\mathbf{x}^c \oplus \mathbf{x})|\Psi\rangle = E(\mathbf{1})|\Psi\rangle = \Sigma^z |\Psi\rangle \tag{8}$$

and thereby (unknowingly) applies a quantum gate on the stored qubit; this follows from Eq. (5). Thus it is of paramount importance to choose the correct error pattern. The optimal decoding strategy depends on the error channel that gave rise to $E(\mathbf{x})$. Here we will always assume $\mathbf{x}$ to be a sequence of uncorrelated Bernoulli random variables $x_i$ with parameter $0 \le p_0^{\mathsf{x}} \le \frac{1}{2}$ so that $\Pr(x_i = 1) = p_0^{\mathsf{x}}$ for all $i = 1, \ldots, L$. Then, the provably best decoder $\Delta$ is (global) *majority voting*,

$$\Delta(\mathbf{s}) \equiv \mathbf{y} \quad \text{with} \quad \partial \mathbf{y} = \mathbf{s} \quad \text{and} \quad |\mathbf{y}| < |\mathbf{y}^c| \tag{9}$$

which realizes maximum-likelihood decoding for repetition codes, i.e., $\mathbf{y}$ is preferred over $\mathbf{y}^c$ because the former requires less errors ($x_i = 1$) and this makes it more probable with respect to a Bernoulli distribution with $p_0^{\mathbf{x}} < \frac{1}{2}$. In the context of quantum codes (in particular the toric code), the prescription (9) is also called *minimum-weight perfect matching*, which is equivalent to majority voting in one dimension (see below). Here, the weight $|\mathbf{x}|$ is the number of non-zero components $x_i = 1$ and we require $L$ to be odd to avoid ties ($|\mathbf{x}| = |\mathbf{x}^c|$). Note that $\Delta$ indeed performs majority voting on $\mathbf{x}$ in the sense that

$$\mathbf{x} \oplus \Delta(\partial \mathbf{x}) = \begin{cases} \mathbf{x} \oplus \mathbf{x} & = \mathbf{0} & \text{if} & |\mathbf{x}| \leq \frac{L-1}{2} \\ \mathbf{x} \oplus \mathbf{x}^c & = \mathbf{1} & \text{if} & |\mathbf{x}| \geq \frac{L+1}{2} \end{cases} \tag{10a}$$

$$= \mathbf{maj}[x_1, \ldots, x_L] . \tag{10b}$$

The majority function on $L$ binary inputs $x_i$ is defined as

$$\mathrm{maj}[x_1, \ldots, x_L] \equiv \left\lfloor \frac{1}{2} + \frac{1}{L}\left(\sum_{i=1}^{L} x_i - \frac{1}{2}\right) \right\rfloor \tag{11}$$

(ties evaluate to 0 with this definition) and the bold version $\mathbf{maj}[\bullet]$ indicates a vectorized result with each entry given by $\mathrm{maj}[\bullet]$; $\lfloor x \rfloor$ denotes the greatest integer less than or equal to $x$.

We conclude that the "quantum handicap" of having only access to the syndrome $\mathbf{s}$ for decoding the topological code does not change the decoding strategy as compared to a *classical* repetition code. Indeed, for a correctable binary error pattern $\mathbf{x}$ ($|\mathbf{x}| < |\mathbf{x}^c| \Leftrightarrow \mathbf{maj}[\mathbf{x}] = \mathbf{0}$), the classical repetition codeword $\Psi \in \{\mathbf{0}, \mathbf{1}\}$, and the quantum codeword $|\Psi\rangle \in \mathscr{C}$, we have for the *classical* code

$$\Psi \xrightarrow{\text{E}} \mathbf{x} \oplus \Psi = \Psi' \tag{12a}$$

$$\xrightarrow{\text{C}} \Delta(\partial \Psi') \oplus \Psi' = \Delta(\partial \mathbf{x}) \oplus \mathbf{x} \oplus \Psi = \mathbf{maj}[\mathbf{x}] \oplus \Psi = \Psi \tag{12b}$$

and the *quantum* analogue

$$|\Psi\rangle \xrightarrow{\text{E}} E(\mathbf{x})|\Psi\rangle \tag{13a}$$

$$\xrightarrow{\text{C}} E(\Delta(\partial \mathbf{x}))E(\mathbf{x})|\Psi\rangle = E(\Delta(\partial \mathbf{x}) \oplus \mathbf{x})|\Psi\rangle = E(\mathbf{maj}[\mathbf{x}])|\Psi\rangle = |\Psi\rangle , \tag{13b}$$

where E (C) denotes application of errors (corrections).

To make connection with another well-known topological quantum code, let us, just for a second, peek into the second dimension: There, the simplest model is given by the toric code [8, 9, 48] which features a four-fold degenerate ground state manifold (the code space) and is defined by the Hamiltonian

$$H_{\text{TC}} = -\sum_{\text{Sites } s} A_s - \sum_{\text{Faces } p} B_p , \tag{14}$$

with stabilizer operators

$$A_s = \prod_{e \in s} \sigma_e^x \quad \text{and} \quad B_p = \prod_{e \in p} \sigma_e^z \tag{15}$$

living on an $L_x \times L_y$ square lattice with periodic boundary conditions and spin-$\frac{1}{2}$ representations $\sigma_e$ (physical qubits) on the edges. While the toroidal geometry of Hamiltonian (14) is crucial for its four-fold ground state degeneracy, it also renders the model experimentally challenging (even more than it already is due to its four-spin interactions). However, if the above Hamiltonian is adapted to a *planar* square lattice with appropriately chosen ("rough"

and "smooth") open boundaries [49], the experimental implementation becomes more attractive while the ground state manifold is still two-fold degenerate and constitutes a topological quantum memory with the two Abelian anyonic excitations $A_s = -1$ and $B_p = -1$, see Fig. 1 (b).

Reducing this code to the degenerate, one-dimensional case $L_x = L$ and $L_y = 1$, yields a 1D spin system which maps directly to the Majorana chain under the Jordan-Wigner transformation

$$\gamma_{2j} \quad \longleftrightarrow \quad \left[\prod_{i=1}^{j-1} \sigma_i^z\right] \sigma_j^y, \tag{16a}$$

$$\gamma_{2j-1} \quad \longleftrightarrow \quad \left[\prod_{i=1}^{j-1} \sigma_i^z\right] \sigma_j^x, \tag{16b}$$

[Fig. 1 (c)] with the identifications

$$S_j \quad \longleftrightarrow \quad \sigma_j^x \sigma_{j+1}^x \, (= A_s), \tag{17a}$$

$$E_j \quad \longleftrightarrow \quad \sigma_j^z. \tag{17b}$$

Note that in one dimension there are no faces and the $B_p$ stabilizers are absent. Then, Hamiltonian (14) describes the 1D Ising model and local errors $E_j = \sigma_j^z$ correspond to spin-flips in the $\sigma^x$-basis; syndromes $S_j = \sigma_j^x \sigma_{j+1}^x = -1$ can be associated with domain walls.

The physical distinction between Majorana chain and 1D toric code/Ising chain becomes evident if one realizes that error strings $E(\mathbf{x})$ can be directly measured by $\sigma_j^x = \pm 1$ in the 1D toric code/Ising chain (and not only their endpoints by $S_j = \pm 1$). The analogous operator for the Majorana chain error strings violates the fermionic parity and is thereby suppressed. Similarly, while $\Sigma^x = \gamma_L$ is forbidden in the fermionic setting of the Majorana chain due to parity superselection, there is no natural symmetry in the spin chain preventing $\Sigma^x = \sigma_1^x$ from depolarizing the logical qubit. This is why it is legit to call the Majorana chain a 1D topological *quantum* memory whereas the mathematically equivalent 1D toric code/Ising chain only protects a *classical* bit by realizing a repetition code.

Nevertheless, from the algorithmic point of view, both theories carry the same syndromes and therefore can be corrected with the same algorithms. In particular, MWPM on the toric code degenerates into majority voting on the Majorana chain. For the degenerate toric code, this active error correction procedure has already been demonstrated experimentally with transmon qubits [11, 12].

## 2.2 Global majority voting

We proceed with a brief analysis of global majority voting. As we argued above, we can ignore the "quantum handicap" that restricts our knowledge to the endpoints of error strings (the syndromes) and instead work with the actual error patterns.

Assume a classical bit $x$, initialized as $x = 0$, is flipped by a (unbiased) Bernoulli process with probability $0 \le p_0^x \le \frac{1}{2}$ per time step $\delta t$. If we think of the state $x = 0$ as the "clean" one while $x = 1$ indicates as site that is error-afflicted, the probability to find $x = 1$ after $t$ time steps is given by

$$p^x(t) = \frac{1}{2}\left[1 - \left(1 - 2p_0^x\right)^t\right], \tag{18}$$

which renormalizes to the completely mixed state $p^x(t) \to \frac{1}{2}$ exponentially fast whenever $0 < p_0^x < 1$. If we copy the bit $L$ times, $(x_1, \ldots, x_L)$, and encode a logical bit $X$ via a simple repetition code,

$$X = 0 \quad \to \quad \mathbf{x} = (0, 0, \ldots, 0) \tag{19}$$

then, for uncorrelated Bernoulli error processes on the physical bits $x_i$, the best decoder is given by the global majority vote

$$X = \text{maj}[x_1, \ldots, x_L] \,. \tag{20}$$

An erroneous logical bit $X = 1$ occurs whenever the majority is altered by the local errors. Formally, the probability to find $X = 1$ after $t$ time steps of accumulating errors is given by ($L$ odd)

$$P^{\textrm{✗}}_{L,p_0^{\textrm{✗}}}(t) = \sum_{k=\frac{L+1}{2}}^{L} \binom{L}{k} \left[ p^{\textrm{✗}}(t) \right]^k \left[ 1 - p^{\textrm{✗}}(t) \right]^{L-k} \tag{21a}$$

$$= \frac{L+1}{2} \binom{L}{\frac{L+1}{2}} \int_0^{p^{\textrm{✗}}(t)} [x - x^2]^{\frac{L-1}{2}} \mathrm{d}x \,, \tag{21b}$$

where in the last line we used the regularized incomplete beta function to express the cumulative Binomial distribution in a closed form [50] (see the Appendix A for details). We illustrate $P^{\textrm{✓}}_{L,p_0^{\textrm{✗}}}(t) = 1 - P^{\textrm{✗}}_{L,p_0^{\textrm{✗}}}(t)$ as a function of $t$ in Fig. 1 (d) for different system sizes $L$ and fixed continuous noise $p_0^{\textrm{✗}}$. Note that the decay time grows very slowly with the system size $L$.

After a *single* time step, we have the logical failure probability $P^{\textrm{✗}}_{L,p_0^{\textrm{✗}}} \equiv P^{\textrm{✗}}_{L,p_0^{\textrm{✗}}}(t = 1)$. Assume that after each time step $\delta t$ the errors that occurred during $\delta t$ are immediately countered by majority voting, i.e., following Eq. (12) for classical and Eq. (13) for quantum codes. The probability of the logical (qu)bit to be in its original state after $t$ time steps is then

$$\tilde{P}^{\textrm{✓}}_{L,p_0^{\textrm{✗}}}(t) = \frac{1}{2} \left[ 1 + \left( 1 - 2P^{\textrm{✗}}_{L,p_0^{\textrm{✗}}} \right)^t \right] \,, \tag{22}$$

which yields the timescale $T_{L,p_0^{\textrm{✗}}}$ for the logical information loss

$$T_{L,p_0^{\textrm{✗}}} = \left[ \log \frac{1}{1 - 2P^{\textrm{✗}}_{L,p_0^{\textrm{✗}}}} \right]^{-1} \xrightarrow{L \to \infty} \frac{1}{P^{\textrm{✗}}_{L,p_0^{\textrm{✗}}}} \tag{23}$$

if $\lim_{L \to \infty} P^{\textrm{✗}}_{L,p_0^{\textrm{✗}}} = 0$. Moreover, it is straightforward to show that it diverges exponentially with the system size for any non-trivial (and non-critical) microscopic error probability $0 < p_0^{\textrm{✗}} < \frac{1}{2}$. Indeed, we can use Eq. (21b) to derive the upper bound

$$P^{\textrm{✗}}_{L,p_0^{\textrm{✗}}} \leq \frac{L+1}{2} \binom{L}{\frac{L+1}{2}} p_0^{\textrm{✗}\, L-1} \sim \sqrt{L}\, e^{L \log(2q)} \,, \tag{24}$$

with $q = \sqrt{p_0^{\textrm{✗}}(1 - p_0^{\textrm{✗}})} < \frac{1}{2}$ for $p_0^{\textrm{✗}} < \frac{1}{2}$; in the last step, we used the asymptotic approximation $\binom{L}{\frac{L+1}{2}} \sim \sqrt{\frac{2}{\pi}} \frac{2^L}{\sqrt{L}}$ for $L \to \infty$. It follows the exponentially diverging decay time of the code

$$T_{L,p_0^{\textrm{✗}}} \gtrsim \frac{e^{L \log \frac{1}{2q}}}{\sqrt{L}} \,. \tag{25}$$

This is illustrated in Fig. 1 (e) by plotting $\tilde{P}^{\textrm{✓}}_{L,p_0^{\textrm{✗}}}(t)$ over time for different system sizes and fixed continuous error rate. Eq. (25) is a quantitative manifestation of the perfect decoding properties of global majority voting on a repetition code. Note that the only error rate for which decoding fails in the thermodynamic limit is the singular point $p_0^{\textrm{✗}} = \frac{1}{2}$ for which $2q = 1 \Rightarrow \log \frac{1}{2q} = 0$.

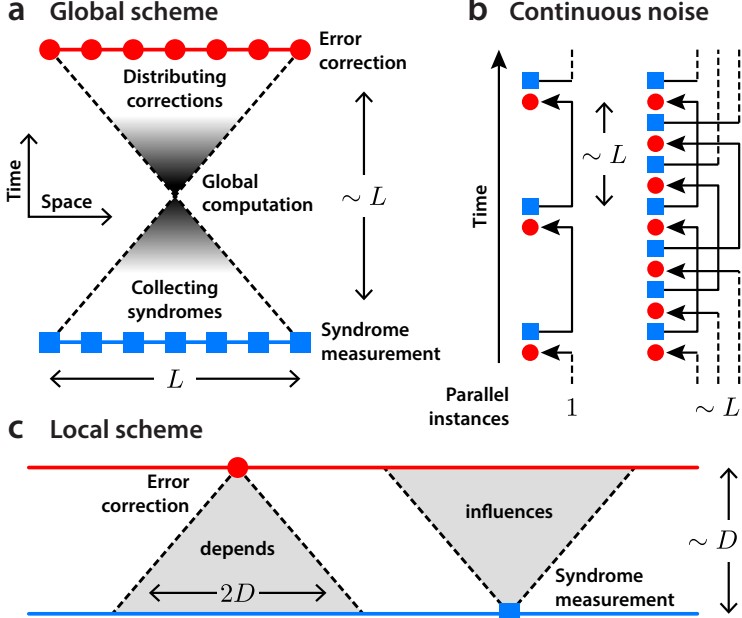

Figure 2: *Locality constraints.* **a** A global correction scheme on a quantum code of linear size $L$ requires the syndrome data to be merged, processed, and afterwards the results to be distributed again. The finite communication speed makes the time between syndrome measurements (blue squares) and error corrections (red circles) scale with $L$ in the best case. **b** With constrained hardware overhead, only one instance of the syndrome processing runs at once, giving rise to intervals without correction growing with $\sim L$. Repeating syndrome measurements and corrections with a period independent of $L$ requires $\sim L$ instances running in parallel, thus increasing the hardware overhead dramatically. **c** Local schemes can be used to keep hardware overhead in check by reducing the computation time needed to sublinear (or even logarithmic) scaling if $D \sim L^\kappa$ with $k < 1$. This restricts the syndromes a local correction operation depends on to a subsystem inside the past light cone of diameter $2D$, and allows syndromes to influence only corrections in their future light cone.

## 2.3 Constraints by locality

Eq. (25) tells us that global majority voting is a very powerful decoding scheme for the 1D quantum code realized by the Majorana chain: its critical error rate $p_c^{\mathsf{X}} = \frac{1}{2}$ is optimal. In physics, nothing is for free. This begs the question what it is that we are paying with by employing the global majority decoder $\Delta$ (or, equivalently, the function $\mathrm{maj}[\dots]$). One particularly expensive feature of $\mathrm{maj}[x_1, \dots, x_L]$ is its global nature: It depends non-trivially on all $L$ inputs while their number grows with the system size [see Eq. (11)]. Indeed, one needs to take into account *at least* $\frac{L+1}{2}$ of the inputs to be sure about the majority; for generic inputs even more. This makes the evaluation of $\mathrm{maj}[\dots]$ a relevant factor that has to be taken into account when the scaling of the quantum code with $L$ is addressed.

A generic global function, depending on $\sim L$ spatially distributed inputs (syndromes) requires *at least* $\sim c^{-1}L$ time steps to gather its input data (for a 1D geometry). Here $c$ denotes the speed of classical information propagation in the auxiliary systems framing the quantum chain. This is illustrated by the light cone in Fig. 2 (a). In addition, the evaluation itself (shaded region) also requires *at least* $\mathcal{O}(L)$ time steps because every input has to be read at least once, see e.g., Eq. (11). Depending on the decoder, the latter may be improved by parallelization (which, in turn, is payed for by additional hardware overhead), whereas the former

argument remains valid as it is based on physical constraints alone.

An immediate consequence for the global majority decoder $\Delta$ is sketched in the left panel of Fig. 2 (b): The time between syndrome measurement (blue square) and correction (red disk) scales with the system size $L$. Depending on the relevant velocity $c$ (which should be henceforth thought of as comprising both information propagation and computations) and chain length $L$, this upper-bounds the rate at which $\Delta$ can be applied to fight continuous noise on the quantum code.

This has important consequences: The probability that $\Delta$ flips the logical (qu)bit after accumulating errors for $t = c^{-1}L$ time steps is given by Eq. (21b),

$$\hat{P}^{\times}_{L,p_0^{\times}} \equiv P^{\times}_{L,p_0^{\times}}(t = c^{-1}L) \xrightarrow{L \to \infty} \frac{1}{2}, \tag{26}$$

where the limit holds for all $0 < p_0^{\times} \le \frac{1}{2}$, see Appendix A for the derivation. This is in contrast to

$$P^{\times}_{L,p_0^{\times}} = P^{\times}_{L,p_0^{\times}}(t = 1) \xrightarrow{L \to \infty} 0, \tag{27}$$

which led to the exponential growth of $T_{L,p_0^{\times}}$ if the correction rate is *independent* of the system size, see Eq. (24). We conclude that the exponential growth of $T_{L,p_0^{\times}}$ (which depends on the exponential vanishing of the logical error probability $P^{\times}_{L,p_0^{\times}}$) is lost if we take into account the time needed to evaluate the global majority vote.

A possibility to keep both a size-independent correction rate and the global majority vote decoder $\Delta$ is illustrated in the right panel of Fig. 2 (b): Multiple copies of $\Delta$ running in parallel can keep up with continuous noise if after each time step $\delta t$ a new instance of $\Delta$ is fed with the syndrome $\partial(\mathbf{x}_{t-1} \oplus \mathbf{x}_t) = \partial \mathbf{x}_{t-1} \oplus \partial \mathbf{x}_t$ that encodes only errors accumulated during $\delta t$. Note that intertwining corrections and errors is acceptable as both commute. The obvious downside of this approach is its hardware overhead: The number of parallel instances required (the "depth" of the decoder) scales with the time needed for a single instance to finish, that is, with $L$.

If we retrace our line of thought, it is obvious that the *global* nature of $\Delta$ is responsible for the $L$-scaling of the depth in the presence of continuous noise. This motivates the question whether the global decoder $\Delta$ can be replaced by a *local* version $\Delta^D$ which requires only syndrome data within a radius $D$ of each site to compute the correction at this very site; the corresponding spacetime diagram is shown in Fig. 2 (c). The benefits of such a local decoder would be less hardware overhead, simpler implementation, and thus better scaling properties. It cannot implement $\text{maj}[\dots]$ perfectly and one has to expect decoding errors in some cases (where $\Delta$ would have succeeded). However, if these cases are rare for low error rates $p_0^{\times} < p_c^{\times}$ with finite critical rate $0 < p_c^{\times} \le \frac{1}{2}$, and the relaxation time $T_{L,p_0^{\times}}$ still scales exponentially with $L$, this would be perfectly acceptable. It is such a "$D$-local" decoder that we describe and analyze in this manuscript:

**Definition 1.** *Given a decoder $\Delta : \mathbb{Z}_2^{L-1} \to \mathbb{Z}_2^L$ for the Majorana chain of length $L$ that maps a syndrome pattern $\partial \mathbf{x} \in \mathbb{Z}_2^{L-1}$ to the correction string $\Delta(\partial \mathbf{x}) \in \mathbb{Z}_2^L$. Let*

$$\pi_i^D \mathbf{x} = \left( x_{\max\{i-D,1\}}, \dots, x_i, \dots, x_{\min\{i+D,L\}} \right) \tag{28}$$

*denote the neighborhood of site $i$ with radius $D$.*

*The decoder $\Delta$ is called D-local (write $\Delta^D$) if*

$$\Delta_i^D(\partial \mathbf{x}) = f_i(\partial \pi_i^D \mathbf{x}) \tag{29}$$

*for some family of functions $\{f_i\}$: Its correction at site $i$ only depends on (syndromes of) error patterns within distance $D$ of $i$.*

Since $D$-local decoders finish after $\sim D$ time steps (if the $f_i$ can be evaluated efficiently), the required depth [cf. Fig. 2 (c)] also scales with $D$.

In the remainder of this subsection, and before we zoom in on our particular decoder, we discuss a constraint that follows for the class of $D$-local decoders $\Delta^D$ quite generally. Namely the (weak) upper bound for the probability $P_{\mathrm{dec}}^{\checkmark}$ of successfully decoding Bernoulli samples with a $D$-local decoder

$$P_{\mathrm{dec}}^{\checkmark} \le \left[ 1 + \left( \frac{p_0^{\boldsymbol{x}}}{1 - p_0^{\boldsymbol{x}}} \right)^{2D+1} \right]^{-\frac{L}{2D+1}}. \tag{30}$$

Note that this result is generic in the sense that it holds for all decoders of the MCQC where the correction of site $i$ depends only on nearby syndromes in the neighborhood $\pi_i^D \mathbf{x}$, irrespective of their local functions $\{f_i\}$. We call Eq. (30) the *light cone constraint*; its proof can be found in Appendix B.1.

Here we discuss some scaling limits of Eq. (30) and their implications for potential decoders replacing $\Delta$. We assume $D = D(L)$ to be a function of the linear size $L$ of the code (the length of the quantum chain). We stress that the interpretation of the radius $D$ can be either a spatial depth of a feed-forward physical circuit or a time-like depth in a spacetime diagram of a truly one-dimensional physical automaton. In the first case, scaling $D$ with $L$ means growing the system into the second dimension; in the second case, it accounts for a longer runtime of the decoder. Note that the class of decoders with at least $D \sim L$ comprises exactly the global ones (e.g., $\Delta$).

We discuss two important cases:

**Case 1:** $D = $ const. This describes a truly one-dimensional feed-forward circuit of finite depth $D$. We find in the thermodynamic limit

$$\lim_{L \to \infty} P_{\mathrm{dec}}^{\checkmark} \le \begin{cases} 0 & \text{for} \quad 0 < p_0^{\boldsymbol{x}} \le \frac{1}{2} \\ 1 & \text{for} \quad p_0^{\boldsymbol{x}} = 0 \end{cases}, \tag{31}$$

i.e., there is no successful decoding possible for any finite microscopic error rate $p_0^{\boldsymbol{x}} > 0$.

**Case 2:** $2D + 1 \sim L^{\kappa}$ ($\kappa > 0$). This describes a truly two-dimensional feed-forward circuit, possibly slowly growing in the second dimension if $\kappa \approx 0$. We find in the thermodynamic limit

$$\lim_{L \to \infty} P_{\mathrm{dec}}^{\checkmark} \le \begin{cases} 1 & \text{for} \quad 0 \le p_0^{\boldsymbol{x}} < \frac{1}{2} \\ 0 & \text{for} \quad p_0^{\boldsymbol{x}} = \frac{1}{2} \text{ and } \kappa < 1 \\ \frac{1}{2} & \text{for} \quad p_0^{\boldsymbol{x}} = \frac{1}{2} \text{ and } \kappa = 1 \\ 1 & \text{for} \quad p_0^{\boldsymbol{x}} = \frac{1}{2} \text{ and } \kappa > 1 \end{cases}, \tag{32}$$

i.e., except for the critical point $p_0^{\boldsymbol{x}} = \frac{1}{2}$, there is no constraint on $P_{\mathrm{dec}}^{\checkmark}$. At the critical point, the upper bounds depend on whether the second dimension scales slower or faster than the length of the chain. For faster scaling depth, there is no constraint, whereas for slower scaling depth, non-trivial upper bounds arise. Note that actually $P_{\mathrm{dec}}^{\checkmark} \le \frac{1}{2}$ follows for $p_0^{\boldsymbol{x}} = \frac{1}{2}$ for *all* decoders (not only $D$-local ones) since a completely mixing Bernoulli process destroys all encoded information. $P_{\mathrm{dec}}^{\checkmark} < \frac{1}{2}$ arises whenever the decoder fails to get rid of all syndromes (this is in contrast to the global decoder $\Delta$ which always succeeds in removing all syndromes). $P_{\mathrm{dec}}^{\checkmark} = \frac{1}{2}$ *can* be realized if the decoder succeeds in removing all syndromes but still fails to recover the original state in 50% of the cases. A detailed derivation of these results is presented in Appendix B.2.

In conclusion, decoding the Majorana chain in a single step with a constant-$D$ decoder is impossible in the thermodynamic limit. However, while $D \sim L^\kappa$ with $\kappa \geq 1$ describes only global decoders (in particular, the global majority vote $\Delta$), there is also no restriction on $P_{\text{dec}}^{\checkmark}$ for the larger class of *local* decoders with $0 < \kappa < 1$. This leaves the possibility open for local decoders with less hardware overhead than $\Delta$.

One of the main results of this manuscript is a *lower* bound on $P_{\text{dec}}^{\checkmark}$ for a class of local decoders which allows to scale $D$ at will. In particular, we find that Eq. (30) is saturated in the thermodynamic limit for $D \sim L^\kappa$ with arbitrary $\kappa > 0$ below a critical error rate $p_c^{\times} > 0$.

# 3 Cellular automata

In this section, we introduce a strictly local decoder for the MCQC. Our approach is based on cellular automata, thus we start with a description of this framework and the relevant properties. In particular, we demonstrate that for the MCQC—where only the syndromes can be measured—we have to resort to CAs that are characterized by *self-duality*, a symmetry of the local evolution rules. The natural choice is then to focus on such CAs which additionally approximate global majority voting. This task is known as *density classification* and we present two CAs that are known to perform well as density classifiers, one of which (called TLV) exhibits self-duality. Since the quantum code is embedded on a finite chain with open boundaries, it is essential to modify TLV at the edges; this new CA is denoted as $\overline{\text{TLV}}$. We argue that it acts as a self-dual density classifier on finite chains, and thereby qualifies as a promising local replacement for global majority voting on the MCQC.

## 3.1 Properties of cellular automata

To describe our local decoder, we make use of the well-known framework of one-dimensional, binary cellular automata [51, 52]; discrete dynamical systems defined on a 1D lattice $\mathscr{L}$ of binary cells $i \in \mathscr{L}$ with indices in $\mathscr{L} = \mathbb{Z}$ (infinite), $\mathbb{N}$ (semi-infinite), or $\{1, \ldots, L\}$ (finite). A *state* $\mathbf{x} \in \mathbb{Z}_2^{\mathscr{L}}$ is formally a map $\mathbf{x} : \mathscr{L} \to \mathbb{Z}_2$ assigning a state $x_i$ to each cell $i \in \mathscr{L}$. Equivalently, $\mathbf{x} \subseteq \mathscr{L}$ may be read as the subset of lattice indices $i \in \mathscr{L}$ where $x_i = 1$. A cellular automaton $\Gamma_{\mathscr{L}} : \mathbb{Z}_2^{\mathscr{L}} \to \mathbb{Z}_2^{\mathscr{L}}$ of radius $R \in \mathbb{N}$ is defined by a collection of binary functions $\gamma_i : \mathbb{Z}_2^{2R+1} \to \mathbb{Z}_2$ that determines a discrete time-evolution on $\mathbb{Z}_2^{\mathscr{L}}$ via

$$x_i' = \Gamma_i(\mathbf{x}) \equiv \gamma_i(\pi_i^R \mathbf{x}). \tag{33}$$

We write $\mathbf{x}' = \Gamma_{\mathscr{L}}(\mathbf{x})$ for short. If $\gamma_i = \gamma$ for all $i \in \mathscr{L}$, $\Gamma_{\mathscr{L}}$ is called *translationally invariant*. For $R > 0$, translational invariant CAs can be defined on infinite chains $\mathscr{L} = \mathbb{Z}$ and finite chains $\mathscr{L} = \{1, \ldots, L\}$ with periodic boundary conditions, as opposed to semi-infinite chains $\mathscr{L} = \mathbb{N}$ and finite chains with open boundaries where modifications at the boundaries are necessary (see below). We write $\mathbf{x}(t) = \Gamma_{\mathscr{L}}^t(\mathbf{x}(0))$ for the state $\mathbf{x}(t)$ that is produced by $t$ consecutive applications of $\Gamma_{\mathscr{L}}$ on the initial state $\mathbf{x}(0)$. A state $\mathbf{x}^*$ with $\mathbf{x}^* = \Gamma_{\mathscr{L}}(\mathbf{x}^*)$ is called *fixed point* of $\Gamma_{\mathscr{L}}$. More generally, a finite subset of states $C \subseteq \mathbb{Z}_2^{\mathscr{L}}$ which is invariant, $\Gamma_{\mathscr{L}}(C) = C$, and does not contain a proper invariant subset is called a *cycle* (a fixed point is a cycle with one element). On finite chains $\mathscr{L} = \{1, \ldots, L\}$, the CA always ends up in a cycle after a finite relaxation time due to the finiteness of the state space $\mathbb{Z}_2^{\mathscr{L}}$. For a given cycle $C$, the maximal set of states $A_C \subseteq \mathbb{Z}_2^{\mathscr{L}}$ with $\lim_{t \to \infty} \Gamma_{\mathscr{L}}^t(A_C) = C$ is called *attractor* of $C$. We will be interested in CAs with the homogeneous fixed points $\mathbf{x}^* = \mathbf{0}$ and $\mathbf{1}$ (characterized by the absence of syndromes) and their corresponding attractors $A_{\mathbf{0}}$ and $A_{\mathbf{1}}$.

The dynamics of a CA can be strongly influenced and restricted by symmetries of the transition rules $\Gamma_{\mathscr{L}}$. In the following, we are particularly interested in the special class of *self-dual* CAs:

**Definition 2.** *A binary cellular automaton $\Gamma_{\mathscr{L}} : \mathbb{Z}_2^{\mathscr{L}} \to \mathbb{Z}_2^{\mathscr{L}}$ described by $x_i' = \Gamma_i(\mathbf{x})$ is called self-dual if $(\Gamma_i(\mathbf{x}))^c = \Gamma_i(\mathbf{x}^c)$ for all $i \in \mathscr{L}$ with the binary complement $x_i^c \equiv x_i \oplus 1$.*

Self-duality is therefore a *symmetry* satisfied only by particular CA rules $\Gamma_{\mathscr{L}}$. For example, local rules based on majority votes of adjacent cells are automatically self-dual because the binary majority function is

$$\text{maj}\left[ x_{i_1}^c, \dots \right] = \left( \text{maj}\left[ x_{i_1}, \dots \right] \right)^c , \tag{34}$$

whereas logical dis– and conjunctions violate the symmetry, e.g.,

$$x_1^c \wedge x_2^c = (x_1 \vee x_2)^c \neq (x_1 \wedge x_2)^c . \tag{35}$$

The importance of self-dual CAs in the context of quantum error correcting the Majorana chain stems from the following observation:

**Lemma 1.** *Let $\Gamma_{\mathscr{L}}$ be a self-dual, binary cellular automaton $\Gamma_{\mathscr{L}} : \mathbb{Z}_2^{\mathscr{L}} \to \mathbb{Z}_2^{\mathscr{L}}$ acting on a one-dimensional chain $\mathscr{L}$ (infinite, semi-infinite, open or periodic boundaries); let $\mathbf{s} = \partial \mathbf{x}$ denote the syndromes.*

*Then there are two equivalent representations of $\Gamma_{\mathscr{L}}$:*

- *The **state-state** representation is given by the conventional transformation rule*

  $$\mathbf{x} \mapsto \mathbf{x}' = \Gamma_{\mathscr{L}}(\mathbf{x}) , \tag{36}$$

  *which transforms the current state $\mathbf{x}$ into the new state $\mathbf{x}'$. It operates on the states of cells $\mathbb{Z}_2^{\mathscr{L}}$ on the lattice $\mathscr{L}$.*

- *The **syndrome-delta** representation is given by the two-step process*

  $$\mathbf{s} \mapsto \boldsymbol{\Delta} = \partial \Gamma_{\mathscr{L}}(\mathbf{s}) , \tag{37a}$$
  $$(\boldsymbol{\Delta}, \mathbf{s}) \mapsto \mathbf{s}' = \partial \boldsymbol{\Delta} \oplus \mathbf{s} , \tag{37b}$$

  *and transforms the current syndrome $\mathbf{s}$ into the new syndrome $\mathbf{s}'$ via the intermediate result (delta) $\boldsymbol{\Delta}$. It operates on states of syndromes $\mathbb{Z}_2^{\partial\mathscr{L}}$ on the dual lattice $\partial\mathscr{L}$. The derived rule $\partial \Gamma_{\mathscr{L}}$ is defined for $i \in \mathscr{L}$ as*

  $$\partial \Gamma_i(\mathbf{s}) \equiv \Gamma_i\left( \left\{ x_k = \bigoplus_{\sigma \in \overline{ki}} s_\sigma \right\} \right) , \tag{38}$$

  *where for two sites $k, i \in \mathscr{L}$, $\overline{ki} = \overline{ik}$ denotes the set of sites in $\partial\mathscr{L}$ (edges in $\mathscr{L}$) between $i$ and $k$ (for periodic boundary conditions, this is well-defined due to the constraint $\oplus_{\sigma \in \partial\mathscr{L}} s_\sigma = 0$).*

The following (rather technical) proof can be skipped on first reading if the existence of the syndrome-delta representation is intuitively understood and/or accepted as a fact.

*Proof.* We show that there is a one-to-one correspondence between the two descriptions by constructing them explicitly. To this end, consider an arbitrary self-dual binary function $f : \mathbb{Z}_2^{\mathscr{L}} \to \mathbb{Z}_2$. First, note that self-duality is equivalent to the property

$$f(\{y \oplus x_i\}) = y \oplus f(\{x_i\}) \tag{39}$$

for $y \in \mathbb{Z}_2$ since $x_i^c = 1 \oplus x_i$. If we use that

$$x_i \oplus x_k = x_i \oplus (x_{i+1} \oplus x_{i+1}) \oplus \cdots \oplus (x_{k-1} \oplus x_{k-1}) \oplus x_k \tag{40a}$$
$$= (x_i \oplus x_{i+1}) \oplus (x_{i+1} \oplus \cdots \oplus x_{k-1}) \oplus (x_{k-1} \oplus x_k) \tag{40b}$$
$$= s_{i+\frac{1}{2}} \oplus \cdots \oplus s_{k-\frac{1}{2}} = \bigoplus_{\sigma \in \overline{ik}} s_\sigma \tag{40c}$$

and therefore

$$x_i = x_k \oplus \bigoplus_{\sigma \in \overline{ik}} s_\sigma \qquad (41)$$

for any $k \in \mathcal{L}$ and $\mathbf{s} = \partial \mathbf{x}$, it follows (for fixed but arbitrary $k$)

$$f(\{x_i\}) = f\left(\left\{x_k \oplus \bigoplus_{\sigma \in \overline{ik}} s_\sigma\right\}\right) = x_k \oplus f\left(\left\{\bigoplus_{\sigma \in \overline{ik}} s_\sigma\right\}\right) \qquad (42a)$$

$$\equiv x_k \oplus \tilde{f}|_k(\{s_\sigma\}) = x_k \oplus \tilde{f}|_k(\partial \mathbf{x}). \qquad (42b)$$

For a self-dual CA in state-state representation, this reads

$$x_i' = \Gamma_i(\{x_j\}) = x_k \oplus \tilde{\Gamma}_i|_k(\mathbf{s}) \qquad (43)$$

for arbitrary $k \in \mathcal{L}$. If we set $k = i$, this becomes

$$x_i' \oplus x_i = \Gamma_i(\{x_j\}) \oplus x_i = \tilde{\Gamma}_i|_i(\mathbf{s}). \qquad (44)$$

If we define the state change as $\boldsymbol{\Delta} \equiv \mathbf{x}' \oplus \mathbf{x}$ and $\partial \Gamma_i(\mathbf{s}) \equiv \tilde{\Gamma}_i|_i(\mathbf{s})$, we arrive at

$$\Delta_i = \partial \Gamma_i(\mathbf{s}) = \Gamma_i\left(\left\{x_k = \bigoplus_{\sigma \in \overline{ki}} s_\sigma\right\}\right). \qquad (45)$$

On the other hand, it is

$$s_{i+\frac{1}{2}}' = x_i' \oplus x_{i+1}' = (x_i \oplus \Delta_i) \oplus (x_{i+1} \oplus \Delta_{i+1}) \qquad (46a)$$

$$= (\Delta_i \oplus \Delta_{i+1}) \oplus (x_i \oplus x_{i+1}) = (\partial \boldsymbol{\Delta})_{i+\frac{1}{2}} \oplus s_{i+\frac{1}{2}} \qquad (46b)$$

so that

$$\mathbf{s}' = \partial \boldsymbol{\Delta} \oplus \mathbf{s} \quad \text{with} \quad \boldsymbol{\Delta} = \partial \Gamma_{\mathcal{L}}(\mathbf{s}) \qquad (47)$$

indeed describes the evolution of the syndrome $\mathbf{s} = \partial \mathbf{x}$ given by the action of $\Gamma_{\mathcal{L}}$ on the states $\mathbf{x}$. Thus we provided a procedure to derive a syndrome-delta representation from a given state-state representation. Conversely, it is enough to realize that the knowledge of $\boldsymbol{\Delta} = \partial \Gamma_{\mathcal{L}}(\partial \mathbf{x})$ allows for the computation of $\mathbf{x}'$ via $\mathbf{x}' = \mathbf{x} \oplus \boldsymbol{\Delta}$, i.e.,

$$\mathbf{x}' = \Gamma_{\mathcal{L}}(\mathbf{x}) = \mathbf{x} \oplus \partial \Gamma_{\mathcal{L}}(\partial \mathbf{x}). \qquad (48)$$

This concludes the proof. ∎

It should be clear that it is exactly the syndrome-delta representation of a self-dual CA that makes it suited for decoding the Majorana chain and comply with the "quantum-handicap": It operates on the measured syndromes $\mathbf{s}$ via the correction operations $\boldsymbol{\Delta}$ that can be applied directly to the quantum chain.

## 3.2 Density classification in 1D

We seek to apply a simple, one-dimensional binary CA as local decoder for the MCQC. If we upper bound the allowed runtime of a radius-$R$ CA $\Gamma_{\mathcal{L}}$ with $T$ time steps, the map $\Gamma_{\mathcal{L}}^T$ is $D = RT$-local per construction (since information spreads over $R$ sites per time step under CA evolution). Then the depth scaling discussed previously becomes a matter of required runtime for a specific CA.

As we know that (global) majority voting is a perfect decoder for the Majorana chain, it is natural to ask whether one can implement the function $\text{maj}[x_1, \dots, x_L]$ by a hypothetical CA $\text{MAJ}_{\mathcal{L}}$ such that

$$\lim_{t \to \infty} \text{MAJ}_{\mathcal{L}}^t(\mathbf{x}) = \mathbf{0}(\mathbf{1}) \quad \text{if} \quad \text{maj}[\mathbf{x}] = 0(1); \qquad (49)$$

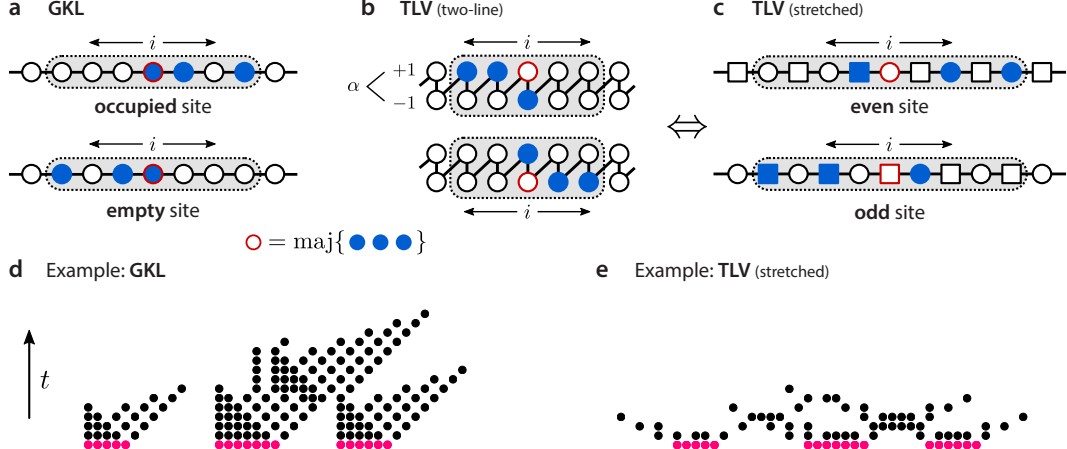

Figure 3: *Two classical density classifiers.* **a** The famous Gács-Kurdyumov-Levin (GKL) CA features $R = 3$-rules with a state-dependent choice for local majority voting. **b** The two-line voting (TLV) CA can be considered a symmetrized version of GKL that gets rid of the state-dependent majority vote by adding an additional bit per site. **c** In our case, the stretched version of TLV is more intuitive: Instead of adding new states, one adds new sites with different transition rules for even and odd positions. **d** Example evolution of a three-cluster state (red) under GKL transitions (time runs upwards). **e** Decoding of the same initial state under TLV transitions. Note the different messaging behavior of GKL (asymmetric) and TLV (symmetric).

this is known as the *density classification problem* [41, 42]. Unfortunately, it can be rigorously shown that perfect majority voting cannot be achieved with binary CAs in any dimension [43]. This, however, is not a deal breaker for majority-based error correction (both classical and quantum) as long as the erroneously classified instances are rare with respect to the noise channel in question. Motivated by applications for classical error correction, there evolved a vivid field concerned with the construction of *approximate* density classifiers (e.g., [35, 53–56]) and extensions capable of performing density classification exactly (e.g., [57–60]), see [42] for a review.

   This is how we address the problem of finding a local decoder for the MCQC: Lemma 1 allows us to filter the literature of one-dimensional binary CAs for *self-dual density classifiers*; rewritten in syndrome-delta representation, these could be directly applied as potential Majorana chain decoders.

   The first, most famous and well-studied (approximate) density classifier is dubbed "soldiers rule" and has been introduced by Gács, Kurdyumov, and Levin (GKL) [53, 61]. On $\mathscr{L} = \mathbb{Z}$, it is defined by the transition rule

$$x_i' = \mathrm{GKL}_i(\mathbf{x}) \equiv \begin{cases} \mathrm{maj}[x_i, x_{i-1}, x_{i-3}] & x_i = 0 \\ \mathrm{maj}[x_i, x_{i+1}, x_{i+3}] & x_i = 1 \end{cases} \tag{50}$$

with radius $R = 3$ [see Fig. 3 (a)]. Unfortunately, it is easy to check that it violates self-duality,

$$(\mathrm{GKL}_{\mathscr{L}}(\mathbf{x}))^c \neq \mathrm{GKL}_{\mathscr{L}}(\mathbf{x}^c) \tag{51}$$

due to the dependence of the evaluated sites in the local majority vote on the state of site $i$. This can also be seen from the exemplary time evolution of a three-cluster configuration under GKL shown in Fig. 3 (d): The emerging patterns are different for the left and right boundaries of clusters. This cannot be interpreted in terms of syndromes because on this level both boundaries are indistinguishable and hence must give rise to the same pattern.

Note that most elementary CAs (one-dimensional binary CAs with radius $R = 1$) violate self-duality as well, and the few that do not are unsuited for (approximate) density classification [62]. Most generalizations capable of exact density classification are not self-dual either [57, 58, 63] and/or reformulate the task such that a solution is no longer applicable as Majorana chain decoder [57, 63].

There are stochastic generalizations of density classifiers, some of which *are* self-dual [54, 55] and some of which are not [56]. However, we prefer *deterministic* CAs due to their simpler realization in terms of elementary logic gates. We therefore resort to the less known "two-line voting" automaton (TLV) introduced by Toom [35]. Originally, it is defined on the extended state space $(\mathbb{Z}_2 \times \mathbb{Z}_2)^{\mathscr{L}}$ describing two parallel binary chains ("two lines") and defined by the transition rule

$$\text{TLV}_{i,\alpha}(\mathbf{x}) \equiv \begin{cases} \text{maj}\left[x_i^{-1}, x_{i-1}^{+1}, x_{i-2}^{+1}\right] & \alpha = +1 \\ \text{maj}\left[x_i^{+1}, x_{i+1}^{-1}, x_{i+2}^{-1}\right] & \alpha = -1 \end{cases}, \tag{52}$$

depicted in Fig. 3 (b). In $x_i^{\alpha}$, the index $i = 1, \ldots, L$ denotes the position along the chains while $\alpha = \pm 1$ selects the subchain (up or down).

The payoff of this more complicated geometry is the sought after self-duality which is easily checked to hold,

$$(\text{TLV}_{\mathscr{L}}(\mathbf{x}))^c = \text{TLV}_{\mathscr{L}}(\mathbf{x}^c), \tag{53}$$

due to the new *in*dependence of the evaluated sites in the local majority vote on the state of site $i$ (as compared to GKL).

For our purpose, it is more convenient to rewrite TLV in its "stretched" form [Fig. 3 (c)] with state space $\mathbb{Z}_2^{\mathscr{L}}$ and state-state representation

$$x_i' = \text{TLV}_i(\mathbf{x}) \equiv \begin{cases} \text{maj}\left[x_{i-1}, x_{i+2}, x_{i+4}\right] & i \text{ even} \\ \text{maj}\left[x_{i+1}, x_{i-2}, x_{i-4}\right] & i \text{ odd} \end{cases} \tag{54}$$

for even and odd sites $i$. Fig. 3 (e) depicts the evolution of the same three-cluster configuration as in (d). In contrast to GKL, left and right boundaries spawn symmetric patterns that eventually annihilate (initially, the majority of cells was white).

Despite the rather abstract rules (54), the spatio-temporal visualization reveals the simple functional principle of TLV [see Fig. 3 (e) and also Fig. 8 (c)]: Domain walls emit "slow signals" of the form $\ldots 010101 \ldots$ *symmetrically* in both directions, seeking for nearby domain walls to pair with. When two counter-propagating slow signals meet, they transmute into "fast signals" that head back and delete the 01-markers along the way. Since the velocity of the fast signal is twice that of the other slow signal traveling into the same direction, the latter is overtaken by the returning fast signal eventually. As a result, TLV fills the gaps between the pairs of domain walls which are closest; if errors are sparse, this implies convergence to the homogeneous state $\mathbf{maj}[\mathbf{x}(0)]$.

We can now apply Lemma 1 to construct the syndrome-delta representation. Namely,

$$\Delta_i = \partial \,\text{TLV}_i(\mathbf{s}) = \text{maj}\left[s_{i \mp \frac{1}{2}}, s_{i \pm \frac{1}{2}} \oplus s_{i \pm \frac{3}{2}}, s_{i \pm \frac{1}{2}} \oplus s_{i \pm \frac{3}{2}} \oplus s_{i \pm \frac{5}{2}} \oplus s_{i \pm \frac{7}{2}}\right] \tag{55}$$

and $s'_{i+\frac{1}{2}} = s_{i+\frac{1}{2}} \oplus (\Delta_i \oplus \Delta_{i+1})$; here the upper (lower) signs correspond to $i$ even (odd).

This describes the action of TLV completely in the quantum mechanically more suitable language of syndromes $\mathbf{s}$ (obtained by measurements) and deltas $\Delta$ (applicable by local operations). Due to the equivalence of both representations, we can (and will) still use the "common" state-state representation Eq. (54) to discuss the properties of TLV. The implementation, however, requires Eq. (55) as a concession to the "quantum handicap".

Both GKL and TLV can be shown to share a property which is known to be responsible for their superior performance as approximate density classifier [36]. Clearly, the homogeneous

(syndrome free) configurations **0** and **1** are fixed points (a necessary condition for density classifiers). What distinguishes them from most other CAs with these fixed points is the structure of the attractors $A_0$ and $A_1$, i.e., the perturbed states which are drawn towards the homogeneous fixed points: Every *finite* perturbation of diameter $l$ on an infinite homogeneous background of zeros or ones is eroded after a time $t_{\text{dec}} \leq m\,l$ where $m \in \mathbb{R}^+$ is a CA-specific constant. Therefore GKL and TLV are called *linear eroders*—a crucial property for their use as approximate density classifiers (see below) and responsible for their stability close the homogeneous fixed points. The time evolutions in Fig. 3 (d) and (e) are examples for the erosion of finite perturbations of ones (red/black) on a background of zeros (white).

### 3.3 Boundary conditions

Often CAs are studied in the limit of infinite system size with state space $\mathbb{Z}_2^{\mathbb{Z}}$. However, we employ the CA for physical means which requires finite systems. Finiteness, in turn, entails a choice of boundary conditions and complicates the analysis due to finite size effects. Periodic boundary conditions (PBC) are common as they mimic the infinite case as closely as possible [see Fig. 4 (a)]: Translational invariant CAs on $\mathbb{Z}$ remain translational invariant on a finite system with PBC and no modification of the rules is necessary.

Again, due to physical constraints we cannot use periodic boundaries: It is crucial that the quantum subsystem is an *open* chain with spatially separated endpoints (edge modes). Thus we are forced to modify TLV close to the endpoints to comply with open boundary conditions. Modifying the rather complicated rules of TLV can go amiss easily. It is therefore helpful to specify our goal: Since the edges of the quantum chain carry edge modes, they can host endpoints of error strings which do not show up in the syndrome, see Eq. (5); physically, this corresponds to a quasiparticle in the delocalized edge mode. It is therefore crucial that solitary quasiparticles close to the edges are transfered into the corresponding edge mode. Thus we have to modify TLV so that syndromes are attracted by the edges (which do not emit signals themselves), while preserving self-duality and the eroder property (in a modified sense, see below).

A neat trick to come up with the correct modifications is to put the finite chain in a "cavity", between two imaginary mirrors placed left (right) of the first (last) site, see Fig. 4 (b). Rules which traverse the edges use the mirrored cells to compute their local update. Formally this is achieved by redefining these rules to use the corresponding "real" cells of the system (note that this is a *local* modification for a stretched open chain, in contrast to periodic boundaries); we call this *mirrored boundary conditions* (MBC). If we denote the finite size version of TLV on $\mathcal{L} = \{1, \dots, L\}$ with mirrored boundary conditions as $\overline{\text{TLV}}$, the modified rules on the left edge read

$$\overline{\text{TLV}}_1(\mathbf{x}) = \text{maj}\left[x_2, x_2, x_4\right], \tag{56a}$$

$$\overline{\text{TLV}}_3(\mathbf{x}) = \text{maj}\left[x_4, x_1, x_2\right], \tag{56b}$$

and on the right edge ($L$ even)

$$\overline{\text{TLV}}_{\overline{1}}(\mathbf{x}) = \text{maj}\left[x_{\overline{2}}, x_{\overline{2}}, x_{\overline{4}}\right], \tag{57a}$$

$$\overline{\text{TLV}}_{\overline{3}}(\mathbf{x}) = \text{maj}\left[x_{\overline{4}}, x_{\overline{1}}, x_{\overline{2}}\right], \tag{57b}$$

where we used the shorthand notation $\overline{k} \equiv L + 1 - k$ to index cells from the end of the chain (e.g., $\overline{1} = L$), see Fig. 4 (b). For all other sites it is $\overline{\text{TLV}}_i = \text{TLV}_i$.

Clearly, **0** and **1** are still fixed points of $\overline{\text{TLV}}$ (there are no static signal sources introduced) and self-duality is also preserved. By construction, a slow signal emitted from a solitary syndrome close to the edge will meet its mirror image at the edge which sends it back as a fast

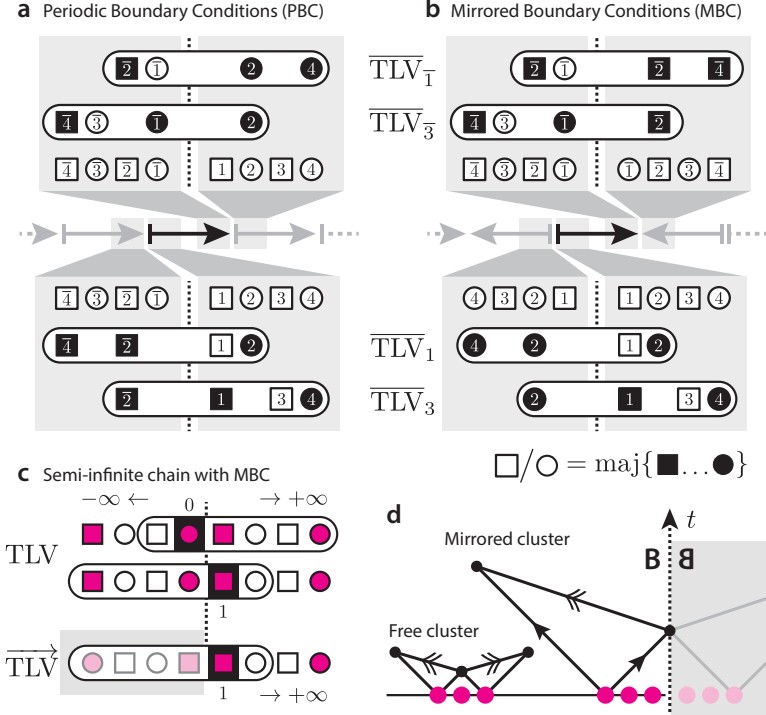

Figure 4: *Boundary conditions.* **a** Periodic boundary conditions (PBC). A finite system of length $L$ (black arrow) is copied and chained without inverting the direction. Four local rules traverse the boundary and are modified accordingly. **b** Mirrored boundary conditions (MBC). Here every other copy is reversed, giving rise to a finite system bounded by two mirrors with modified rules $\overline{\mathrm{TLV}}_{\mathscr{L}}$. **c** An unmodified TLV operating on an infinite chain and restricted to bond-inversion symmetric states (magenta/white sites for $x_i = 1/0$) is equivalent to a modified $\overrightarrow{\mathrm{TLV}}$ operating on a semi-infinite chain with MBC on the edge. This is a consequence of the bond-inversion symmetry of the unmodified TLV rules. **d** A finite cluster of errors can be effectively doubled in size if it is close to the mirror. Consequently, correction times close to the mirror can be longer than for a free cluster of the same size.

signal to capture the other slow signal heading into the bulk and thereby initiates pairing towards the edge, see Fig. 4 (d). Note that this mechanism affects the time needed to erode a contiguous cluster of errors: Adjacent (or close) to the mirror, the number of errors is "doubled" artificially; correction time and affected territory double accordingly. In Fig. 4 (d) we illustrate this effect by comparing the same cluster far away and close to the edge.

An important observation allows for the analysis of systems with mirrored boundary conditions in terms of the unmodified rules (on the infinite chain $\mathscr{L} = \mathbb{Z}$): Let $\mathbf{x} \in \mathbb{Z}_2^{\mathbb{Z}}$ be an arbitrary state and define the bond-centered inversion $\mathfrak{I}_s$ as

$$(\mathfrak{I}_s \mathbf{x})_i \equiv x_{2s-i+1}. \tag{58}$$

$\mathfrak{I}_s \mathbf{x}$ describes the configuration that is obtained by inversion of $\mathbf{x}$ at the bond $(s, s+1)$. We define the set of invariant configurations,

$$\mathcal{K}_s \equiv \left\{ \mathbf{x} \in \mathbb{Z}_2^{\mathbb{Z}} \mid \mathfrak{I}_s \mathbf{x} = \mathbf{x} \right\} \tag{59}$$

and argue that $\mathfrak{I}_s$ is a symmetry of TLV for any $s \in \mathbb{Z}$, namely

$$\mathrm{TLV}_{\mathscr{L}}(\mathfrak{I}_s \mathbf{x}) = \mathfrak{I}_s \, \mathrm{TLV}_{\mathscr{L}}(\mathbf{x}). \tag{60}$$

Indeed, this follows from the fact that $TLV_i$ is related to $TLV_{i+1}$ by a bond-centered inversion at $(i, i+1)$; this is true for both even and odd sites $i$, see Fig. 3 (c). It follows that $\mathscr{K}_s \subset \mathbb{Z}_2^{\mathbb{Z}}$ is invariant under the evolution of TLV which hence can be restricted to $\mathscr{K}_s$. Note that this is a special feature of TLV, in contrast to GKL, for instance. Without loss of generality, we set $s = 0$ in the following, i.e., $(\mathfrak{I}_0 \mathbf{x})_i = x_{1-i}$. Then we can describe a semi-infinite chain on $\mathscr{L} = \mathbb{N}$ with a single mirrored boundary (we write $\overrightarrow{TLV}$) by the unmodified rules of TLV operating on the infinite chain $\mathscr{L} = \mathbb{Z}$ if we restrict the state space to $\mathscr{K}_0$. Indeed,

$$\overrightarrow{TLV}_1(\mathbf{x}) = \text{maj}\left[x_2, x_2, x_4\right] = \text{maj}\left[x_2, x_{-1}, x_{-3}\right] = TLV_1(\mathbf{x}), \tag{61a}$$

$$\overrightarrow{TLV}_3(\mathbf{x}) = \text{maj}\left[x_4, x_1, x_2\right] = \text{maj}\left[x_4, x_1, x_{-1}\right] = TLV_3(\mathbf{x}), \tag{61b}$$

where we used $x_4 = x_{-3}$ and $x_2 = x_{-1}$ for $\mathbf{x} \in \mathscr{K}_0$; see Fig. 4 (c) for an example. This allows us to trade the rule modifications of $\overrightarrow{TLV}$ for a restriction on the state space of TLV which, in turn, simplifies the analysis of the *finite* version $\overline{TLV}$ (see below). As an immediate consequence, it follows that the semi-infinite $\overrightarrow{TLV}$ is an eroder because TLV is one (and mirrored finite perturbations remain finite).

While the previously introduced definition of eroders carries over to semi-infinite chains, it cannot be applied to *finite* systems because there is no longer a qualitative difference between perturbation and background (both of which are necessarily finite). A possible finite-size modification reads as follows: A cellular automaton on a finite chain $\mathscr{L} = \{1, \dots, L\}$ is a finite-size linear eroder if there exist real constants $0 < a < 1$ and $m \in \mathbb{R}^+$ such that for any size $L < \infty$ and any finite perturbation of $\mathbf{0}$ ($\mathbf{1}$) with diameter $l \leq a L$, the unperturbed state $\mathbf{0}$ ($\mathbf{1}$) is recovered at $t_{\text{dec}} \leq m l$. It is easy to check that $\overline{TLV}$ is an eroder in this sense if one uses that $\overrightarrow{TLV}$ is an eroder in the original sense (see Appendix C). Alternatively, note that the majority function is *monotonic*, i.e., changing an input bit from 0 to 1 cannot change the output bit from 1 to 0. Therefore the evolution of a generic, non-contiguous, finite cluster of errors under $TLV/\overrightarrow{TLV}/\overline{TLV}$ can be constructed from the evolution of a contiguous cluster of the same size by erasing errors in the spacetime diagram, and it is sufficient to consider contiguous intervals of errors to check for the eroder property (which is straightforward to verify).

# 4 Decoding with a self-dual density classifier

In the previous section, we introduced $\overline{TLV}$ and argued that it is a self-dual, finite-size linear eroder. These properties make $\overline{TLV}$ a promising candidate for decoding errors $E(\mathbf{x})$ that are small compared to $L$ and/or *sparse* enough. In the following, we assess the decoding performance of $\overline{TLV}$ by numerical and analytical means. We show that error patterns for which the erosion (viz. decoding) fails are rare for reasonably small error rates, and that the time $t_{\text{dec}}$ required for decoding scales sublinearly with the chain length—one of the main benefits of locality.

## 4.1 Numerical results

We start with a qualitative discussion of possible evolutions under $\overline{TLV}$: Apart from the two stable homogeneous fixed points $\mathbf{0}$ and $\mathbf{1}$, there are four additional (unstable) fixed points for TLV [35] two of which cannot be realized by $\overline{TLV}$ on finite chains with MBC (see Appendix D). This leads to the four possible fixed points depicted in Fig. 5 (a) and (b). Note that their realization on finite chains with MBC (vertical lines) is only possible if their realization on $\mathscr{L} = \mathbb{Z}$ is consistent with the boundary conditions given by the mirrors. Whereas the homogeneous

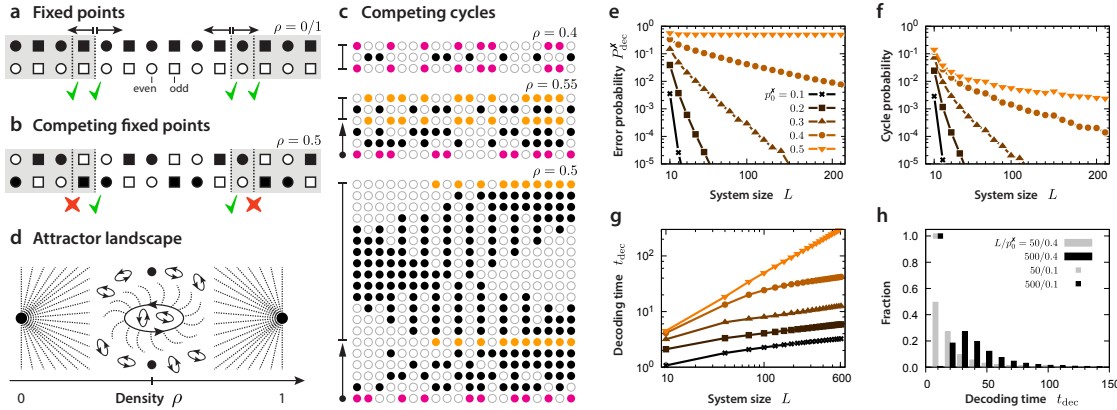

Figure 5: *Properties of* $\overline{\mathrm{TLV}}$. **a** The homogeneous fixed points of $\overline{\mathrm{TLV}}$ with minimum/maximum filling $\rho = 0/1$ correspond to the syndrome-free states of the quantum code. The eroder property makes them stable in that (small enough) perturbations are erased and do not proliferate. **b** Two of four additional fixed points of TLV are inherited by $\overline{\mathrm{TLV}}$ if the leftmost/rightmost site are labeled by even/odd indices (check marks). Those are characterized by a critical filling $\rho = 0.5$. **c** Three examples of random initial configurations (magenta) which relaxed into cycles of various lengths. The first recurring configurations are highlighted with the same color to separate the cycle from the relaxation path. We find only cycles close to criticality with $\rho \approx 0.5$. **d** Sketch of the state space with dashed attractor paths, based on the results in **a-c**. The two homogeneous fixed points are attractors of all states away from criticality; this motivates the application of $\overline{\mathrm{TLV}}$ as decoder. **e** Numerical results for the probability of erroneous decoding $P_{\mathrm{dec}}^{\textbf{X}} = 1 - P_{\mathrm{dec}}^{\checkmark}$ vs. chain length $L$ for different microscopic error probabilities $p_0^{\textbf{X}}$. $P_{\mathrm{dec}}^{\textbf{X}}$ vanishes exponentially in the thermodynamic limit for any $p_0^{\textbf{X}} < 0.5$. **f** Away from criticality (presumably for $p_0^{\textbf{X}} < 0.5$) the probability to relax into a cycle vanishes exponentially. For realistic error rates ($p_0^{\textbf{X}} \leq 0.1$), cycles cannot be observed in reasonable sample sizes. **g** Averaged time needed to reach a homogeneous fixed point ($t_{\mathrm{dec}}$) as a function of the system size $L$ for various error rates $p_0^{\textbf{X}}$. The growth is remarkably slow but unbounded for $p_0^{\textbf{X}} > 0$. Whether $t_{\mathrm{dec}}$ grows algebraically or only logarithmically for small $p_0^{\textbf{X}} > 0$ cannot be inferred from these results. **h** Distributions of the decoding times $t_{\mathrm{dec}}$ for two error rates $p_0^{\textbf{X}} = 0.1/0.4$ and system sizes $L = 50/500$. For $p_0^{\textbf{X}} = 0.1$, there is barely any difference between $L = 50$ and $L = 500$ visible (squares). We sampled $10^6$ random initial states for each data point in **e-h**.

fixed points survive, independent of the bond where the mirror is placed (a), the two additional fixed points can only be realized if the leftmost (rightmost) site is denoted by an even (odd) index (b). Henceforth, we will take the first index to be odd (i.e., $i = 1$) and the last to be even (i.e., $L$), which eliminates these two additional fix points. Note that this choice coincides with the default labeling of sites $\mathscr{L} = \{1, \ldots, L\}$. We stress that the elimination of the two additional fixed points (which are not syndrome free) is not crucial for the performance of the decoder: First, both are characterized by a density of set bits $\rho = 0.5$ which is far from the relevant error densities realistic for small $p_0^{\textbf{X}}$. Second, simulations suggest that their attractors are trivial, i.e., contain only the fixed points themselves (Appendix D).

However, there are competing cycles of various lengths and with non-trivial attractors, three examples are shown in Fig. 5 (c). The longer a cycle and the larger its attractor, the more probable it is with respect to Bernoulli noise. This explains why the largest cycle in (c) is by far the most common in simulations. Note that it also illustrates the MBC nicely by "bouncing" a cluster of errors hence and forth between the two mirrors. Again, these cycles are characterized

by densities $\rho$ close to criticality, which renders them rare for $p_0^{\chi} \ll \frac{1}{2}$. In Fig. 5 (d) we sketch the attractor landscape of the total state space ordered by the density $\rho$: Close to the extreme densities $\rho = 0/1$ every configuration is drawn towards the corresponding homogeneous fixed point due to the eroder property. This is where $\overline{\text{TLV}}$ implements effectively majority voting by local rules and therefore becomes a viable replacement for the global decoder $\Delta$. Only close to criticality $\rho \approx 0.5$, $\overline{\text{TLV}}$ fails to decode a (still small) fraction of error patterns by evolving them into cycles instead of cleaning them according to a global majority vote. This underpins our previous statement that the impossibility of realizing global majority voting perfectly is not too much of an issue if it fails in regions of the state space which are exponentially suppressed by Bernoulli noise for physically realistic error rates.

In the remainder of this subsection, we will quantify these statements by sampling error patterns from a Bernoulli distribution with fixed rate $p_0^{\chi}$ and evolving them with $\overline{\text{TLV}}$ until we can decide whether it reached a fixed point or entered a cycle. We interpret the empty state $\mathbf{0}$ as error free and define the probability of successful decoding as

$$P_{\text{dec}}^{\checkmark} \equiv \Pr\left(\left\{\mathbf{x} \in \mathbb{Z}_2^{\mathscr{L}} \mid \lim_{t\to\infty} \overline{\text{TLV}}_{\mathscr{L}}^{t}(\mathbf{x}) = \mathbf{0}\right\}\right). \tag{62}$$

We stress that in addition to $\lim_{t\to\infty} \overline{\text{TLV}}_{\mathscr{L}}^{t}(\mathbf{x}) = \mathbf{1}$, and in contrast to the global decoder $\Delta$, $\overline{\text{TLV}}$ can also fail by evolving into cycles which are not syndrome free. *Both* cases make up for the failed decodings of $\overline{\text{TLV}}$ and are measured by the probability $P_{\text{dec}}^{\chi} = 1 - P_{\text{dec}}^{\checkmark}$. As a consequence, $P_{\text{dec}}^{\chi} > \frac{1}{2}$ is possible for $\overline{\text{TLV}}$ even for $p_0^{\chi} \leq \frac{1}{2}$.

In Fig. 5 (e) we plot estimates for $P_{\text{dec}}^{\chi}$ as function of the system size for various error rates $0 < p_0^{\chi} \leq 0.5$. Except for the critical value $p_0^{\chi} = 0.5$, the probability of unsuccessful decoding vanishes exponentially with the chain length $L$, confirming our hope that $\overline{\text{TLV}}$ is a viable replacement for $\Delta$. Note that this result already tells us that the measure of all attractors of cycles vanishes quickly for $L \to \infty$. Indeed, in Fig. 5 (f) we plot the probability of an error pattern to belong to the attractor of a non-trivial cycle, again as function of $L$ for the same error rates as in (e): For $p_0^{\chi} < 0.5$ and $L \gtrsim 50$, there seems to be an exponential decay which is in accordance with the results in (e). Whether at criticality $p_0^{\chi} = 0.5$ the probability vanishes or saturates at a small but non-zero value cannot be inferred from (f). Interestingly, the results so far not only support the hope that $\overline{\text{TLV}}$ can replace $\Delta$ for $p_0^{\chi} \leq p_c^{\chi}$ with a non-trivial critical rate $0 < p_c^{\chi} < \frac{1}{2}$, but even suggest that $p_c^{\chi} = \frac{1}{2}$ is still optimal (at least $0.4 \lesssim p_c^{\chi}$).

Now that we know that the decoding probability of $\overline{\text{TLV}}$ approaches 1 exponentially with $L \to \infty$ comes a crucial question we shunned so far: How many steps $t_{\text{dec}}$ does $\overline{\text{TLV}}$ need, on average, to evolve an error pattern $\mathbf{x}$ into the error-free state $\mathbf{0}$? If the decoding time scaled linearly, $t_{\text{dec}} \sim L$, there would be barely any benefit from replacing the global decoder $\Delta$ by the local one. Fortunately, Fig. 5 (g) reveals that the average decoding time grows linearly *only* at criticality whereas the growth for $p_0^{\chi} < 0.5$ is much slower. E.g., for $L = 600$ and $p_0^{\chi} = 0.1$ on average only $t_{\text{dec}} \approx 3$ steps are necessary to eliminate all errors correctly. We stress that due to the almost vanishing slope in (g) it is not possible to decide whether $t_{\text{dec}} \propto L^{\kappa}$ for $0 < \kappa \ll 1$ or $t_{\text{dec}} \sim \log L$, even though the very fact that $t_{\text{dec}}$ grows so slowly hints at a logarithmic scaling. To describe the required decoding times in detail, we show the complete probability distribution in Fig. 5 (h) for two sizes $L = 50/500$ and error rates $p_0^{\chi} = 0.1/0.4$ in bins of $\Delta t_{\text{dec}} = 10$. Most strikingly, for the lower error rate $p_0^{\chi} = 0.1$ there is no difference between $L = 50$ and a chain of the tenfold length; again a manifestation of the extremely slow growth of $t_{\text{dec}}$ for reasonable error rates.

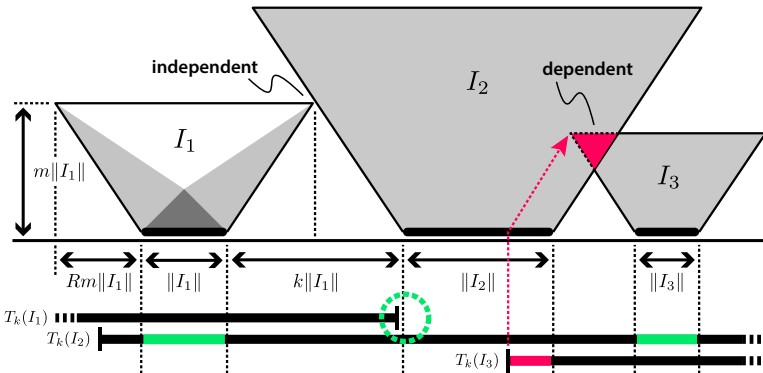

Figure 6: *Independent clusters.* A pattern of three clusters $I_i$ ($i = 1, 2, 3$) with $\|I_3\| < \|I_1\| < \|I_2\|$. The erosion process of TLV is sketched for $I_1$ whereas for $I_2$ and $I_3$ only the causal patches that cover the erosion are shaded gray (time runs upwards). For a linear eroder, the erasure of an independent cluster requires at most $m\|I_i\|$ time steps, the height of the shown trapezoids. During this time, signals can travel at most $Rm\|I_i\|$ sites from the boundary of a cluster $I_i$. Clusters that are *independent* with sparseness parameter $k = 2Rm$ do not interact; this is true for $I_1$ and $I_2$ since $I_2 \cap T_k(I_1) = \emptyset$ (dashed circle). Note that $I_{1,3} \cap T_k(I_2) \neq \emptyset$ (green intervals) has no effect on their (in-)dependence because $\|I_2\| \geq \|I_1\|, \|I_3\|$. In contrast, since $I_2 \cap T_k(I_3) \neq \emptyset$ (red interval), the causal trapezoids of $I_2$ and $I_3$ intersect (red triangle). Thus, $I_2$ and $I_3$ are *dependent* and may not be erased separately.

## 4.2 Rigorous analytical results

In this subsection, we prove a central statement of this work: The probability for $\overline{\text{TLV}}$ on a chain of length $L$ with MBC to be in a non-empty state $\mathbf{x}(t) \neq \mathbf{0}$ after $t \propto L^\kappa$ time steps vanishes exponentially with $L$ for arbitrary $\kappa > 0$ if the initial state $\mathbf{x}(0)$ is a Bernoulli random configuration with single-site error probability $p_0^{\boldsymbol{\chi}} < p_c^{\boldsymbol{\chi}}$ for some critical value $0 < p_c^{\boldsymbol{\chi}} \leq \frac{1}{2}$. In Section 5 we will use this result to construct a completely local decoder for the Majorana chain with length $L$ and depth $\propto L^\kappa$ that stabilizes a logical qubit for times that grow exponentially with $L$. Furthermore, it confirms the numerical results of Subsection 4.1: Neither competing fixed points nor cycles threaten the performance of $\overline{\text{TLV}}$ as long as $p_0^{\boldsymbol{\chi}}$ is small enough. To prove the claimed result, we follow the lines of [64] with modifications to account for the finiteness of $\overline{\text{TLV}}$ and the mirrored boundaries. In the following, we present three crucial steps but provide only brief sketches of their proofs; the details are presented in Appendix E.

Before we can state our first result, we have to introduce the pivotal concepts of *independence* and *sparseness* [26, 45, 46, 64, 65]. Let $\mathbf{x} \subseteq \mathbb{Z}$ be an arbitrary subset (error pattern). A finite subset $I \subseteq \mathbf{x}$ is called *cluster* of diameter $\|I\| = \max\{|x - y| \,|\, x, y \in I\}$. If we fix an integer $k > 0$ (the *sparseness parameter*, to be chosen later), the *territory* $T_k(I)$ is defined as the interval of integers with distance at most $k\|I\|$ from $I$. Two clusters $I_1$ and $I_2$ are called *independent* if at least one does not intersect the territory of the other, i.e., $I_2 \cap T_k(I_1) = \emptyset$ or $I_1 \cap T_k(I_2) = \emptyset$ (or both); since $I \subset T_k(I)$, this implies $I_1 \cap I_2 = \emptyset$. This concept is illustrated in the lower part of Fig. 6. If, in addition, there exists a partition of $\mathbf{x}$ into a family $\mathscr{I} = \{I_a\}$ of pairwise independent clusters $I_a$, $\mathbf{x} = \bigcup_a I_a$, then $\mathbf{x}$ is called *sparse*.

We need some additional terminology: First, $\mathscr{I}_{\leq l}$ denotes the family of clusters $I \in \mathscr{I}$ with diameter $\|I\| \leq l$ and $\mathbf{x} \setminus \mathscr{I}_{\leq l} \equiv \mathbf{x} \setminus \bigcup_{I \in \mathscr{I}_{\leq l}} I$ is the subset of sites for given $\mathbf{x}$ that remains after cleaning all independent clusters of diameter at most $l$. Second, a *(infinite) mirrored* Bernoulli random configuration $\mathbf{x} \subseteq \mathbb{Z}$ is defined by the single-site probability $\Pr(x_i = 1) = p_0^{\boldsymbol{\chi}}$ for sites $i > 0$ and the mirror constraint $x_i = x_{1-i}$ [recall Eq. (59)].

We can now state our variation of the main result of Ref. [64]:

**Proposition 1.** *Consider **infinite mirrored** Bernoulli random configurations* $\mathbf{x}$ *with single-site probability* $p_0^{\cancel{x}}$*. Let* $k \in \mathbb{N}$ *be a given sparseness parameter.*

*Then, for each instance* $\mathbf{x}$*, there exists a constructive family* $\mathscr{I}^{\mathbf{x}}$ *of pairwise independent clusters (it is not necessarily* $\mathbf{x} = \bigcup_{I \in \mathscr{I}^{\mathbf{x}}} I$*, i.e.,* $\mathbf{x}$ *does not have to be sparse) such that the probability of a site* $i \in \mathbb{Z}$ *to be in* $\mathbf{x}$ *and remain uncovered by independent clusters of diameter* $l$ *or less (write* $\mathscr{I}^{\mathbf{x}}_{\leq l}$*) is upper-bounded by*

$$\Pr\left(i \in \mathbf{x} \setminus \mathscr{I}^{\mathbf{x}}_{\leq l}\right) \leq \alpha^{l^{\beta}} \tag{63}$$

*for* $\beta = \ln(2)/\ln(4k+3)$ *(and therefore* $0 < \beta < 1$*) and* $\alpha = (2k)(4k+3)\sqrt{p_0^{\cancel{x}}}$*. If we define the critical value*

$$\tilde{p}_c^{\cancel{x}} \equiv \left[(2k)(4k+3)\right]^{-2}, \tag{64}$$

*for* $p_0^{\cancel{x}} < \tilde{p}_c^{\cancel{x}}$ *it is* $\alpha < 1$ *and Eq. (63) becomes an exponentially decaying upper bound.*

The proof can be divided roughly into three steps: First, the family $\mathscr{I}^{\mathbf{x}}$ is constructed recursively in the cluster diameter $l$ for a given instance $\mathbf{x}$. In a second step it is shown that this prescription always yields a family of pairwise independent clusters. In the crucial third step, an upper bound on the probability for a site $i \in \mathbf{x}$ to be *not* covered by a cluster in $\mathscr{I}^{\mathbf{x}}$ of diameter $l$ or less is derived. To do this, one constructs so called *explanation trees*, hypothetical error patterns that explain why a given site $i$ could survive the construction of $\mathscr{I}^{\mathbf{x}}$ without being covered by clusters up to diameter $l$. The probability for its survival is then estimated by finding an upper bound on the number of possible explanation trees and calculating their probability with respect to a mirrored Bernoulli distribution. One finds that the number of explanation trees *grows* exponentially (with cluster diameter $l$) while the probability for a single explanation tree to be realized by a Bernoulli process *vanishes* exponentially. The latter factor dominates for $p_0^{\cancel{x}} < \tilde{p}_c^{\cancel{x}}$ so that the probability for the existence of at least one explanation tree vanishes exponentially for increasing $l$; this leads to Eq. (63).

The rationale behind TLV (or any other linear eroder) is the following: For an error to survive the erosion process, there must be other errors nearby that protect it; and these, in turn, require further errors in *their* neighborhood to survive and so forth. Such a structure of errors that protect each other from being eroded constitutes an explanation tree which prevents a global error pattern from decaying into independent clusters. Explanation trees are *dense* in a very specific sense—and this denseness renders their existence improbable for low error rates. In contrast, *sparse* error patterns are those without explanation trees that span the whole system. They are initial states of linear eroders such that the causal regions of correlated sites in the spacetime diagram do not percolate through the system, but instead separate into many local patches which are eroded independently. The initial seeds of these patches are the independent clusters from above: Linear eroders clean a single cluster $I$ after at most $m\|I\|$ time steps, and can therefore influence only sites with maximum distance $Rm\|I\|$ from $I$. Then, a collection of pairwise independent clusters is eroded *independently* if the sparseness parameter is set to $k = 2Rm$, see Fig. 6. It is this causal locality on sparse sets which results in the sublinear scaling of decoding times for TLV [recall Fig. 5 (g)].

Eventually we want to use Prop. 1 to derive an upper bound for the probability of errors to survive the first $t$ steps of $\overline{\text{TLV}}$ on a *finite* chain with mirrored boundaries. To this end, we first need an intermediate step:

**Lemma 2.** *Consider a **semi-infinite** chain on $\mathscr{L} = \mathbb{N}$ governed by $\overrightarrow{\text{TLV}}_{\mathscr{L}}$ with initial configurations $\mathbf{x}(0) \subseteq \mathscr{L}$ drawn from a Bernoulli distribution with parameter $p_0^{\chi}$. Let $\mathscr{J} \subset \mathscr{L}$ be an arbitrary finite interval on the chain.*

*Then the probability of $\mathbf{x}(t) = \overrightarrow{\text{TLV}}_{\mathscr{L}}^t(\mathbf{x}(0))$ to be non-empty on $\mathscr{J}$ is upper-bounded by*

$$\Pr(\mathbf{x}(t) \cap \mathscr{J} \neq \emptyset) \leq (2tR + |\mathscr{J}|) \exp\left(-\gamma \lfloor t/m \rfloor^{\beta}\right), \tag{65}$$

*with $\gamma = -\log(\alpha)$ ($\gamma > 0$ for $p_0^{\chi} < \tilde{p}_c^{\chi}$), and $0 < \beta < 1$ as in Prop. 1. Here the sparseness parameter is given by $k = 2Rm = 8$ where $m = 1$ and $R = 4$ are the eroder parameter and the radius of TLV, respectively. $\lfloor x \rfloor$ denotes the greatest integer less than or equal to $x$.*

The proof exploits that $\overrightarrow{\text{TLV}}$ is equivalent to TLV for symmetric states in $\mathscr{K}_0$. Then Prop. 1 provides us with a family $\mathscr{I}^{\mathbf{x}}$ that fails to cover errors in $\mathbf{x}$ with a probability that vanishes exponentially with increasing cluster diameter $l$. If an error $x_i = 1$ belongs to an independent cluster of diameter $l$, the linear eroder property of TLV ensures that it is eroded after at most $ml$ time steps. It is important to realize that this does *not* imply $x_i = 0$ for all later times as signals from distant, larger clusters may enter the territory of smaller ones (e.g., $I_1$ and $I_2$ in Fig. 6). With $R$ the radius of local rules, the neighborhood $U_{tR}(\mathscr{J})$ includes all sites that potentially influence sites in $\mathscr{J}$ after $t$ time steps, i.e., sites with distance at most $tR$ from $\mathscr{J}$. Therefore one has to demand that *all* sites in the growing neighborhood $U_{tR}(\mathscr{J})$ belong to clusters of maximum diameter $l$ in $\mathscr{I}^{\mathbf{x}}$ to guarantee that $\mathscr{J}$ is clean after $t = ml$ time steps. Subadditivity of probability measures then leads to the upper bound of Lemma 2 where $(2tR + |\mathscr{J}|)$ is the size of $U_{tR}(\mathscr{J})$.

With Lemma 2, we are now prepared to tackle the case of *finite* chains:

**Lemma 3.** *Consider a **finite** chain of length $L$ on $\mathscr{L} = \{1, \ldots, L\}$ governed by $\overline{\text{TLV}}$ with mirrored boundaries and initial configurations $\mathbf{x}(0) \subseteq \mathscr{L}$ drawn from a Bernoulli distribution with parameter $p_0^{\chi}$.*

*Then the probability of $\mathbf{x}(t) = \overline{\text{TLV}}_{\mathscr{L}}^t(\mathbf{x}(0))$ to be non-empty is upper-bounded by*

$$\Pr(\mathbf{x}(t) \neq \emptyset) \leq (4R\{t\} + L) \exp\left(-\gamma \lfloor \{t\}/m \rfloor^{\beta}\right), \tag{66}$$

*with $\{t\} \equiv \min\{t, t_L^*\}$ and $t_L^* = \lfloor L/2R \rfloor$. The parameters are the same as in Prop. 1 and Lemma 2.*

Whereas the *infinite* TLV and the *semi-infinite* $\overrightarrow{\text{TLV}}$ are qualitatively similar due to the discussed equivalence on $\mathscr{K}_0$, there are fundamental differences to the *finite* $\overline{\text{TLV}}$. This can be understood intuitively as follows: $\overrightarrow{\text{TLV}}$ equals TLV with pairwise correlations between mirrored sites. These pairwise correlations lead to the square in the expression for $\tilde{p}_c^{\chi}$ (recall Prop. 1). In contrast, $\overline{\text{TLV}}$ introduces an *infinite* number of perfectly correlated partners for each of the $L$ sites due to the cavity geometry (imagine standing in front of a single mirror vs. standing between two opposing mirrors). To avoid these complications, we use a trick: For times $t \leq t_L^* = \lfloor L/2R \rfloor$, there is no site with *both* boundaries (mirrors) in its past light cone (the site(s) closest to the center of the chain can get "aware" of the cavity geometry earliest at $t_L^* + 1$). Therefore locally the *finite* system $\overline{\text{TLV}}$ behaves exactly as the semi-infinite system $\overrightarrow{\text{TLV}}$ for $t \leq t_L^*$ and the results of Lemma 2 apply. For $t > t_L^*$ we can exploit the finiteness of $\mathscr{L}$: Recall that in the context of Lemma 2 we stressed that an empty interval does not necessarily remain empty on a (semi-)infinite chain because signals from outside the interval may interfere at later times. Now $\mathscr{L}$ is finite and the argument no longer holds: if $\mathbf{x}(t) = \emptyset$ at some time $t$, it follows $\mathbf{x}(t') = \emptyset$ for all later times $t' > t$. Thus the probability of $\mathbf{x}(t) \neq \emptyset$ is monotonically decreasing in $t$. This leads to the replacement $t \to \{t\} = \min\{t, t_L^*\}$ in Lemma 3.

Note that the lower-bounded decay of the probability in Lemma 3 is to be expected for finite systems: Due to the finite state space, there is an upper bound for $t$ (depending on $L$) such that the system either (1) relaxed to the clean state, (2) to a non-clean fixed point, or (3) entered a non-trivial cycle. In the first case, it is clean forever, whereas in the latter two cases, it can never become clean. Therefore the probability to be not clean cannot decrease arbitrarily and must be bounded from below for fixed $L$ and $t \to \infty$. However, if we are interested in the limit $L \to \infty$, we can ask how long one has to wait for $\overline{\text{TLV}}$ to clean the system almost surely.

This leads to our main result:

**Corollary 1.** *Consider a **finite** chain of length $L$ on $\mathscr{L} = \{1, \ldots, L\}$ governed by $\overline{\text{TLV}}$ with mirrored boundaries and initial configurations $\mathbf{x}(0) \subseteq \mathscr{L}$ drawn from a Bernoulli distribution with parameter $p_0^{\boldsymbol{\times}}$.*

*For $\kappa \in \mathbb{R}$ with $0 < \kappa < 1$, the probability of $\mathbf{x}(t) = \overline{\text{TLV}}_{\mathscr{L}}^t(\mathbf{x}(0))$ to be non-empty after*

$$t_{\max}(L) \equiv \lfloor L^\kappa \rfloor \tag{67}$$

*time steps is upper-bounded by*

$$\Pr(\mathbf{x}(t_{\max}) \neq \emptyset) \leq (4R+1) L \exp\left(-\gamma \lfloor L^\kappa/m \rfloor^\beta\right) \tag{68}$$

*for $L \geq L_R$ with $0 < L_R < \infty$ a R-dependent constant. For $p_0^{\boldsymbol{\times}} < \tilde{p}_c^{\boldsymbol{\times}}$ it follows that*

$$\Pr(\mathbf{x}(t_{\max}(L)) \neq \emptyset) \to 0 \quad \text{for} \quad L \to \infty \tag{69}$$

*exponentially fast. The parameters are the same as in Prop. 1 and Lemma 2.*

To prove this, we use the result of Lemma 3 with $t_{\max}(L) < t_L^*$ for $L \geq L_R$ large enough, thus $\{t_{\max}(L)\} = \min\{t_{\max}(L), t_L^*\} = t_{\max}(L)$. With $\lfloor L^\kappa \rfloor \leq L$ and $\lfloor \lfloor L^\kappa \rfloor/m \rfloor = \lfloor L^\kappa/m \rfloor$ (for $m \in \mathbb{N}$), Eq. (68) follows immediately.

The important message of Corollary 1 is that the probability

$$\overline{P}_{\text{dec}}^{\checkmark} \equiv \Pr\left(\left\{\mathbf{x} \in \mathbb{Z}_2^{\mathscr{L}} \mid \overline{\text{TLV}}_{\mathscr{L}}^{t_{\max}(L)}(\mathbf{x}) = \mathbf{0}\right\}\right) \tag{70}$$

of successfully decoding a Bernoulli random configuration *with time constraint* $t_{\max}(L)$ [cf. Eq. (62)] approaches 1 rapidly for longer chains even if the allocated decoding time $t_{\max}(L)$ increases sublinearly as long as $p_0^{\boldsymbol{\times}} < \tilde{p}_c^{\boldsymbol{\times}}$. We stress that there is no statement about $p_0^{\boldsymbol{\times}} \geq \tilde{p}_c^{\boldsymbol{\times}}$; Corollary 1 only asserts that there is a finite range for $p_0^{\boldsymbol{\times}}$ where decoding with $\overline{\text{TLV}}$ is possible and that the required decoding time $t_{\text{dec}}$ scales favorably with $L$ on average.

To conclude this subsection, we present numerical results for $\overline{P}_{\text{dec}}^{\checkmark}$ as a function of the microscopic error probability $0.3 \leq p_0^{\boldsymbol{\times}} \leq 0.5$ and for different chain lengths $L = 16, \ldots, 784$ in Fig. 7. As constraints we use (a) $t_{\max} = \infty$, (b) $t_{\max} = L$, (c) $t_{\max} = L^{0.5}$, and (d) $t_{\max} = \text{const} = 20$. Instances that are not empty after $t_{\max}(L)$ time steps count as failed decodings, even if they are cleaned eventually (for $t \to \infty$). In addition, forbidden regions due to the light cone constraint (30) with $D = R\, t_{\max}(L)$ ($R = 4$) are shaded for $L = 16$ (orange) and $L = 784$ (black). Note hat for $t_{\max} = \infty$ it is $\overline{P}_{\text{dec}}^{\checkmark} = P_{\text{dec}}^{\checkmark}$, compare Eq. (62) and Eq. (70).

All numerical results satisfy the rigorous bounds of the light cone constraint which manifests as a weak upper bound for $p_0^{\boldsymbol{\times}}$ close to criticality. Note the difference between (a), (b), and (c) where the light cone constraint does not rule out successful decoding for any non-critical $p_0^{\boldsymbol{\times}}$ and $L \to \infty$, and (d) where it does. The rigorous results from above complete this picture by providing *lower* bounds that imply $\lim_{L \to \infty} \overline{P}_{\text{dec}}^{\checkmark} = 1$ for $p_0^{\boldsymbol{\times}} < \tilde{p}_c^{\boldsymbol{\times}}$. However, we do not know the true error threshold $p_c^{\boldsymbol{\times}}$ except that it is larger than $\tilde{p}_c^{\boldsymbol{\times}} \approx 3.2 \times 10^{-6}$ for $\overline{\text{TLV}}$ and

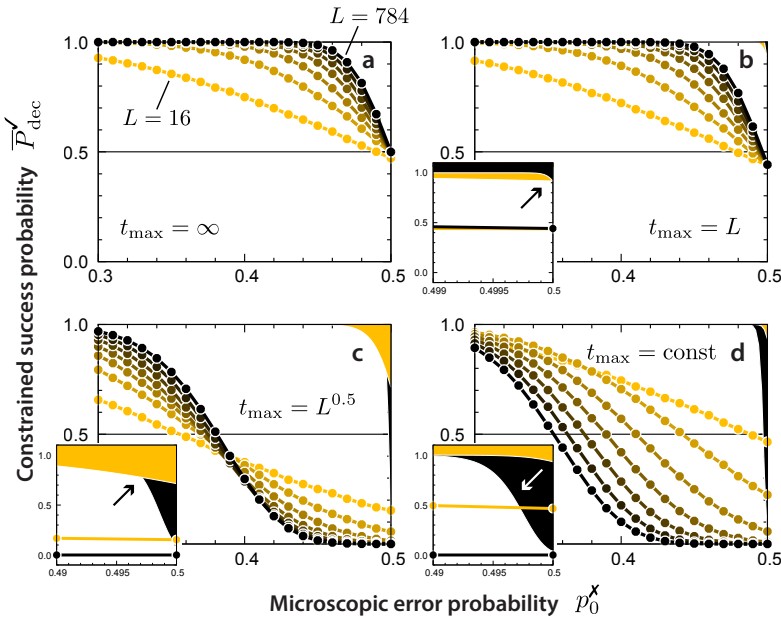

Figure 7: *Constrained $\overline{\text{TLV}}$-decoding.* The four plots show the probability $\overline{P}_{\text{dec}}^{\checkmark}$ of successfully decoding an initial Bernoulli random configuration with microscopic error probability $p_0^{\text{✗}}$ using $\overline{\text{TLV}}$ on chains of length $L = 16, \ldots, 784$. Evolutions that take longer than $t_{\max}(L)$ steps are regarded as failed decodings. Upper bounds from the light cone constraint are shaded orange (black) for $L = 16$ ($L = 784$) and compared with the respective numerical results in the insets close to $p_0^{\text{✗}} = 0.5$. We sampled over $10^6$ realizations for each data point. **a** Unrestricted: $t_{\max} = \infty$. The minimum $\overline{P}_{\text{dec}}^{\checkmark} = 0.5$ is reached only for $p_0^{\text{✗}} = 0.5$ for $L \to \infty$. **b** Linear: $t_{\max} = L$. The minimum of $\overline{P}_{\text{dec}}^{\checkmark}$ dips slightly below 0.5 for the critical system $p_0^{\text{✗}} = 0.5$. Presumably there is still perfect performance for $p_0^{\text{✗}} < 0.5$ and $L \to \infty$. **c** Sublinear: $t_{\max} = L^{0.5}$. The results suggest $\lim_{L\to\infty} \overline{P}_{\text{dec}}^{\checkmark} = 1$ for $0 \leq p_0^{\text{✗}} < p_c^{\text{✗}}$. Whether $p_c^{\text{✗}} < \frac{1}{2}$ cannot be inferred from numerics (note that the light cone constraint allows for $p_c^{\text{✗}} = \frac{1}{2}$). **d** Constant: $t_{\max} = \text{const}(= 20)$. As dictated by the light cone constraint, there is no decoding possible for $p_0^{\text{✗}} > 0$ and consequently $\lim_{L\to\infty} \overline{P}_{\text{dec}}^{\checkmark} = 0$.

therefore finite (see Appendix F). Fig. 7 (a) and (b) suggest that $p_c^{\text{✗}} = \frac{1}{2}$ for (super-)linear $t_{\max}$ which matches the performance of global majority voting (but also requires at least the same runtime scaling). In contrast, Fig. 7 (c) is compatible with a non-optimal $0 < p_c^{\text{✗}} < \frac{1}{2}$, even though we believe that still $p_c^{\text{✗}} = \frac{1}{2}$ due to a (slow) tendency of the crossing point towards $\frac{1}{2}$ for $L \to \infty$. Finally, Fig. 7 (d) confirms that $p_c^{\text{✗}} = 0$ for a fixed-depth decoder, in compliance with both Lemma 3 and the light cone constraint.

# 5 Error correction for continuous noise

So far, we focused on the *decoding* of initial error patterns $\mathbf{x}(0)$, the probability of success $P_{\text{dec}}^{\checkmark}$ (respectively $\overline{P}_{\text{dec}}^{\checkmark}$) and the time needed to clean the system $t_{\text{dec}}$. The ultimate goal, however, is the preservation of the logical qubit in the presence of *continuous noise*: While we assume error-free processing of classical information, decoherence of the quantum chain is a constant source of errors, characterized by the microscopic error rate $p_0^{\text{✗}}$ per time step $\delta t$. In this

section, we first demonstrate numerically that $\overline{\text{TLV}}$ *cannot* cope with such perturbations of its evolution in that the lifetime of the logical qubit grows only subexponentially with the chain length. In the second part, we resolve this problem by extending the $\overline{\text{TLV}}$ decoder into the second dimension, and show that its depth grows weakly (sublinearly) with the chain length. We conclude that shallow circuits suffice for reasonably low error rates.

## 5.1 Continuous noise in strictly one dimension

As a first step, we evaluate the performance of $\overline{\text{TLV}}$ as follows: Starting from an error-free chain, $\mathbf{x}(0) = \mathbf{0}$, we apply errors $\mathbf{x}'(t+1) = \mathbf{e}(t+1) \oplus \mathbf{x}(t)$ and $\overline{\text{TLV}}$-steps $\mathbf{x}(t+1) = \overline{\text{TLV}}_{\mathscr{L}}(\mathbf{x}'(t+1))$ in turns. Here $\mathbf{e}(t+1) \in \mathbb{Z}_2^{\mathscr{L}}$ is drawn from a Bernoulli distribution with parameter $p_0^{\boldsymbol{\mathsf{x}}}$ and describes the accumulated errors on the quantum chain between time $t$ and $t+1$. To quantify the ability of $\overline{\text{TLV}}$ to prevent errors from accumulating, we introduce the time to the first majority flip $T_{\text{ff}}$, i.e., $\text{maj}[\mathbf{x}(T_{\text{ff}})] = 1$ and $\text{maj}[\mathbf{x}(t)] = 0$ for $t < T_{\text{ff}}$. Sampling over many error histories $\{\mathbf{e}(t)\}$ yields the average $\langle T_{\text{ff}} \rangle$ characterizing the time scale over which the logical qubits survives (decay time).

Numerical results are shown in Fig. 8. In (a) $\langle T_{\text{ff}} \rangle$ is plotted versus $1/p_0^{\boldsymbol{\mathsf{x}}}$ for different lengths $L$, revealing an substantial growth of the decay time for $p_0^{\boldsymbol{\mathsf{x}}} \to 0$. In contrast, the dependence of $\langle T_{\text{ff}} \rangle$ on $L$ seems to be much less pronounced. This is confirmed in (b) where $\langle T_{\text{ff}} \rangle$ is plotted as function of the system size $L$ for two error rates $p_0^{\boldsymbol{\mathsf{x}}} = 0.050$ and $0.125$: The growth with $L$ is clearly subexponential, although the absolute scale of $\langle T_{\text{ff}} \rangle$ strongly depends on the error rate. To asses the gain in decay time by using $\overline{\text{TLV}}$, we compare it with global majority voting ($\triangle$; complete correction after each time step) and no correction at all ($\boldsymbol{\mathsf{x}}$; accumulating errors without corrective actions). As shown in Fig. 8 (b), global majority voting exhibits perfectly exponential growth of $\langle T_{\text{ff}} \rangle$ and outperforms $\overline{\text{TLV}}$ clearly. The comparison of a system without corrective actions and $\overline{\text{TLV}}$ reveals that the latter does *not* improve on the scaling but only increases the absolute values of $\langle T_{\text{ff}} \rangle$ and their susceptibility to variations in $p_0^{\boldsymbol{\mathsf{x}}}$. In conclusion, continuous noise thwarts the benefits one expects from making the quantum chain longer. This is in contrast to the previous sections where we considered the *decoding* of static error patterns and found an exponentially suppressed failure rate $P_{\text{dec}}^{\boldsymbol{\mathsf{x}}}$ for increasing chain length.

The susceptibility of $\overline{\text{TLV}}$ to continuous noise can be most easily understood by example: Fig. 8 (c) depicts the spacetime diagram encoding the evolution of an initial cluster of errors under $\overline{\text{TLV}}$ *without* continuous noise; as this is the decoding procedure discussed above, $\overline{\text{TLV}}$ erodes the cluster reliably. Since $\overline{\text{TLV}}$ effectively operates on the syndrome space, it is instructive to think of the evolution as attraction and subsequent annihilation of $\mathbb{Z}_2$-charges (the syndromes, red bullets). If continuous noise is switched on [Fig. 8 (d)], the attractive interaction is screened by a bath of noise-induced charge-anticharge pairs and the cluster's endpoints are governed by an undirected, diffusive process. From a renormalization-group perspective, there is a confinement-deconfinement transition at $p_0^{\boldsymbol{\mathsf{x}}} = 0$ which prevents the erosion of large clusters of errors, supporting their proliferation throughout the system. The susceptibility of simple one-dimensional CAs to continuous noise is a well-known phenomenon, see e.g. [36] for TLV and GKL. Indeed, due to the lack of counterexamples, it was conjectured that *all* one-dimensional CAs subject to noise are ergodic, that is, forget about their initial state eventually; this is known as the *positive rates conjecture* [44]. Peter Gács proved it wrong by providing an extraordinary complex counterexample that relies on self-simulation [26, 45, 46]. To the authors knowledge, there is no simpler counterexample known till this day, and it is widely believed that any non-ergodic CA in 1D must, in some form or another, implement the core mechanisms of Gács' automaton. It is therefore highly unlikely that a simple CA (such as $\overline{\text{TLV}}$) can retain information about its initial state for $t \to \infty$ if continuous noise is switched on. This is exactly what our numerical results suggest: The timescale $T_{\text{ff}}$ after which $\overline{\text{TLV}}$ forgets about

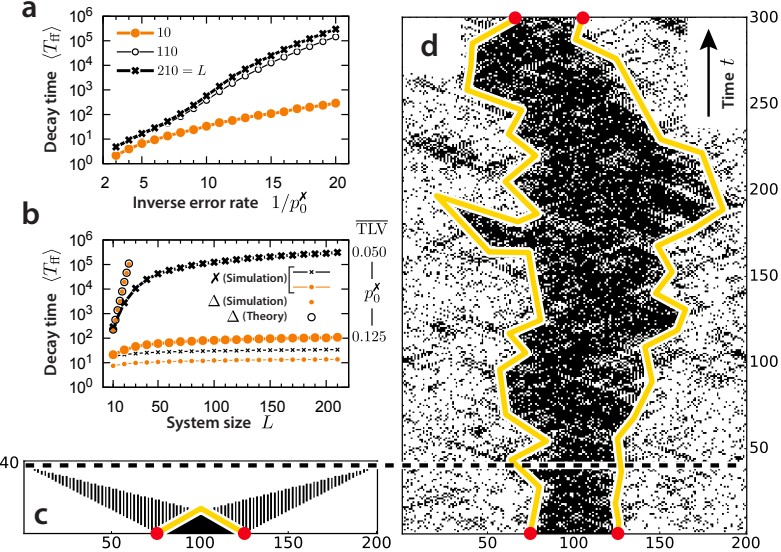

Figure 8: *Noise-induced deconfinement.* **a** Average time to the first majority flip $\langle T_{ff} \rangle$ vs. the inverse microscopic error rate $1/p_0^{\chi}$ for different system sizes $L = 10, \ldots, 210$ for $\overline{\text{TLV}}$. The dependence on $1/p_0^{\chi}$ is approximately exponential. **b** The same data vs. the system size $L$ for different error rates $p_0^{\chi} = 0.050$ and $0.125$ (joined bold crosses and bullets for $\overline{\text{TLV}}$). The numerics clearly suggests that there is *no* exponential growth of $\langle T_{ff} \rangle$ for $L \to \infty$, i.e., the storage time of the encoded qubit grows considerably slower than for *global* majority voting ($\Delta$) with constant correction rate; for comparison, we show simulations and theory for $\Delta$ with $p_0^{\chi} = 0.125$ (disjoined bullets and circles). With no correction ($\chi$), $\langle T_{ff} \rangle$ becomes almost constant (joined small crosses and bullets for $p_0^{\chi} = 0.050$ and $0.125$, respectively). For statistics, we sampled $10^3$ evolutions per data point to measure $T_{ff}$; the standard error of the shown sample mean is $\sim 3\%$ such that the error bars are not visible. **c** Spacetime diagram of a large cluster without continuous noise ($p_0^{\chi} = 0$) and $\overline{\text{TLV}}$- evolution. **d** Evolution of the same initial state with continuous noise $p_0^{\chi} = 0.1$ and the same scale as in **c**. Note that the messaging between left and right boundary of the cluster is jammed by the noise within the cluster, leading to an effective deconfinement of the charges at the cluster boundaries.

the initial majority does *not* diverge exponentially in the thermodynamic limit—an indicator for ergodicity.

## 5.2 Evading noise with a two-dimensional extension

To protect the evolution of $\overline{\text{TLV}}$ in the face of continuous noise, we pay with (classical) hardware by unrolling the time evolution into the (spatial) second dimension, perpendicular to the quantum chain. Then, our previous discussions and results on the *time* required for decoding translate directly into statements about the scaling of the *depth* of this "overhead dimension". We start with a description of the envisioned setup in Fig. 9. Note that the details of its implementation, described in the next few paragraphs, serve as a *proof of principle* only and may be subject to optimizations depending on the physical setup chosen for its realization.

We start with the quantum chain which is placed on top of a 2D substrate that hosts a classical two-layer circuit parallel and attached to the chain. The circuit has the topology of a cylinder glued to the quantum chain, see Fig. 9 (a); we illustrate both layers by slicing the cylinder along the chain and unfolding the circuit into the plane. The logical wiring of the circuit is sketched in Fig. 9 (b), (c) and (d) on various levels of detail. Its width equals

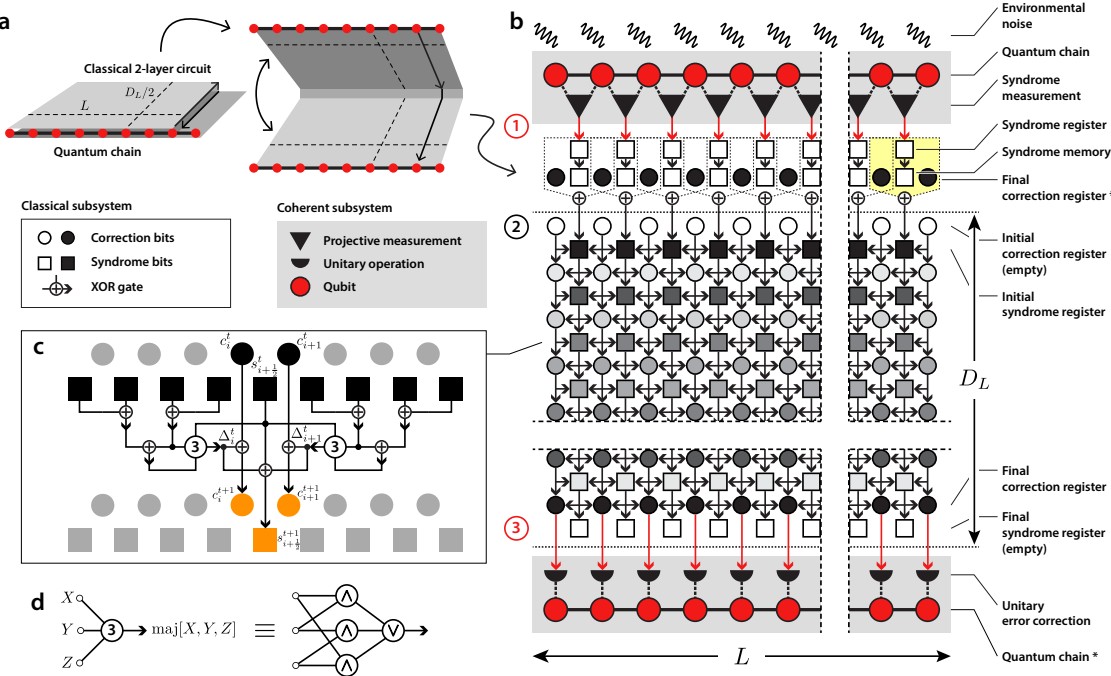

Figure 9: *2D-evolved* $\overline{\text{TLV}}$. **a** The quantum chain is placed on top and framed by a 2D substrate that allows for the implementation of classical two-layer circuitry (e.g., by photo lithography) that connects to projective (measurement-) and unitary gates along the chain. The classical circuits are used to process measurement results and control unitary gates in an integrated, scalable fashion. For illustrative purposes, the two layers are drawn unfolded with a copy of the quantum chain at top and bottom. The length of the chain is $L$ and the depth of the (unfolded) circuit is denoted by $D_L$, the scaling of which is discussed in the text. **b** Detailed setup to fight continuous noise on the quantum chain. Information propagates in a feed-forward manner from top (syndrome measurements) to bottom (correction operations). At the beginning of each time step, the syndrome pattern from projective measurements is fed into the `syndrome register` (Substep 1, red arrows). Subsequently, the content of all horizontal registers is evolved to the next layer (Substep 2, black arrows). Finally, the results in the `final correction register` are applied to the quantum chain (Substep 3, red arrows). The `initial syndrome register` is fed by the parity of `syndrome memory`, `syndrome register` and the syndrome of the last applied correction in the `final correction register` (indicated by the yellow box). The shading of classical bits (squares and circles) from black to white (or vice versa) illustrates the typical operation of the circuit: The `syndrome register` starts off in a non-empty state at the top whereas the cumulative `correction register` is initialized with all bits zero. Propagation to the bottom transforms the `correction register` into the non-trivial result of the decoding procedure while depleting the `syndrome register`. The latter reaches an empty state at the bottom (with high probability, see text). The elements marked by (*) are drawn twice for illustrative purposes and exist only once in hardware (see **a**). **c** Detailed logical flow in the scalable bulk that defines the new state of the next row in dependence of the state of the last row. The shown operations implement the evolution of $\overline{\text{TLV}}$ in syndrome-delta representation directly on the `syndrome registers` (squares) while accumulating all applied operations on the qubits in the `correction registers` (circles). Note that actual qubit rotations are only applied once at the bottom, defined by the state of the `final correction register`. For the sake of compactness, we write $c_i^t = c_i(t)$ etc. as compared to the text. **d** Besides the ubiquitous XOR-gates, the only additional gate required is the majority gate with three inputs (MAJ3-gate) which can be easily translated into a network of elementary AND- and OR-gates.

the length of the chain $L$ whereas the depth $D_L$ (folded up $D_L/2$) is arbitrary (up to a fixed overhead), see (b) and below. All syndrome measurements are performed periodically—once per time step—and fed into the `syndrome register` in the top layer of the circuit [Substep 1 in (b), red]. Subsequently each (binary) cell of the two-dimensional classical system is updated synchronously according to specific rules that take as inputs values of spatially adjacent cells [Substep 2 in (b), black]. Finally, set bits in the `final correction register` on the lower layer (again adjacent to the chain) are used to determine the unitary error correction, the application of which completes a single time step [Substep 3 in (b), red]. Note that due to the spatially local computations, the time needed for a single time step is constant and does neither scale with $L$ nor with $D_L$.

The rules defining the classical automaton (applied in Substep 2) can be divided roughly into two functional classes [see Fig. 9 (b) Substep 2]. The first is independent of the depth $D_L$ and located close to the chain. It consists of the `syndrome register` (upper layer), `syndrome memory` (upper layer), and the `final correction register` (lower layer) and computes the syndrome of errors that accumulated since the last syndrome measurement, taking into account the correction operations at the end of the last step. Formally,

$$\mathbf{x}(t+1) = \mathbf{e}(t+1) \oplus \mathbf{c}(t) \oplus \mathbf{x}(t) \tag{71a}$$

$$\Rightarrow \partial \mathbf{e}(t+1) = \partial \mathbf{x}(t+1) \oplus \partial \mathbf{x}(t) \oplus \partial \mathbf{c}(t) = \mathbf{s}(t+1) \oplus \mathbf{s}(t) \oplus \partial \mathbf{c}(t), \tag{71b}$$

with

$$\partial c_{i+\frac{1}{2}}(t) = c_i(t) \oplus c_{i+1}(t), \tag{72}$$

where $\mathbf{s}(t+1)$ denotes the newly measured syndrome (in the `syndrome register`), $\mathbf{s}(t)$ is the *previously* measured syndrome (in the `syndrome memory`), and $\mathbf{c}(t)$ encodes the previously applied correction (in the `final correction register`); the (inaccessible) error configuration is $\mathbf{x}(t)$ and $\mathbf{e}(t+1)$ denotes the accumulated errors during $[t, t+1]$. Eventually, the `syndrome memory` is overwritten with the values of the `syndrome register`.

The result (71b) describes *only errors* that occurred in the previous time interval $[t, t+1]$ and ignores both older errors (which are already taken into account) and previous corrections (which are not to be "corrected"). (71b) is fed into the first row (`initial syndrome register`) of the second sector, a translationally invariant 2D circuit (except for the boundaries) with freely adjustable depth $D_L$. Its purpose is to simulate $\overline{\mathrm{TLV}}$ in the syndrome-delta representation in a feed-forward manner, from top to bottom in Fig. 9 (b), where $\boldsymbol{\Delta} = \partial \overline{\mathrm{TLV}}_{\mathscr{L}}(\mathbf{s})$ is accumulated modulo 2 in the `correction registers` (circles) and $\mathbf{s}' = \partial \boldsymbol{\Delta} \oplus \mathbf{s}$ is written into the `syndrome register` of the next row. $\partial \overline{\mathrm{TLV}}_{\mathscr{L}}$ is given by Eq. (55) and the MBC modifications in (56) and (57)—and can be implemented as illustrated in Fig. 9 (c):

Notably, there are only two types of logic gates required, both of which can be easily reduced to elementary gates. The `XOR`-gates ($\oplus$) are equivalent to $X \oplus Y = (X \vee Y) \wedge \neg(X \wedge Y)$ while the majority gates on three bits (`MAJ3`) can be rewritten as

$$\mathrm{maj}[X, Y, Z] = (X \wedge Y) \vee (X \wedge Z) \vee (Y \wedge Z), \tag{73}$$

see Fig. 9 (d). The `XOR`-gate can be realized in established CMOS technology with only 3 transistors [66, 67], while the `MAJ3`-gate requires about 14 transistors. However, going beyond CMOS may be beneficial [68], depending on the environment preferred by the coherent subsystem (the quantum chain): Whereas the `MAJ3`-gate is rather complex in CMOS technology, it becomes an elementary logic gate in the framework of quantum-dot cellular automata [69]. The second sector evolves $\overline{\mathrm{TLV}}$ on $\partial \mathbf{e}(t+1)$ *in space* and thereby circumvents the noise-induced deconfinement because subsequent errors $\mathbf{e}(t+2)\ldots$ are tackled by a completely decoupled evolution of $\overline{\mathrm{TLV}}$. If $\overline{\mathrm{TLV}}$ decodes the syndrome $\partial \mathbf{e}(t+1)$ successfully in

$$t_{\mathrm{dec}} \leq t_{\mathrm{max}}(L) = D_L \tag{74}$$

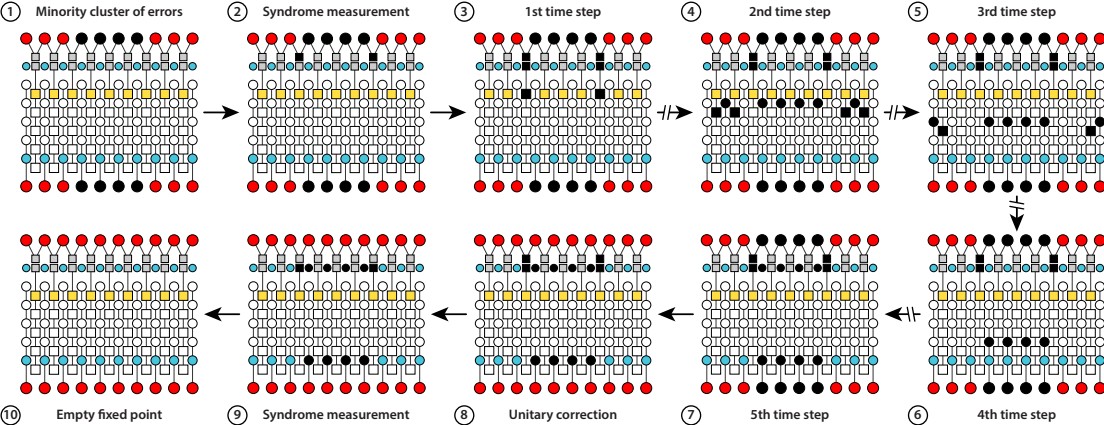

Figure 10: *Exemplary evolution of 2D-evolved* $\overline{\text{TLV}}$. Exemplary evolution of a depth $D_L = 5$ 2D-evolved $\overline{\text{TLV}}$-automaton of length $L = 10$ without continuous noise, starting from a contiguous minority cluster of errors in the center. (See Fig. 9 (b) for a description of the setup.) The `initial syndrome register` and the `final correction register` are highlighted yellow and green, respectively. A copy of the `final correction register` is reproduced between `syndrome register` and `syndrome memory` (gray) to emphasize that the initialization of the `initial syndrome register` depends on all of them. Shown are 6 time steps in total, each consisting of three substeps: syndrome measurement, a single step of the 2D CA, and a unitary correction. We omit trivial substeps (measurements and corrections), indicated by broken arrows. The first time step comprises frames 1-3 where the correction step is omitted at the end. Time steps 2-4 are shown in frames 4-6 where both measurements and corrections are omitted. The 5th time step starts in frame 7 and ends with the first (and last) non-trivial correction in frame 8. The 6th and final time step starts with a non-trivial syndrome measurement in frame 9 and resets the CA to the empty fixed point in frame 10. Details are given in the text.

time steps, the `final syndrome register` is empty at $t^* = t + 1 + D_L$ and the `final correction register` contains $\mathbf{c}(t^*) = \mathbf{e}(t + 1)$ which is applied to the quantum chain in Substep 3 to cancel the errors $E(\mathbf{e}(t + 1))$. We point out that the occurrence of errors $\mathbf{e}$ and the application of corresponding corrections $\mathbf{c}$ are separated by $D_L$ time steps, the depth of the circuit, which reflects the finite speed of information transfer in a spatially extended decoder [recall Fig. 2 (c)]. Since errors and correction operations commute, this is not an issue.

We demonstrate the evolution of the complete circuit for a single (minority) cluster of errors without continuous noise in Fig. 10 on a chain of length $L = 10$ with depth $D_L = 5$. Note how the `syndrome memory` prevents the automaton from issuing multiple instances of the same computation (Frames 4-6), and how the `final correction register` prevents the "correction of the correction" in Frame 9. Most importantly, the classical subsystem only uses syndrome information and is *not* aware of the actual error pattern (which we plot as blacked out qubits for convenience only).

Our setup fails to protect the qubit if, at some point in time, an error pattern accumulates during a single time step which cannot be successfully corrected by $\overline{\text{TLV}}$ within $D_L$ time steps. This may be because it is eroded to **1** instead of **0** or there are syndromes left after $D_L$ time steps so that residual errors survive. The last case splits into two subcases: First, $\overline{\text{TLV}}$ might have succeeded and reached **0** after $t_{\text{dec}} > D_L$ time steps, or, second, the initial configuration was in the attractor of a cycle such that no correction was possible anyway (even for $t \to \infty$).

The time that quantifies the performance of our setup is then the *time-to-first-failure* $T_{\text{tff}}$ ("decay time"), i.e., the time after which the first uncorrectable (in the above sense) error

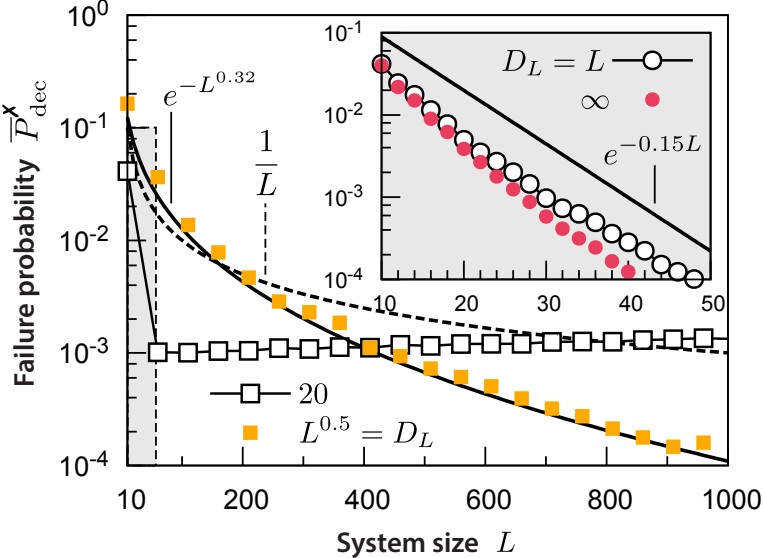

Figure 11: *Failure probability for 2D-evolved* $\overline{\text{TLV}}$. Failure probability $\overline{P}^{\textbf{X}}_{\text{dec}}$ of decoding error configurations with a 2D-evolved $\overline{\text{TLV}}$-decoder of depth $D_L$ as a function of the system size $L$ for microscopic error rate $p^{\textbf{X}}_0 = 0.2$. Note that failed decodings include both syndrome-free states with corrupted logical qubit and states with residual syndromes. We sampled $5 \cdot 10^6$ instances per data point. The gray rectangle (dashed boundary) is shown as inset. The solid and dashed lines show the analytic functions $e^{-L^{0.32}}$, $1/L$, and $e^{-0.15L}$ as a guide to the eye; these are neither fits nor analytical results. We compare setups with constant-depth $D_L = \text{const} = 20$ (empty squares) and sublinear-depth $D_L = L^{0.5}$ (filled squares). Note that the curves intersect for $L = 400$ because $\sqrt{400} = 20$. The inset depicts the much faster decreasing cases of unbounded depth $D_L = \infty$ (filled circles) and linear depth $D_L = L$ (empty circles). There is no qualitative difference between the two for the shown parameters.

pattern appears. Its expectation value is given by

$$\langle T_{\text{tff}} \rangle = \sum_{T=1}^{\infty} T \, \overline{P}^{\textbf{X}}_{\text{dec}} (1 - \overline{P}^{\textbf{X}}_{\text{dec}})^{T-1} = \frac{1}{\overline{P}^{\textbf{X}}_{\text{dec}}} \, , \tag{75}$$

where $\overline{P}^{\textbf{X}}_{\text{dec}} = 1 - \overline{P}^{\checkmark}_{\text{dec}}$ denotes the restricted failure probability of $\overline{\text{TLV}}$ with $t_{\max}(L) = D_L$, as discussed previously. Eq. (75) follows because each time step corresponds to a Bernoulli sample independent of the previous error patterns, a consequence of the spatial evolution of $\overline{\text{TLV}}$ in our 2D circuit. In Fig. 11 we show simulations of $\overline{P}^{\textbf{X}}_{\text{dec}}$ as function of $L$ for fixed error rate $p^{\textbf{X}}_0 = 0.2$ and four different depth scalings $D_L$ [see also Fig. 7]. Decreasing failure probabilities $\overline{P}^{\textbf{X}}_{\text{dec}}$ translate via Eq. (75) into growing decay times $\langle T_{\text{tff}} \rangle$: A constant depth $D_L = 20$ decoder does not benefit from longer quantum chains whereas both "infinite depth" and linear depth decoder perform similarly and yield exponentially increasing decay times $T_{\text{tff}}$. Decoders with slowly growing algebraic depths, such as the shown $D_L = L^{0.5}$, still exhibit exponential growth of $T_{\text{tff}}$, although weaker than that of global decoders with $D_L \gtrsim L$.

# 6  Conclusion

Motivated by the requirement for scalable and modular decoders for topological quantum memories, we set out to construct a strictly local decoder for the one-dimensional Majorana chain quantum code. As the latter constitutes the quantum analogue of the classical repetition code, it can be efficiently decoded and stabilized by global majority voting *if* spatio-temporal constraints are ignored. Taking into account the time needed for classical syndrome processing and communication suggests the implementation of decoders as cellular automata. We argued that the decoding problem at hand translates into the problem of one-dimensional density classification with the additional symmetry constraint of self-duality; this led us to the *two-line voting* automaton TLV as promising local decoder. We equipped the latter with mirrored boundaries (called $\overline{\mathrm{TLV}}$) to comply with the requirement of open boundaries on the level of the quantum chain.

Both numerics and rigorous analytical results showed that $\overline{\mathrm{TLV}}$ succeeds in decoding Bernoulli random patterns with exponentially vanishing failure rate as $L \to \infty$. Whereas the rigorous results are restricted to small but finite microscopic error probabilities $p_0^{\chi} < \tilde{p}_c^{\chi} \approx 3.2 \times 10^{-6}$, numerics suggest that $p_0^{\chi} < \frac{1}{2}$ may be enough for successful decoding. In addition, the time needed for decoding scales sublinearly for $p_0^{\chi} < \frac{1}{2}$. In particular, we proved that the failure rate for decoding a code of length $L$ in at most $t \propto L^{\kappa}$ time steps ($\kappa > 0$ arbitrary) vanishes exponentially with $L \to \infty$ for small but finite $p_0^{\chi} < \tilde{p}_c^{\chi}$. In a nutshell: for low error rates, *global* majority voting is not required.

In the final section we investigated the performance of $\overline{\mathrm{TLV}}$ in the presence of continuous noise. In accordance with the expected ergodicity of simple, one-dimensional cellular automata, we argued that $\overline{\mathrm{TLV}}$ *cannot* fight continuous noise because long-range communication is cut off by locally created charge-anticharge pairs. As a consequence, we had to evolve $\overline{\mathrm{TLV}}$ into the second dimension to prevent errors from accumulating during the syndrome processing. Thereby the superior (i.e., sublinear) scaling of decoding times for $\overline{\mathrm{TLV}}$—as opposed to the linear scaling of global majority voting—was turned into a modest scaling of classical hardware overhead: For reasonably low error rates, simple, shallow circuits, lacking the capability of global communication, can replace the hardware-expensive global majority voting. These results add to the quest of scalable and modular realizations of actively corrected topological quantum memories.

# Acknowledgments

N.L. thanks S. Taati for useful comments on the latter's proof of strongly sparse Bernoulli random sets. This research has received funding from the European Research Council (ERC) under the European Union's Horizon 2020 research and innovation programme (grant agreement No 681208). This research was supported in part by the National Science Foundation under Grant No. NSF PHY-1125915.

# A Cumulative Bernoulli distribution

Here we prove some of the statements about global majority voting used in Sections 2.2 and 2.3 of the main text. To this end, we start with the probability for more than half of $L$ (odd) binary sites $x_i$ to be error afflicted (i.e., in state $x_i = 1$) after $t$ rounds of additive, uncorrelated Bernoulli noise:

$$P_{L,p_0^{\mathsf{X}}}^{\mathsf{X}}(t) = \sum_{k=\frac{L+1}{2}}^{L} \binom{L}{k} \left[ p^{\mathsf{X}}(t) \right]^k \left[ 1 - p^{\mathsf{X}}(t) \right]^{L-k}, \tag{76}$$

where

$$p^{\mathsf{X}}(t) = \frac{1}{2} \left[ 1 - \left( 1 - 2p_0^{\mathsf{X}} \right)^t \right] \tag{77}$$

is the renormalized single-site probability for $X(t) = X_1 \oplus \cdots \oplus X_t$ with $X_i$ Bernoulli random variables with parameter $p_0^{\mathsf{X}}$, i.e.,

$$\Pr(X(t) = 1) = \sum_{k \, \text{odd}} \binom{t}{k} \left( p_0^{\mathsf{X}} \right)^k \left( 1 - p_0^{\mathsf{X}} \right)^{t-k} = \frac{1}{2} \left[ 1 - \left( 1 - 2p_0^{\mathsf{X}} \right)^t \right]. \tag{78}$$

Eq. (76) is a special case of the cumulative Bernoulli distribution function which is known to be expressible in a closed form by the incomplete beta function,

$$B(x; a, b) = \int_0^x y^{a-1} (1-y)^{b-1} \, \mathrm{d}y \tag{79}$$

via [50]

$$P_{L,p_0^{\mathsf{X}}}^{\mathsf{X}}(t) = I_{p^{\mathsf{X}}(t)} \left( \frac{L+1}{2}, L - \frac{L+1}{2} + 1 \right), \tag{80}$$

where $I_x(a, b) = B(x; a, b)/B(1; a, b)$ is called *regularized* incomplete beta function. With $a, b \in \mathbb{N}$ we can use

$$B(1; a, b) = \frac{\Gamma(a)\Gamma(b)}{\Gamma(a+b)} = \frac{(a-1)!(b-1)!}{(a+b-1)!} \tag{81}$$

to evaluate

$$B\left( 1; \frac{L+1}{2}, L - \frac{L+1}{2} + 1 \right) = \left[ \frac{L+1}{2} \binom{L}{\frac{L+1}{2}} \right]^{-1}. \tag{82}$$

Then Eq. (80) reads

$$P_{L,p_0^{\mathsf{X}}}^{\mathsf{X}}(t) = \frac{L+1}{2} \binom{L}{\frac{L+1}{2}} \int_0^{p^{\mathsf{X}}(t)} \left[ x - x^2 \right]^{\frac{L-1}{2}} \mathrm{d}x, \tag{83}$$

which is a useful form to derive estimates and limits of Eq. (76). In particular, we can now derive the limit for $0 < p_0^{\mathsf{X}} \leq \frac{1}{2}$ and $0 < c < \infty$

$$\lim_{L \to \infty} P_{L,p_0^{\mathsf{X}}}^{\mathsf{X}}(t = c^{-1}L) = \frac{1}{2} \tag{84}$$

if we use the asymptotic expression (Stirling formula)

$$\binom{L}{\frac{L+1}{2}} \sim \sqrt{\frac{2}{\pi}} \frac{2^L}{\sqrt{L}} \quad \text{for} \quad L \to \infty. \tag{85}$$

Indeed, with $t = \frac{L}{c}$

$$P^{\mathsf{X}}_{L, p^{\mathsf{X}}_0}(t) = \frac{L+1}{2} \binom{L}{\frac{L+1}{2}} \int_0^{p^{\mathsf{X}}(t)} [x - x^2]^{\frac{L-1}{2}} \, dx \tag{86a}$$

$$\sim \frac{\sqrt{L}}{\sqrt{2\pi}} \int_0^{p^{\mathsf{X}}(t)} [2(2x) - (2x)^2]^{\frac{L-1}{2}} \, d(2x) \tag{86b}$$

$$\stackrel{y=2x}{=} \frac{\sqrt{L}}{\sqrt{2\pi}} \int_0^{2p^{\mathsf{X}}(t)} [2y - y^2]^{\frac{L-1}{2}} \, dy \tag{86c}$$

$$\stackrel{u=y-1}{=} \frac{\sqrt{L}}{\sqrt{2\pi}} \int_{-1}^{2p^{\mathsf{X}}(t)-1} [1 - u^2]^{\frac{L-1}{2}} \, du \tag{86d}$$

$$\sim \frac{1}{\sqrt{\pi}} \int_{-\sqrt{\frac{L-1}{2}}}^{-\sqrt{\frac{L-1}{2}}(1-2p^{\mathsf{X}}_0)^{\frac{L}{c}}} \left[1 - \frac{x^2}{\frac{L-1}{2}}\right]^{\frac{L-1}{2}} \, dx \tag{86e}$$

where we used the substitution $x = \sqrt{\frac{L-1}{2}} u$ in the last row. If we use that $(0 < p^{\mathsf{X}}_0 \leq \frac{1}{2})$

$$\lim_{L \to \infty} \int_{-\sqrt{\frac{L-1}{2}}(1-2p^{\mathsf{X}}_0)^{\frac{L}{c}}}^0 \left[1 - \frac{x^2}{\frac{L-1}{2}}\right]^{\frac{L-1}{2}} \, dx = 0 \tag{87}$$

and

$$\lim_{n \to \infty} \int_{-n}^0 \left(1 - \frac{x^2}{n^2}\right)^{n^2} dx = \lim_{n \to \infty} n \int_0^1 (1 - y^2)^{n^2} \, dy = \frac{\sqrt{\pi}}{2} \lim_{n \to \infty} \frac{n \, \Gamma(1 + n^2)}{\Gamma(\frac{3}{2} + n^2)} = \frac{\sqrt{\pi}}{2}, \tag{88}$$

we find for $0 < p^{\mathsf{X}}_0 \leq \frac{1}{2}$ and $0 < c < \infty$ the final result

$$\lim_{L \to \infty} P^{\mathsf{X}}_{L, p^{\mathsf{X}}_0}(c^{-1}L) = \frac{1}{2} = \frac{1}{\sqrt{\pi}} \int_{-\infty}^0 e^{-x^2} \, dx \,. \tag{89}$$

Note that $\lim_{L \to \infty} P^{\mathsf{X}}_{L, p^{\mathsf{X}}_0}(c^{-1}L) > 0$ only because $p^{\mathsf{X}}(t)$ renormalizes to $\frac{1}{2}$ exponentially fast with $t$, such that the upper bound of the integral converges to zero. Similarly, for $\lim_{L \to \infty} P^{\mathsf{X}}_{L, p^{\mathsf{X}}_0}(t = 1)$ one easily re-derives the exponential decay to zero (modified by $\sqrt{L}$ from the integral bounds).

## B  Light cone constraint

### B.1  Derivation

Here we prove the following upper bound for the decoding probability $P^{\checkmark}_{\text{dec}}$ of a $D$-local physical decoder of linear size $L$:

$$P^{\checkmark}_{\text{dec}} \leq \left[1 + \left(\frac{p^{\mathsf{X}}_0}{1 - p^{\mathsf{X}}_0}\right)^{2D+1}\right]^{-\frac{L}{2D+1}}. \tag{90}$$

The microscopic error probability per qubit and time step is $p^{\mathsf{X}}_0$. Let the error pattern be described by the vector $\mathbf{x}$ of length $L$ with syndrome $\mathbf{s} = \partial \mathbf{x}$. A given $D$-local decoder $\Delta^D$ then

calculates a correction $\Delta^D(\mathbf{s})$ such that $\Delta^D(\mathbf{s}) \oplus \mathbf{x}$ describes the new error state after the correction has been applied. For an arbitrary but fixed site $1 \le i \le L$ we define two sets:

$$\mathscr{X}_i^{\checkmark} \equiv \left\{ \mathbf{x} \,|\, \Delta_i^D(\mathbf{s}) \oplus x_i = 0 \right\}, \tag{91a}$$

$$\mathscr{X}_i^{\times} \equiv \left\{ \mathbf{x} \,|\, \Delta_i^D(\mathbf{s}) \oplus x_i = 1 \right\}. \tag{91b}$$

$\mathscr{X}_i^{\checkmark}$ and $\mathscr{X}_i^{\times}$ describe the sets of all error patterns that $\Delta^D$ (un)successfully corrects at site $i$, respectively. We define the local complement operator $C_i^D$ such that for $\mathbf{x}' = C_i^D \mathbf{x}$

$$x_j' = \begin{cases} x_j & \text{for } |j - i| > D \\ x_j \oplus 1 & \text{for } |j - i| \le D \end{cases}, \tag{92}$$

i.e., it inverts the error pattern in a region of radius $D$ around site $i$. Clearly $C_i^D \circ C_i^D = \mathbb{1}$, such that $C_i^D$ defines a bijection on the total error state space $\mathscr{X} = \mathscr{X}_i^{\checkmark} \dot\cup \mathscr{X}_i^{\times} = \mathbb{Z}_2^L$. Let $\pi_i^D : \mathscr{X} \to \mathbb{Z}_2^{2D+1}$ be the projector that slices the range on which $C_i^D$ acts non-trivially from a state $\mathbf{x}$. We have $\partial \mathbf{x} \ne \partial C_i^D \mathbf{x}$ due to the boundaries of the partial complement. However,

$$\partial \pi_i^D \mathbf{x} = \partial \pi_i^D C_i^D \mathbf{x} \tag{93}$$

since the syndrome does not change inside the range of the local complement. We can now define the two sets

$$\bar{\mathscr{X}}_i^{\checkmark} \equiv C_i^D \mathscr{X}_i^{\checkmark} = \left\{ C_i^D \mathbf{x} \,|\, \mathbf{x} \in \mathscr{X}_i^{\checkmark} \right\}, \tag{94a}$$

$$\bar{\mathscr{X}}_i^{\times} \equiv C_i^D \mathscr{X}_i^{\times} = \left\{ C_i^D \mathbf{x} \,|\, \mathbf{x} \in \mathscr{X}_i^{\times} \right\}. \tag{94b}$$

Since $C_i^D$ is a bijection, we still have $\mathscr{X} = \bar{\mathscr{X}}_i^{\checkmark} \dot\cup \bar{\mathscr{X}}_i^{\times}$.

Now comes a crucial step: Because $\Delta^D$ is $D$-local, its action on site $i$ only depends on the syndromes within $\pi_i^D \mathbf{x}$, i.e., $\partial \pi_i^D \mathbf{x}$. Therefore we find that if $\mathbf{x} \in \mathscr{X}_i^{\checkmark}$ is successfully corrected at site $i$, then $\Delta^D$ fails to correct $C_i^D \mathbf{x}$ because its action on site $i$ is the same. In a nutshell,

$$\mathbf{x} \in \mathscr{X}_i^{\checkmark} \iff C_i^D \mathbf{x} \in \mathscr{X}_i^{\times}. \tag{95}$$

Thus we have $\bar{\mathscr{X}}_i^{\checkmark} = \overline{\mathscr{X}_i^{\checkmark}} = \mathscr{X}_i^{\times}$ and $\bar{\mathscr{X}}_i^{\times} = \overline{\mathscr{X}_i^{\times}} = \mathscr{X}_i^{\checkmark}$ where $\overline{\bullet}$ denotes the complement in $\mathscr{X}$.

For a Bernoulli process, the probability of the error pattern $\mathbf{x}$ is

$$\Pr_{p_0^{\times}}(\mathbf{x}) = \left( p_0^{\times} \right)^{|\mathbf{x}|} \left( 1 - p_0^{\times} \right)^{L - |\mathbf{x}|}, \tag{96}$$

with $|\mathbf{x}| = \sum_i x_i$ the total number of errors. For the local complement, we have

$$\Pr_{p_0^{\times}}\left( C_i^D \mathbf{x} \right) = \left( p_0^{\times} \right)^{|C_i^D \mathbf{x}|} \left( 1 - p_0^{\times} \right)^{L - |C_i^D \mathbf{x}|} = \left( \frac{p_0^{\times}}{1 - p_0^{\times}} \right)^{|C_i^D \mathbf{x}| - |\mathbf{x}|} \Pr_{p_0^{\times}}(\mathbf{x}), \tag{97}$$

where

$$|C_i^D \mathbf{x}| - |\mathbf{x}| = (2D + 1) - 2|\pi_i^D \mathbf{x}| \tag{98}$$

is the change of errors in the light cone due to the local complement. Thus

$$\Pr_{p_0^{\times}}\left( C_i^D \mathbf{x} \right) = \left( \frac{p_0^{\times}}{1 - p_0^{\times}} \right)^{(2D+1) - 2|\pi_i^D \mathbf{x}|} \Pr_{p_0^{\times}}(\mathbf{x}). \tag{99}$$

We start from the trivial relation

$$\sum_{\mathbf{x} \in \mathscr{X}_i^{\checkmark}} \Pr_{p_0^{\times}}(\mathbf{x}) + \sum_{\mathbf{x} \in \mathscr{X}_i^{\times}} \Pr_{p_0^{\times}}(\mathbf{x}) = 1 \tag{100}$$

and rewrite the first term

$$\sum_{\mathbf{x}\in\mathscr{X}_i^{\checkmark}}\mathrm{Pr}_{p_0^{\mathsf{X}}}(\mathbf{x})=\sum_{\mathbf{x}\in\bar{\mathscr{X}}_i^{\checkmark}}\mathrm{Pr}_{p_0^{\mathsf{X}}}\left(C_i^D\mathbf{x}\right)=\sum_{\mathbf{x}\in\mathscr{X}_i^{\mathsf{X}}}\left(\frac{p_0^{\mathsf{X}}}{1-p_0^{\mathsf{X}}}\right)^{(2D+1)-2|\pi_i^D\mathbf{x}|}\mathrm{Pr}_{p_0^{\mathsf{X}}}(\mathbf{x})\,, \tag{101}$$

which yields

$$\sum_{\mathbf{x}\in\mathscr{X}_i^{\mathsf{X}}}\left[1+\left(\frac{p_0^{\mathsf{X}}}{1-p_0^{\mathsf{X}}}\right)^{(2D+1)-2|\pi_i^D\mathbf{x}|}\right]\mathrm{Pr}_{p_0^{\mathsf{X}}}(\mathbf{x})=1\,. \tag{102}$$

So far, all statements are exact and valid for $0\leq p_0^{\mathsf{X}}\leq 1$. Now we assume $0\leq p_0^{\mathsf{X}}\leq\frac{1}{2}$ and estimate

$$\left(\frac{p_0^{\mathsf{X}}}{1-p_0^{\mathsf{X}}}\right)^{(2D+1)-2|\pi_i^D\mathbf{x}|}\leq\left(\frac{1-p_0^{\mathsf{X}}}{p_0^{\mathsf{X}}}\right)^{2D+1}\,, \tag{103}$$

where we used that $|\pi_i^D\mathbf{x}|\leq 2D+1$ and $p_0^{\mathsf{X}}/(1-p_0^{\mathsf{X}})\leq 1$ for $p_0^{\mathsf{X}}\leq\frac{1}{2}$. We have

$$1\leq\left[1+\left(\frac{1-p_0^{\mathsf{X}}}{p_0^{\mathsf{X}}}\right)^{2D+1}\right]\cdot\sum_{\mathbf{x}\in\mathscr{X}_i^{\mathsf{X}}}\mathrm{Pr}_{p_0^{\mathsf{X}}}(\mathbf{x}) \tag{104}$$

and therefore the lower bound on the error probability

$$\left[1+\left(\frac{1-p_0^{\mathsf{X}}}{p_0^{\mathsf{X}}}\right)^{2D+1}\right]^{-1}\leq\sum_{\mathbf{x}\in\mathscr{X}_i^{\mathsf{X}}}\mathrm{Pr}_{p_0^{\mathsf{X}}}(\mathbf{x})\,. \tag{105}$$

Note that this is the probability that an error at site $i$ survives a single correction procedure with $\Delta^D$. This lower bound can be easily recast as an upper bound on the probability of successful correction,

$$\sum_{\mathbf{x}\in\mathscr{X}_i^{\checkmark}}\mathrm{Pr}_{p_0^{\mathsf{X}}}(\mathbf{x})=1-\sum_{\mathbf{x}\in\mathscr{X}_i^{\mathsf{X}}}\mathrm{Pr}_{p_0^{\mathsf{X}}}(\mathbf{x})\leq\left[1+\left(\frac{p_0^{\mathsf{X}}}{1-p_0^{\mathsf{X}}}\right)^{2D+1}\right]^{-1}\,. \tag{106}$$

The last step is to use this result for an upper bound on the *global* correction probability

$$P_{\mathrm{dec}}^{\checkmark}=\sum_{\mathbf{x}\in\bigcap_i\mathscr{X}_i^{\checkmark}}\mathrm{Pr}_{p_0^{\mathsf{X}}}(\mathbf{x})=\sum_{\mathbf{x}\in\mathscr{X}}\mathrm{Pr}_{p_0^{\mathsf{X}}}(\mathbf{x})\prod_i\mathbb{1}_{\mathscr{X}_i^{\checkmark}}(\mathbf{x})=\left\langle\prod_i\mathbb{1}_{\mathscr{X}_i^{\checkmark}}\right\rangle\neq\prod_i\left\langle\mathbb{1}_{\mathscr{X}_i^{\checkmark}}\right\rangle\,, \tag{107}$$

where $\mathbb{1}_{\mathscr{X}_i^{\checkmark}}$ denotes the indicator function of $\mathscr{X}_i^{\checkmark}$. The last inequality follows from the fact that $\mathbb{1}_{\mathscr{X}_i^{\checkmark}}$ and $\mathbb{1}_{\mathscr{X}_j^{\checkmark}}$ may be correlated random variables for $|i-j|<2D+1$, i.e., if their past light cones overlap and they depend on common syndrome measurements. This motivates the second estimate

$$\left\langle\prod_i\mathbb{1}_{\mathscr{X}_i^{\checkmark}}\right\rangle\leq\left\langle\prod_{k=1}^{L/(2D+1)}\mathbb{1}_{\mathscr{X}_{k(2D+1)}^{\checkmark}}\right\rangle\,, \tag{108}$$

where we assume for simplicity that $L$ is a multiple of $2D+1$. We can separate the system into subsystems $\mathbf{x}_k$ of length $2D+1$ such that $\mathbb{1}_{\mathscr{X}_{k(2D+1)}^{\checkmark}}(\mathbf{x})=\mathbb{1}_{\mathscr{X}_{k(2D+1)}^{\checkmark}}(\mathbf{x}_k)$ (slight abuse of notation) with $\mathbf{x}_k=\pi_{k(2D+1)}^D\mathbf{x}$. Here we use the fact that the correctability of site $k(2D+1)$ only depends

on a causal region of radius $D$. The last step is to realize that $\text{Pr}_{p_0^{\times}}(\mathbf{x})$ is a product measure due to the uncorrelated Bernoulli process,

$$\text{Pr}_{p_0^{\times}}(\mathbf{x}) = \prod_k \text{Pr}_{p_0^{\times}}(\mathbf{x}_k)\,, \tag{109}$$

so that

$$\left\langle \prod_{k=0}^{L/(2D+1)} \mathbb{1}_{\mathscr{X}_{k(2D+1)}^{\checkmark}} \right\rangle = \prod_{k=1}^{L/(2D+1)} \left\langle \mathbb{1}_{\mathscr{X}_{k(2D+1)}^{\checkmark}} \right\rangle \tag{110}$$

factorizes. Using translational invariance and our result Eq. (106), it follows the final result

$$P_{\text{dec}}^{\checkmark} \leq \left\langle \mathbb{1}_{\mathscr{X}_i^{\checkmark}} \right\rangle^{\frac{L}{2D+1}} \leq \left[ 1 + \left( \frac{p_0^{\times}}{1-p_0^{\times}} \right)^{2D+1} \right]^{-\frac{L}{2D+1}}. \tag{111}$$

Note that this result is generic and we used only that the decoder (1) has only access to the syndrome $\partial\mathbf{x}$ which is invariant under complementation of error patterns and (2) the correction of site $i$ only depends on nearby syndromes in the neighborhood $\pi_i^D \mathbf{x}$.

## B.2 Scaling behavior

Here we derive some scaling limits of Eq. (111). To this end, we assume $D = D(L)$ to be a function of the linear size $L$ of the code.

There are three major cases:

- $D = \text{const.}$ This describes a truly one-dimensional feed-forward circuit of finite depth $D$. We find in the thermodynamic limit

$$\lim_{L \to \infty} P_{\text{dec}}^{\checkmark} \leq \lim_{L \to \infty} \left[ 1 + \left( \frac{p_0^{\times}}{1-p_0^{\times}} \right)^{2D+1} \right]^{-\frac{L}{2D+1}} = \begin{cases} 0 & \text{for} \quad 0 < p_0^{\times} \leq \frac{1}{2} \\ 1 & \text{for} \quad p_0^{\times} = 0 \end{cases}, \tag{112}$$

  i.e., there is no successful decoding possible for any finite microscopic error rate $p_0^{\times} > 0$.

- $2D + 1 \sim L^{\kappa}$ ($\kappa > 0$). This describes a truly two-dimensional feed-forward circuit, possibly slowly growing in the second dimension if $\kappa \approx 0$. We find in the thermodynamic limit

$$\lim_{L \to \infty} P_{\text{dec}}^{\checkmark} \leq \lim_{L \to \infty} \left[ 1 + \left( \frac{p_0^{\times}}{1-p_0^{\times}} \right)^{L^{\kappa}} \right]^{-L^{1-\kappa}} = \begin{cases} 1 & \text{for} \quad 0 \leq p_0^{\times} < \frac{1}{2} \\ 0 & \text{for} \quad p_0^{\times} = \frac{1}{2} \text{ and } \kappa < 1 \\ \frac{1}{2} & \text{for} \quad p_0^{\times} = \frac{1}{2} \text{ and } \kappa = 1 \\ 1 & \text{for} \quad p_0^{\times} = \frac{1}{2} \text{ and } \kappa > 1 \end{cases}, \tag{113}$$

  i.e., except for the critical point $p_0^{\times} = \frac{1}{2}$, there is no constraint on $P_{\text{dec}}^{\checkmark}$ coming from Eq. (111). At the critical point, the upper bounds depend on whether the second dimension scales slower or faster than the length of the chain. For faster scaling depth, there is no constraint, whereas for slower scaling depth, non-trivial upper bounds arise. Note that $P_{\text{dec}}^{\checkmark} \geq \frac{1}{2}$ follows for $p_0^{\times} = \frac{1}{2}$ since a completely mixing Bernoulli process destroys all encoded information about the majority. $P_{\text{dec}}^{\checkmark} > \frac{1}{2}$ arises whenever the decoder fails to get rid of all syndromes. $P_{\text{dec}}^{\checkmark} = \frac{1}{2}$ *can* be realized if the decoder succeeds in removing all syndromes but still fails to recover the original state in 50% of the cases.

To prove the result for $0 \leq p_0^{\text{✗}} < \frac{1}{2}$, we write $p_0^{\text{✗}}/(1-p_0^{\text{✗}}) = q$ with $0 \leq q < 1$. First, note that

$$\lim_{L \to \infty} \left[1 + q^{L^{\kappa}}\right]^{-L^{1-\kappa}} \leq \lim_{L \to \infty} 1^{-L^{1-\kappa}} = 1 \tag{114}$$

because of $q \geq 0$. Furthermore it is $-L^{\kappa} \log q \geq \log L$ for $q < 1$, $\kappa > 0$ and $L$ large enough. This allows us to estimate

$$\left[1 + q^{L^{\kappa}}\right]^{L} = \left[1 + \left(\frac{1}{e}\right)^{-L^{\kappa}\log q}\right]^{L} \leq \left[1 + \left(\frac{1}{e}\right)^{\log L}\right]^{L} = \left[1 + \frac{1}{L}\right]^{L} \leq C \tag{115}$$

for some constant $C > 0$ and where we used that $1/e < 1$ and $\lim_{L \to \infty}(1 + 1/L)^L = e$. With this result, we can find a lower bound as follows:

$$\lim_{L \to \infty} \left[1 + q^{L^{\kappa}}\right]^{-L^{1-\kappa}} \geq \lim_{L \to \infty} \frac{1}{C^{\frac{1}{L^{\kappa}}}} = 1 \tag{116}$$

for $\kappa > 0$. In conclusion, we have shown $\lim_{L \to \infty}\left[1 + q^{L^{\kappa}}\right]^{-L^{1-\kappa}} = 1$ for $q < 1$ and $\kappa > 0$.

- $2D + 1 \sim \log L^{\kappa}$ ($\kappa > 0$). This describes still a two-dimensional feed-forward circuit, but with an exponentially smaller second dimension. In a certain sense, it interpolates between the one- and two-dimensional cases above. Indeed,

$$\lim_{L \to \infty} P_{\text{dec}}^{\checkmark} \leq \lim_{L \to \infty} \left[1 + \left(\frac{p_0^{\text{✗}}}{1-p_0^{\text{✗}}}\right)^{\kappa \log L}\right]^{-\frac{L}{\kappa \log L}} = \begin{cases} 1 & \text{for} \quad p_0^{\text{✗}} \leq p_c^{\text{✗}} \\ 0 & \text{for} \quad p_0^{\text{✗}} > p_c^{\text{✗}} \end{cases}, \tag{117}$$

where the critical microscopic error rate is

$$p_c^{\text{✗}} = \frac{1}{1 + e^{1/\kappa}}. \tag{118}$$

To show this, we write

$$\left[1 + q^{\kappa \log L}\right]^{L} = \left[1 + \frac{1}{L^{-\kappa \log q}}\right]^{L} = \left[1 + \frac{1}{L^{\eta}}\right]^{L}, \tag{119}$$

with $q = p_0^{\text{✗}}/(1-p_0^{\text{✗}})$ and $\eta = -\kappa \log q$. Using $\lim_{x \to 0} \log(1+x)/x = \lim_{x \to 0} 1/(1+x) = 1$, we find

$$\lim_{L \to \infty} \log\left(1 + L^{-\eta}\right)/L^{-\eta} = 1 \tag{120}$$

for $\eta > 0$. Hence

$$\lim_{L \to \infty} \left[1 + q^{\kappa \log L}\right]^{-\frac{L}{\kappa \log L}} = \lim_{L \to \infty} \exp\left[-\frac{L^{1-\eta}}{\kappa \log L} \cdot \log\left(1 + L^{-\eta}\right)/L^{-\eta}\right] \tag{121a}$$

$$= \begin{cases} 0 & \text{for} \quad \eta < 1 \\ 1 & \text{for} \quad \eta \geq 1 \end{cases}. \tag{121b}$$

The critical value $\eta_c = 1$ corresponds to $-\kappa \log q_c = 1 \Leftrightarrow q_c = e^{-1/\kappa}$ and therefore $p_c^{\text{✗}} = 1/(1 + e^{1/\kappa})$.

Whereas constant depth allows for no correction if $p_0^{\text{✗}} > 0$ and algebraically growing $D$, in principle, imposes no restriction at all (except for $p_0^{\text{✗}} = \frac{1}{2}$ of course), a logarithmically growing depth could still be sufficient for low enough error rates $p_0^{\text{✗}} \leq p_c^{\text{✗}} < \frac{1}{2}$.

# C   Linear eroders on finite chains

Regarding the eroder property of a *finite* chain, we have to relax the definition to account for the finiteness of the system as there is no qualitative difference between perturbation and background (both of which are finite). A possible modification reads as follows:

**Definition 3.** *A cellular automaton on a finite chain $\mathscr{L} = \{1, \ldots, L\}$ (with arbitrary boundary conditions) is a finite-size linear eroder if there exist real constants $0 < a < 1$ and $m \in \mathbb{R}^+$ such that for any size $L < \infty$ and any finite perturbation of $\mathbf{0}$ ($\mathbf{1}$) with diameter $l \leq a\,L$, the unperturbed state $\mathbf{0}$ ($\mathbf{1}$) is recovered at $t_{dec} \leq m\,l$.*

Here we focus on $\overline{\text{TLV}}$. Clearly, a contiguous cluster of errors touching the mirror is eventually eroded by the modified TLV rules if it is small enough (so that no signal reaches the opposite boundary before dissolving). Since the majority function is monotonic (changing an input bit $0 \to 1$ never changes the output bit from $1 \to 0$), the evolution of $\overline{\text{TLV}}$ from a non-contiguous, finite cluster of errors can be constructed from the evolution of a contiguous cluster of the same size (its convex hull) by erasing errors in the spacetime diagram. This implies that $\overline{\text{TLV}}$ is an eroder in the above sense.

We can make this statement more rigorous [see Fig. 12 (a)]: It is straightforward to verify that a finite cluster $I$ of diameter $\|I\| = l$ (without loss of generality contiguous, due to monotonicity) is eroded by TLV in a spacetime rectangle of dimensions $[(2Rm+1)l] \times (m\,l)$ for appropriately chosen $m \in \mathbb{R}^+$ (for TLV it is $m = 1$ and $R = 4$, see Appendix F). This holds also for $\overline{\text{TLV}}$ if $I$ is separated from the edges by more than $\delta(l) \equiv Rml$ sites since $(2Rm+1)l = l + 2\delta(l)$ guarantees that the cluster is eroded before the boundaries can have any effect, see Fig. 12 (a-1). Necessary for this situation is

$$(2Rm+1)l \leq L \iff l \leq \frac{L}{2Rm+1}\,. \tag{122}$$

If, on the other hand, $I$ is *closer* than $\delta(l)$ to one of the edges, we can no longer guarantee that it can be eroded in the neighborhood given by $\delta(l)$ due to possible interactions with its mirror image. We define a padded interval $I' \supseteq I$ of length $l \leq l' < l + \delta(l) = (Rm+1)l$ that closes the gap between $I$ and the critical edge. Now we know that this interval is eroded in a spacetime box of dimensions $[(2Rm+1)2l'] \times (m\,2l')$ due to the mirror. In the "real" chain, this accounts for an interval of length $l' + \delta(2l') = (2Rm+1)l'$ adjacent to the corresponding edge. If the latter does *not* make contact with the opposite edge, the original cluster $I$ is guaranteed to be eroded, see Fig. 12 (a-2). We have the sufficient condition

$$(2Rm+1)l' \leq L \iff l' \leq \frac{L}{2Rm+1}\,. \tag{123}$$

If we require instead

$$l + \delta(l) \leq \frac{L}{2Rm+1} \iff l \leq \frac{L}{(2Rm+1)(Rm+1)}\,, \tag{124}$$

this implies Eq. (123) for all critical lengths $l' < l + \delta(l)$ and Eq. (122) trivially. We conclude that with $a \equiv (2Rm+1)^{-1}(Rm+1)^{-1}$ any cluster of diameter $l \leq aL$ is eroded in finite time (linear in $l$). For $\overline{\text{TLV}}$ we find $a = 1/45 \approx 0.02$ (which is an extremely conservative lower bound; $\overline{\text{TLV}}$ allows for much larger values of $a$ as simulations suggest).

Note that if the interval $l' + \delta(2l')$ is larger than the system [Fig. 12 (a-3)], it is possible that the cluster relaxes into non-homogeneous fixed points or non-trivial cycles. This is a consequence of the two mirrors which allow for the periodic reflection of messages in this "CA cavity".

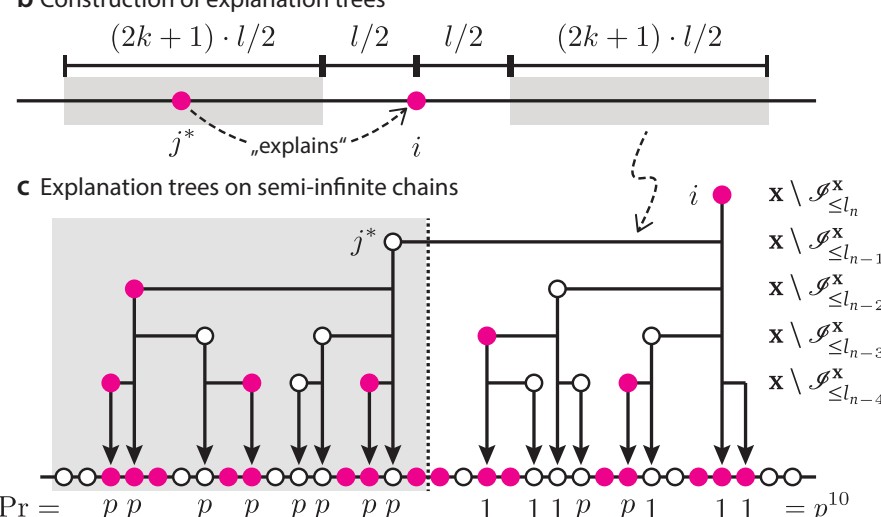

Figure 12: *Proofs.* **a** Eroder property for a finite system with mirrored boundary conditions. Errors in the cluster $I$ are marked red. Without loss of generality, $I$ can be made contiguous by padding holes with additional errors (black). An explanation is given in Appendix C. **b** Illustration of the implication in Eq. (131b): An element $i \in \mathbf{x} \setminus \mathscr{I}^{\mathbf{x}}_{\leq l}$ requires the existence of another element $j^* \in \mathbf{x} \setminus \mathscr{I}^{\mathbf{x}}_{\leq l-1}$ in a specific range given by $k$ and $l$. **c** Explanation tree on the semi-infinite chain with mirrored boundary condition. Cells in state 0 (1) are marked white (red). The probability to realize the shown explanation tree by a mirrored Bernoulli process with rate $p = p_0^{\mathbf{x}}$ is labeled by Pr. Note that the shown error pattern (red) fails to realize this explanation tree (arrows) since some explanatory sites are empty. Details are given in Appendix E.

# D   Fixed points

In addition to the two homogeneous fixed points (which are stable due to the eroder property), TLV-based automata can feature up to 4 unstable fixed points, depending on the boundary conditions imposed. As shown rigorously in Appendix E, these are no threat to the decoding capabilities because their occurrence in a Bernoulli random process is exponentially suppressed. For the sake of completeness, we discuss them in the following:

- *Infinite chain.* The original TLV features six fixed points [35]: The two homogeneous (stable) ones and, in addition, the four periodic (unstable) configurations shown in Fig. 13 (b).

- *Semi-infinite chain with mirrored boundary.* A modified $\overrightarrow{\text{TLV}}$ with a single mirrored boundary features 2 of the 4 unstable fixed points of the infinite chain. Note that the first two patterns in Fig. 13 (b) are not bond-inversion symmetric and therefore cannot be interpreted as a valid configuration on the semi-infinite chain. However, the latter two are bond-inversion symmetric if the mirror is placed such that the first cell is even. In contrast, if the cell next to the mirror is odd [lower two patterns in Fig. 13 (c)], there are no additional fixed points.

- *Finite chain with periodic boundaries.* If TLV is placed on a closed ring of length $L \in 2\mathbb{N}$, potential fixed points can be used to construct periodic ones on the infinite chain. Since there are only the four depicted in Fig. 13 (b), we have to check which of those remains invariant under periodic boundary conditions. As illustrated in Fig. 13 (d), if $L$ is a multiple of 4, all four fixed points in (b) can be transfered to the finite chain with PBCs. However, if $L \notin 4\mathbb{N}$, the two 4-periodic patterns are no longer invariant and only the two 2-periodic patterns survive (compare the yellow patterns on the left with the colored patterns on the right).

- *Finite chain and mirrored boundaries.* If TLV is placed on a chain of length $L \in 2\mathbb{N}$ with mirrored boundaries, we can infer from the semi-infinite case in Fig. 13 (c) that only if the first (left) cell is even and the last (right) cell is odd, two additional fixed points survive. Otherwise the homogeneous configurations are the only ones, Fig. 13 (e). This is the modification $\overline{\text{TLV}}$ we use in this paper.

None of the additional fixed points are relevant for the correction of Bernoulli random patterns because the probability of their occurrence is exponentially suppressed with $L$. This follows directly from the fact that there are no non-trivial preimages of these fixed points, i.e., their attractors are trivial. The only way to end up in one of them is that the noise gives rise to its pattern by chance. We checked this for finite chains of $\overline{\text{TLV}}$ by solving the corresponding systems of boolean equations to determine all fixed points and their preimages.

# E   Sparse errors and correction time

Here we prove a central statement of this work: The probability for a chain of length $L$, ruled by $\overline{\text{TLV}}$ with MBC to be in a non-empty state $\mathbf{x}(t) \neq \mathbf{0}$ after $t \propto L^\kappa$ time steps vanishes exponentially with $L$ for arbitrary $\kappa > 0$ if the initial state is a Bernoulli random configuration with single-site error probability $p_0^\mathsf{x} < p_c^\mathsf{x}$ for some critical value $0 < p_c^\mathsf{x} \leq \frac{1}{2}$.

For convenience, we reproduce the definitions from the main text: Let $\mathbf{x} \subseteq \mathbb{Z}$ be an arbitrary subset (error pattern). A finite subset $I \subseteq \mathbf{x}$ is called *cluster* of diameter $\|I\| = \max\{|x - y| \,|\, x, y \in I\}$. If we fix an integer $k > 0$ (the *sparseness parameter*, to be

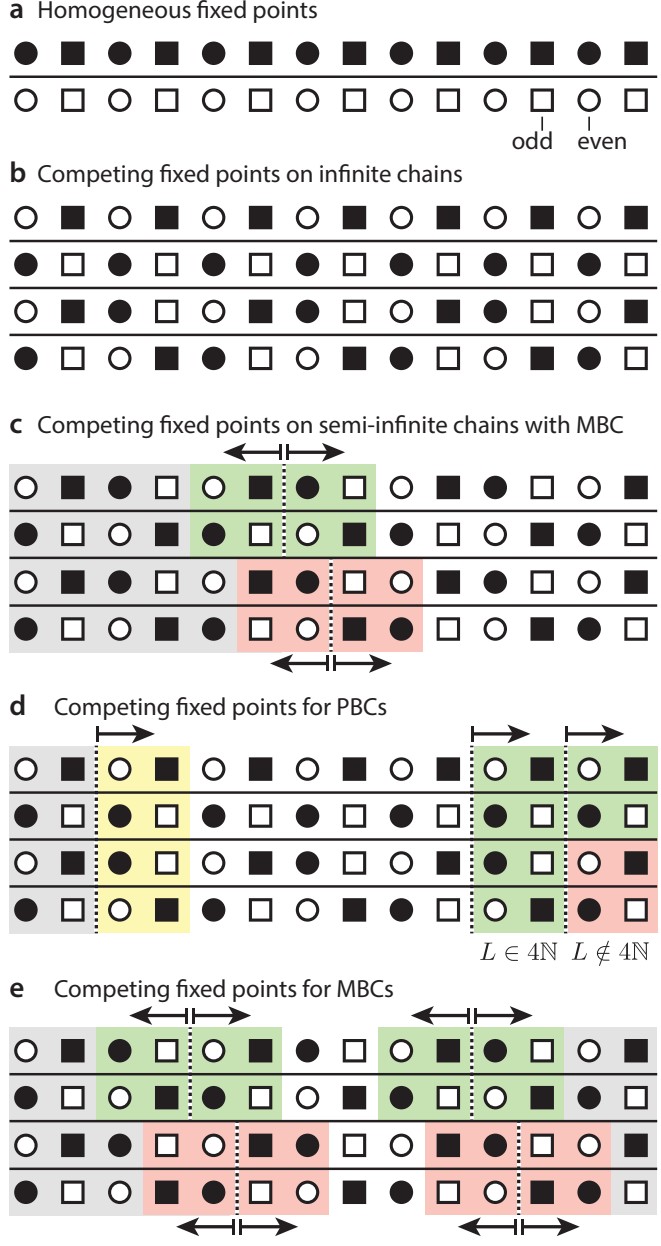

Figure 13: *Fixed points.* **a** The homogeneous fixed points are always present and the only stable ones (due to the eroder property). Competing (unstable) fixed points are possible but depend on the boundary conditions: **b** Infinite chain (4 additional fixed points). **c** Semi-infinite chain with mirrored boundary condition (2 additional fixed points if the first cell is even, none otherwise). **d** Periodic boundary conditions (4 additional fixed points if $L$ is a multiple of 4, 2 otherwise) **e** Mirrored boundary conditions (2 additional fixed points if the first cell is even and the last is odd, none otherwise).

chosen later), the *territory* $T_k(I)$ is defined as the interval of integers with distance at most $k\|I\|$ from $I$. Two clusters $I_1$ and $I_2$ are called *independent* if at least one does not intersect the territory of the other, i.e., $I_2 \cap T_k(I_1) = \emptyset$ or $I_1 \cap T_k(I_2) = \emptyset$ (or both); since $I \subset T_k(I)$, this implies $I_1 \cap I_2 = \emptyset$. If there exists a partition of $\mathbf{x}$ into a family $\mathscr{I} = \{I_a\}$ of pairwise independent clusters $I_a$, $\mathbf{x} = \bigcup_a I_a$, then $\mathbf{x}$ is called *sparse*. A cluster $I \subseteq \mathbf{x}$ with $T_k(I) \cap \mathbf{x} = I$ is called *independent in* $\mathbf{x}$ and we write $I \sqsubseteq \mathbf{x}$.

To state our first result, we need some additional terminology: First, $\mathscr{I}_{\leq l}$ denotes the family of clusters $I \in \mathscr{I}$ with diameter $\|I\| \leq l$ and $\mathbf{x} \setminus \mathscr{I}_{\leq l} \equiv \mathbf{x} \setminus \bigcup_{I \in \mathscr{I}_{\leq l}} I$ is the subset of sites for given $\mathbf{x}$ that remains after cleaning all clusters of diameter at most $l$. Second, a *(infinite) mirrored Bernoulli random configuration* $\mathbf{x} \subseteq \mathbb{Z}$ is defined by the single-site probability $\Pr(x_i = 1) = p_0^{\mathbf{x}}$ for sites $i > 0$ and the mirror constraint $x_i = x_{1-i}$.

We can now state our main result (an adaptation of Theorem 4 in Ref. [64] for mirrored Bernoulli random configurations):

**Proposition 1.** *Consider **infinite mirrored** Bernoulli random configurations* $\mathbf{x}$ *with single-site probability* $p_0^{\mathbf{x}}$. *Let* $k \in \mathbb{N}$ *be a given sparseness parameter.*

*Then, for each instance* $\mathbf{x}$*, there exists a constructive family* $\mathscr{I}^{\mathbf{x}}$ *of pairwise independent clusters (it is not necessarily* $\mathbf{x} = \bigcup_{I \in \mathscr{I}^{\mathbf{x}}} I$*, i.e.,* $\mathbf{x}$ *does not have to be sparse) such that the probability of a site* $i \in \mathbb{Z}$ *to be in* $\mathbf{x}$ *and remain uncovered by independent clusters of diameter $l$ or less (write* $\mathscr{I}_{\leq l}^{\mathbf{x}}$*) is upper-bounded by*

$$\Pr\left(i \in \mathbf{x} \setminus \mathscr{I}_{\leq l}^{\mathbf{x}}\right) \leq \alpha^{l^{\beta}} \tag{125}$$

*for* $\beta = \ln(2)/\ln(4k+3)$ *(and therefore* $0 < \beta < 1$*) and* $\alpha = (2k)(4k+3)\sqrt{p_0^{\mathbf{x}}}$. *If we define the critical value*

$$\tilde{p}_c^{\mathbf{x}} \equiv [(2k)(4k+3)]^{-2}, \tag{126}$$

*for* $p_0^{\mathbf{x}} < \tilde{p}_c^{\mathbf{x}}$ *it is* $\alpha < 1$ *and Eq. (125) becomes an exponentially decaying upper bound.*

*Proof.* Let $\mathbf{x} \subseteq \mathbb{Z}$ be an arbitrary mirrored configuration. Fix the sparseness parameter $k \in \mathbb{N}$. We proceed stepwise:

1. *Construction of* $\mathscr{I}^{\mathbf{x}}$.

   Ignoring the inversion symmetry of $\mathbf{x}$, we construct the family of cluster $\mathscr{I}^{\mathbf{x}}$ recursively: Set $\mathscr{I}_0^{\mathbf{x}} := \emptyset$ and $\mathscr{I}_{<l} := \bigcup_{0 \leq k < l} \mathscr{I}_k^{\mathbf{x}}$. Define $\mathscr{I}_l^{\mathbf{x}}$ ($l > 0$) as the family of all independent clusters $I \sqsubseteq (\mathbf{x} \setminus \mathscr{I}_{<l}^{\mathbf{x}})$ of diameter $\|I\| = l$; i.e., $I$ consists of errors in $\mathbf{x}$ that do not belong to already defined smaller clusters and $I$ is independent in $\mathbf{x}$ after deleting all these smaller clusters. The construction of $\mathscr{I}_l^{\mathbf{x}}$ is well-defined because two different clusters $I_1$ and $I_2$ of diameter $l$ cannot intersect due to $T_k(I_a) \cap (\mathbf{x} \setminus \mathscr{I}_{<l}^{\mathbf{x}}) = I_a$, $a = 1, 2$ (note that a cluster $I_a$ can be thought of as an interval of length $l$ with "holes" away from the edges). Then we define $\mathscr{I}^{\mathbf{x}} := \bigcup_l \mathscr{I}_l^{\mathbf{x}}$.

2. *Independence of* $\mathscr{I}^{\mathbf{x}}$.

   We show that distinct clusters in $\mathscr{I}^{\mathbf{x}}$ are pairwise independent by construction.

   Since being independent is an *asymmetric* relation between two clusters of unequal diameter (if $\|I_1\| \geq \|I_2\|$, then $I_2 \cap T_k(I_1) = \emptyset \Rightarrow I_1 \cap T_k(I_2) = \emptyset$ and it is enough to check for $I_1 \cap T_k(I_2) = \emptyset$), it is sufficient to check that (1) all equal-sized clusters in $\mathscr{I}_l^{\mathbf{x}}$ are pairwise independent and (2) they do not intersect the territories of the smaller clusters in $\mathscr{I}_{<l}^{\mathbf{x}}$.

   (1) follows immediately from the well-defined construction of $\mathscr{I}_l^{\mathbf{x}}$ (see previous paragraph).

(2) follows because $I \in \mathscr{I}_l^{\mathbf{x}}$ belongs to $\mathbf{x} \setminus \mathscr{I}_{<l'}^{\mathbf{x}}$ for all $l' < l$ and hence $I' \cap I = \emptyset$ for all $I' \in \mathscr{I}_{<l}^{\mathbf{x}}$. Since $I'$ was chosen independent from all other elements in $\mathbf{x} \setminus \mathscr{I}_{<l'}^{\mathbf{x}}$, it follows that $T_k(I') \cap I = \emptyset$, i.e., $I'$ and $I$ are independent.

3. *Completeness of $\mathscr{I}^{\mathbf{x}}$.*

   Here we show that the constructed $\mathscr{I}^{\mathbf{x}}$ is complete in the sense that

   $$i \in \mathbf{x} \setminus \mathscr{I}_{\leq l}^{\mathbf{x}} \Rightarrow \forall_{I \sqsubseteq (\mathbf{x} \setminus \mathscr{I}_{<l}^{\mathbf{x}}), \|I\| \leq l} : i \notin I \tag{127}$$

   which we will use in the step 4 below.

   Eq. (127) is a bit subtle because only

   $$i \in \mathbf{x} \setminus \mathscr{I}_{\leq l}^{\mathbf{x}} \Rightarrow \forall_{I \sqsubseteq (\mathbf{x} \setminus \mathscr{I}_{<l}^{\mathbf{x}}), \|I\| = l} : i \notin I \tag{128}$$

   follows trivially from the construction of $\mathscr{I}_l^{\mathbf{x}}$ (see step 1 above). To prove Eq. (127), we show that $\forall_{I \sqsubseteq (\mathbf{x} \setminus \mathscr{I}_{<l}^{\mathbf{x}})} : \|I\| \geq l$, i.e., our construction never recreates clusters of smaller diameter (which could, in principle, happen because we are successively deleting clusters):

   Assume $\exists_{I^* \sqsubseteq (\mathbf{x} \setminus \mathscr{I}_{<l}^{\mathbf{x}})} : \|I^*\| = l^* < l$. It must have been $T_k(I^*) \cap (\mathbf{x} \setminus \mathscr{I}_{<l^*}^{\mathbf{x}}) \supset I^*$ because otherwise our prescription demands $I^* \in \mathscr{I}_{l^*}^{\mathbf{x}}$ and we had $I^* \cap (\mathbf{x} \setminus \mathscr{I}_{<l}^{\mathbf{x}}) = \emptyset$. Since $T_k(I^*) \cap (\mathbf{x} \setminus \mathscr{I}_{<l}^{\mathbf{x}}) = I^*$ by assumption, there must have been a cluster $\tilde{I} \in \mathscr{I}_{\tilde{l}}^{\mathbf{x}}$ with $l^* \leq \tilde{l} < l$ and $T_k(I^*) \cap \tilde{I} \neq \emptyset$. But because $\|\tilde{I}\| \geq \|I^*\|$, this implies $T_k(\tilde{I}) \cap I^* \neq \emptyset$. Since $I^* \not\subseteq \tilde{I}$ and $I^* \subseteq \mathbf{x} \setminus \mathscr{I}_{<\tilde{l}}^{\mathbf{x}}$, this contradicts the independence of $\tilde{I}$ in $\mathbf{x} \setminus \mathscr{I}_{<\tilde{l}}^{\mathbf{x}}$ and we are done.

4. *Explanation trees.*

   Clearly $(\mathbf{x} \setminus \mathscr{I}_{\leq l}^{\mathbf{x}}) \subseteq (\mathbf{x} \setminus \mathscr{I}_{<l}^{\mathbf{x}})$, i.e., fewer and fewer errors in $\mathbf{x}$ survive with increasing $l$ because on each level additional clusters are deleted from $\mathbf{x}$. This monotonicity holds also for all monotonic sequences $(l_n) \in \mathbb{N}^{\mathbb{N}}$ with $l_n > l_{n-1}$ for all $n \in \mathbb{N}$: $(\mathbf{x} \setminus \mathscr{I}_{\leq l_n}^{\mathbf{x}}) \subseteq (\mathbf{x} \setminus \mathscr{I}_{\leq l_{n-1}}^{\mathbf{x}})$. In the end, we aim to upper-bound the probability of an arbitrary site $i \in \mathbb{Z}$ to belong to $\mathbf{x} \setminus \mathscr{I}_{\leq l}^{\mathbf{x}}$. We first prove this for $\mathbf{x} \setminus \mathscr{I}_{\leq l_n}^{\mathbf{x}}$ instead, where $(l_n)$ will be specified below, and generalize our result (with some tradeoff) to $l_n = n$ in the next (and last) step 5.

   For the sake of simplicity, let $l_n$ be an odd integer for $n \geq 1$ and define $l_0 \equiv 0$ in the following. To bound the probability for $i \in \mathbf{x} \setminus \mathscr{I}_{\leq l_n}^{\mathbf{x}}$ from above, we start with a trivially true, *sufficient* condition: For an arbitrary configuration $\mathbf{y}$ with $i \in \mathbf{y}$, we have ($n \geq 1$)

   $$\forall_{j \in \mathbf{y}} : |i - j| \leq \frac{l_n}{2} \ \vee \ |i - j| > \left( k + \frac{1}{2} \right) l_n$$
   $$\Rightarrow \exists_{I^* \sqsubseteq \mathbf{y}, \|I^*\| \leq l_n} : i \in I^* \tag{129}$$

   ($I^*$ includes all $j \in \mathbf{y}$ with $|i - j| \leq l_n/2$; the strict ">" becomes important only for *even* $l_n$).

   The (equivalent) contraposition reads

   $$\forall_{I \sqsubseteq \mathbf{y}, \|I\| \leq l_n} : i \notin I$$
   $$\Rightarrow \exists_{j^* \in \mathbf{y}} : \frac{l_n}{2} < |i - j^*| \leq \left( k + \frac{1}{2} \right) l_n . \tag{130}$$

If we now set $\mathbf{y} = \mathbf{x} \setminus \mathscr{I}^{\mathbf{x}}_{\leq l_n - 1} = \mathbf{x} \setminus \mathscr{I}^{\mathbf{x}}_{< l_n}$ and use Eq. (127) with $l = l_n$ (this is the crucial step that exploits the structure of $\mathscr{I}^{\mathbf{x}}$), we end up with the sequence of implications

$$i \in \mathbf{x} \setminus \mathscr{I}^{\mathbf{x}}_{\leq l_n}$$

$$\Rightarrow \ \forall_{I \sqsubseteq (\mathbf{x} \setminus \mathscr{I}^{\mathbf{x}}_{< l_n}), \|I\| \leq l_n} \ : \ i \notin I \tag{131a}$$

$$\Rightarrow \ \exists_{j^* \in \mathbf{x} \setminus \mathscr{I}^{\mathbf{x}}_{\leq l_n - 1}} \ : \ \frac{l_n}{2} < |i - j^*| \leq \left( k + \frac{1}{2} \right) l_n \tag{131b}$$

$$\Rightarrow \ \exists_{j^* \in \mathbf{x} \setminus \mathscr{I}^{\mathbf{x}}_{\leq l_{n-1}}} \ : \ \frac{l_n}{2} < |i - j^*| \leq \left( k + \frac{1}{2} \right) l_n \,, \tag{131c}$$

where we used $\mathbf{x} \setminus \mathscr{I}^{\mathbf{x}}_{\leq l_n - 1} \subseteq \mathbf{x} \setminus \mathscr{I}^{\mathbf{x}}_{\leq l_{n-1}}$ (since $l_{n-1} \leq l_n - 1$) in the last line; this is illustrated in Fig. 12 (b). In combination with $i \in \mathbf{x} \setminus \mathscr{I}^{\mathbf{x}}_{\leq l_n} \Rightarrow i \in \mathbf{x} \setminus \mathscr{I}^{\mathbf{x}}_{\leq l_{n-1}}$, Eq. (131c) gives rise to a binary tree of depth $n$ (with sites as vertices) that explains the existence of $i \in \mathbf{x} \setminus \mathscr{I}^{\mathbf{x}}_{\leq l_n}$ at its root if the sites at all its leafs belong to $\mathbf{x} \setminus \mathscr{I}^{\mathbf{x}}_{\leq 0} = \mathbf{x}$; it is aptly called *explanation tree* (ET) [65]. An example is shown in Fig. 12 (c).

By counting possible ETs and calculating their probability of being realized based on the (mirrored) Bernoulli distribution on their leafs, it is possible to upper bound the probability for $i \in \mathbf{x} \setminus \mathscr{I}^{\mathbf{x}}_{\leq l_n}$ because the existence of at least one realized explanation tree is a *necessary* condition. Counting explanation trees and computing their probability (by counting their leafs) is complicated by the fact that for arbitrary $l_n > l_{n-1} > l_{n-2} > \cdots > l_0$ the allowed ranges for $j^*$ on different levels $n$ intersect. Therefore the number of leafs is not fixed and only upper-bounded by $2^n$ (reducing the number of leafs can be achieved by "reusing" a site to explain more than one other site). This complicates the derivation of the probability for the existence of a realized explanation tree considerably. If, in contrast, $(l_n)$ is chosen so that different subtrees cannot intersect, the number of leafs for any ET is fixed at $2^n$. This can be guaranteed if on each level $1 \leq m \leq n$ the distance between any site $i$ and its explanatory site $j^*$ is larger than the maximum width of the subtrees emanating from each of them. Formally,

$$\frac{l_m}{2} \geq 2 \left( k + \frac{1}{2} \right) \sum_{k=1}^{m-1} l_k \,, \tag{132}$$

where the factor of 2 is necessary because *two* subtrees (one at $i$ and one at $j^*$) grow independently. Equating both sides yields the tightest solution via the recursion

$$l_{m+1} = 4 \left( k + \frac{1}{2} \right) \sum_{k=1}^{m} l_k = 4 \left( k + \frac{1}{2} \right) l_m + l_m = (4k + 3) l_m \tag{133}$$

which is solved by

$$l_m = (4k + 3)^{m-1} \tag{134}$$

with $l_1 = 1$.

Let $N_n$ denote the number of possible ETs of depth $n$ (i.e., the number of ETs that can explain $i \in \mathbf{x} \setminus \mathscr{I}^{\mathbf{x}}_{\leq l_n}$, all of which are *realized* in the completely filled state $\mathbf{x} = \mathbf{1}$). It holds recursively that $N_m \leq 2k l_m N_{m-1}^2$ for all $1 \leq m \leq n$ (for non-overlapping trees this is an equality). One factor, $N_{m-1}$, counts the possible subtrees attached to $i$ while another factor, $2k l_m N_{m-1}$, counts the possible positions for the new root $j^*$ [$2k l_m$, see Eq. (131c)] and the possible subtrees attached to this root ($N_{m-1}$). Clearly

$$\log N_m \leq a (m - 1) + b + 2 \log N_{m-1} \tag{135}$$

with $a = \log(4k + 3) > 0$ and $b = \log(2k) > 0$ for our specific choice of $l_m$ in Eq. (134). An upper bound on $\log N_m$ can be found by solving the corresponding *equality* which corresponds to the affine recursion of two sequences

$$\begin{bmatrix} g_m \\ f_m \end{bmatrix} = \begin{bmatrix} 2 & a \\ 0 & 1 \end{bmatrix} \cdot \begin{bmatrix} g_{m-1} \\ f_{m-1} \end{bmatrix} + \begin{bmatrix} b \\ 1 \end{bmatrix} \tag{136}$$

with $g_m = \log N_m$ and $f_m = m$ and initial conditions $g_0 = 0 = f_0$ ($N_0 = 1$). Diagonalization of the matrix yields

$$\begin{bmatrix} \chi_m \\ \varphi_m \end{bmatrix} = \begin{bmatrix} 2 & 0 \\ 0 & 1 \end{bmatrix} \cdot \begin{bmatrix} \chi_{m-1} \\ \varphi_{m-1} \end{bmatrix} + \begin{bmatrix} a + b \\ \sqrt{1 + a^2} \end{bmatrix} \tag{137}$$

with $\chi_m = a f_m + g_m$ and $\varphi_m = \sqrt{1 + a^2} f_m$. Now we can use that recursions of the form $X_n = A X_{n-1} + B$ are generically solved by

$$X_n = X_0 A^n + B \frac{1 - A^n}{1 - A} \quad \text{for} \quad A \neq 1 \tag{138}$$

and $X_n = X_0 + B n$ for $A = 1$. With the initial conditions, we find immediately $\varphi_m = \sqrt{1 + a^2}\, m$ and $\chi_m = (2^m - 1)(a + b)$. Transforming $\chi_m$ and $\varphi_m$ back into $g_m$ and $f_m$ yields the result $f_m = m$ and

$$g_m = (2^m - 1)(a + b) - a m \tag{139}$$

which is the solution of the recursive *equality* in Eq. (135). It is automatically an upper bound, thus

$$N_m \leq \exp\left[(2^m - 1)(a + b) - a m\right] \leq \exp\left[2^m (a + b)\right] \tag{140}$$

and we find $N_m \leq [(2k)(4k + 3)]^{2^m}$ with $e^{a+b} = (2k)(4k + 3)$.

Now comes the only step where we use the mirror symmetry of the Bernoulli configuration $\mathbf{x}$: The probability for all $2^n$ leafs of a particular ET to be occupied is $(p_0^{\mathbf{x}})^{2^n}$ for sites that are *independent* Bernoulli random variables. The mirror symmetry, however, introduces perfect correlations between pairs of sites. Since all leafs are distinct sites (on $\mathbb{Z}$) there are at least $2^n/2$ independent Bernoulli random variables associated to an ET (the worst case being a completely mirror-symmetric explanation tree). Therefore the probability for an arbitrary ET to be realized is upper-bounded by $\sqrt{p_0^{\mathbf{x}}}^{2^n}$ (as compared to $(p_0^{\mathbf{x}})^{2^n}$ in systems without mirror symmetry). This reflects the fact that mirrors "enlarge" error clusters artificially by their mirror images. This is illustrated in Fig. 12 (c).

In conclusion, we find an upper bound

$$\Pr\left(i \in \mathbf{x} \setminus \mathscr{I}_{\leq l_n}^{\mathbf{x}}\right) \leq N_n \sqrt{p_0^{\mathbf{x}}}^{2^n} \leq \left[(2k)(4k + 3)\sqrt{p_0^{\mathbf{x}}}\right]^{2^n} \tag{141}$$

for an arbitrary site $i \in \mathbb{Z}$ to be in $\mathbf{x}$ but uncovered by clusters up to diameter $l_n$ of the constructive family $\mathscr{I}^{\mathbf{x}}$. This follows from the subadditivity of probability measures and the statement that $i \in \mathbf{x} \setminus \mathscr{I}_{\leq l_n}^{\mathbf{x}}$ if there is *at least* one of $N_n$ possible ETs realized by $\mathbf{x}$.

If we define $(2k)(4k + 3)\sqrt{\tilde{p}_c^{\mathbf{x}}} = 1 \Leftrightarrow \tilde{p}_c^{\mathbf{x}} \equiv [(2k)(4k + 3)]^{-2}$, it follows with $\alpha \equiv (2k)(4k + 3)\sqrt{p_0^{\mathbf{x}}}$

$$\Pr\left(i \in \mathbf{x} \setminus \mathscr{I}_{\leq l_n}^{\mathbf{x}}\right) \leq \alpha^{2^n}. \tag{142}$$

For on-site probabilities $p_0^{\mathbf{x}} < \tilde{p}_c^{\mathbf{x}} \Leftrightarrow \alpha < 1$, this leads to a *double-exponential* decay of the probability to remain uncovered on level $l_n$.

5. *Upper bound.*

Above we showed that $\Pr\left(i \in \mathbf{x} \setminus \mathscr{I}^{\mathbf{x}}_{\leq l_n}\right) \leq \alpha^{2^n}$ with $l_n = (4k+3)^{n-1}$. The double-exponential decay of the probability with $n$ and the exponential growth of the level $l_n$ suggest that there is an exponentially decaying upper bound with $l$ (recall that our choice of $l_n$ was technically motivated: it is easier to count the leafs of ETs if the branches do not intersect).

Indeed, if we use the monotonicity $\Pr\left(i \in \mathbf{x} \setminus \mathscr{I}^{\mathbf{x}}_{\leq l}\right) \leq \Pr\left(i \in \mathbf{x} \setminus \mathscr{I}^{\mathbf{x}}_{\leq l-1}\right)$, it follows that

$$\Pr\left(i \in \mathbf{x} \setminus \mathscr{I}^{\mathbf{x}}_{\leq l}\right) \leq \alpha^{l^\beta} \tag{143}$$

if we require $\alpha^{l_n^\beta} \overset{!}{=} \alpha^{2^{n-1}}$ for $\beta > 0$, because for $l \in [l_{n-1}, l_n]$ we know that

$$\Pr\left(i \in \mathbf{x} \setminus \mathscr{I}^{\mathbf{x}}_{\leq l}\right) \leq \alpha^{2^{n-1}} \quad \text{and} \quad \alpha^{2^{n-1}} \leq \alpha^{l^\beta} \tag{144}$$

per construction (because $\alpha < 1$).

This determines $\beta$ via $(4k+3)^{\beta(n-1)} \overset{!}{=} 2^{n-1}$, i.e.,

$$\beta = \frac{\ln(2)}{\ln(4k+3)} < 1. \tag{145}$$

This concludes the proof. ∎

Note that for $p_0^{\mathbf{x}} > \tilde{p}_c^{\mathbf{x}}$ the upper bounds become trivial which still allows for an exponential decay of $\Pr\left(i \in \mathbf{x} \setminus \mathscr{I}^{\mathbf{x}}_{\leq l}\right)$. Therefore we conclude that there is a critical value $p_c^{\mathbf{x}}$ with $0 < \tilde{p}_c^{\mathbf{x}} \leq p_c^{\mathbf{x}}$ such that $\Pr\left(i \in \mathbf{x} \setminus \mathscr{I}^{\mathbf{x}}_{\leq l}\right)$ vanishes exponentially for $l \to \infty$ if $p_0^{\mathbf{x}} < p_c^{\mathbf{x}}$. Simulations suggest that $p_c^{\mathbf{x}} = \frac{1}{2}$ so that $\tilde{p}_c^{\mathbf{x}} \ll 1$ is a rather weak lower bound on the true critical value, see Appendix F.

Eventually we want to employ Prop. 1 to derive an upper bound for the probability of errors to survive the first $t$ steps of $\overline{\text{TLV}}$ on a *finite* chain with mirrored boundaries. To this end, we first need a consequence of Prop. 1:

**Lemma 2.** *Consider a **semi-infinite** chain on $\mathscr{L} = \mathbb{N}$ governed by $\overrightarrow{\text{TLV}}_\mathscr{L}$ with initial configurations $\mathbf{x}(0) \subseteq \mathscr{L}$ drawn from a Bernoulli distribution with parameter $p_0^{\mathbf{x}}$. Let $\mathscr{J} \subset \mathscr{L}$ be an arbitrary finite interval on the chain.*

*Then the probability of $\mathbf{x}(t) = \overrightarrow{\text{TLV}}^t_\mathscr{L}(\mathbf{x}(0))$ to be non-empty on $\mathscr{J}$ is upper-bounded by*

$$\Pr\left(\mathbf{x}(t) \cap \mathscr{J} \neq \emptyset\right) \leq (2tR + |\mathscr{J}|) \exp\left(-\gamma \lfloor t/m \rfloor^\beta\right) \tag{146}$$

*with $\gamma = -\log(\alpha)$ ($\gamma > 0$ for $p_0^{\mathbf{x}} < \tilde{p}_c^{\mathbf{x}}$), and $0 < \beta < 1$ as in Prop. 1. Here the sparseness parameter is given by $k = 2Rm = 8$ where $m = 1$ and $R = 4$ are the eroder parameter and the radius of TLV, respectively.*

*Proof.* Because $\overrightarrow{\text{TLV}}$ is an eroder, there is a constant $m$ such that any cluster of errors $I$ on a background of zeros is erased for $t \geq m\|I\|$. During this process, signals emitted beyond the boundaries of $I$ can at most travel $Rm\|I\|$ sites where $R$ is the radius of the local rules (or the propagation speed of information). If we set $k = 2Rm$ as sparseness parameter, an error cluster $I \sqsubseteq \mathbf{x}(0)$ that is independent in $\mathbf{x}(0)$, is safely erased after at most $m\|I\|$ time steps without interfering with its environment. This follows because signals from $I$ and $\mathbf{x}(0) \setminus I$ can meet only after traversing the void territory $T_k(I) \setminus I$ which takes at least $k\|I\|/(2R) = m\|I\|$ time steps—but the last trace of $I$ is erased after $m\|I\|$ time steps. Therefore the evolution of $\mathbf{x}(t)$

for $t \geq m\|I\|$ is completely independent of the configuration within the boundaries of $I$ (this motivates the notion of *independent* clusters). See Fig. 6 of the main text for an illustration.

If $\mathbf{x}(0)$ is a mirrored Bernoulli random configuration with parameter $p_0^{\mathsf{X}} < \tilde{p}_c^{\mathsf{X}}$ with $k$ set as above, we know from Prop. 1 that the probability of any site $i \in \mathbb{N}$ to be uncovered by clusters in $\mathscr{I}^{\mathbf{x}(0)}$ of diameter at most $l$ is upper-bounded by

$$\Pr\left(i \in \mathbf{x}(0) \setminus \mathscr{I}_{\leq l}^{\mathbf{x}(0)}\right) \leq \alpha^{l^\beta} \tag{147}$$

with $0 < \alpha, \beta < 1$. By subadditivity, an analogous bound holds for any finite subset $J \subset \mathbb{N}$,

$$\Pr\left(J \cap \left(\mathbf{x}(0) \setminus \mathscr{I}_{\leq l}^{\mathbf{x}(0)}\right) \neq \emptyset\right) \leq |J| \alpha^{l^\beta} . \tag{148}$$

Let $U_r(\mathscr{J})$ be the interval of all sites within distance $r \geq 0$ of $\mathscr{J}$ and set $J = U_{tR}(\mathscr{J})$ for time $t \geq 0$. Then

$$\Pr\left(U_{tR}(\mathscr{J}) \cap \left(\mathbf{x}(0) \setminus \mathscr{I}_{\leq l}^{\mathbf{x}(0)}\right) \neq \emptyset\right) \leq (2tR + |\mathscr{J}|) \alpha^{l^\beta} . \tag{149}$$

This holds for all $l \in \mathbb{N}$, especially for $l = \lfloor t/m \rfloor$ ($\lfloor \bullet \rfloor$ is the floor function):

$$\Pr\left(U_{tR}(\mathscr{J}) \cap \left(\mathbf{x}(0) \setminus \mathscr{I}_{\leq \lfloor t/m \rfloor}^{\mathbf{x}(0)}\right) \neq \emptyset\right) \leq (2tR + |\mathscr{J}|) \alpha^{\lfloor t/m \rfloor^\beta} . \tag{150}$$

If we exploit that no signal from outside $U_{tR}(\mathscr{J})$ can reach $\mathscr{J}$ up to time $t$ and that all errors that belong to independent clusters of diameter $l \leq \lfloor t/m \rfloor \leq t/m$ are erased at time $t$, we can conclude that

$$U_{tR}(\mathscr{J}) \cap \left(\mathbf{x}(0) \setminus \mathscr{I}_{\leq \lfloor t/m \rfloor}^{\mathbf{x}(0)}\right) = \emptyset$$
$$\Rightarrow \mathbf{x}(t) \cap \mathscr{J} = \emptyset \tag{151}$$

and consequently

$$\Pr(\mathbf{x}(t) \cap \mathscr{J} \neq \emptyset) \leq \Pr\left(U_{tR}(\mathscr{J}) \cap \left(\mathbf{x}(0) \setminus \mathscr{I}_{\leq \lfloor t/m \rfloor}^{\mathbf{x}(0)}\right) \neq \emptyset\right) . \tag{152}$$

Therefore we find

$$\Pr(\mathbf{x}(t) \cap \mathscr{J} \neq \emptyset) \leq (2tR + |\mathscr{J}|) \alpha^{\lfloor t/m \rfloor^\beta} . \tag{153}$$

If we define $\gamma = -\log(\alpha)$ (with $\gamma > 0$ for $p_0^{\mathsf{X}} < \tilde{p}_c^{\mathsf{X}}$), it follows

$$\Pr(\mathbf{x}(t) \cap \mathscr{J} \neq \emptyset) \leq (2tR + |\mathscr{J}|) \exp\left(-\gamma \lfloor t/m \rfloor^\beta\right) \tag{154}$$

and we are done. ∎

With Lemma 2, we are ready to tackle the case of *finite* chains:

**Lemma 3.** *Consider a **finite** chain of length $L$ on $\mathscr{L} = \{1, \dots, L\}$ governed by $\overline{\mathrm{TLV}}$ with mirrored boundaries and initial configurations $\mathbf{x}(0) \subseteq \mathscr{L}$ drawn from a Bernoulli distribution with parameter $p_0^{\mathsf{X}}$.*

*Then the probability of $\mathbf{x}(t) = \overline{\mathrm{TLV}}_{\mathscr{L}}^t(\mathbf{x}(0))$ to be non-empty is upper-bounded by*

$$\Pr(\mathbf{x}(t) \neq \emptyset) \leq (4R\{t\} + L) \exp\left(-\gamma \lfloor \{t\}/m \rfloor^\beta\right) , \tag{155}$$

*with $\{t\} \equiv \min\{t, t_L^*\}$ and $t_L^* = \lfloor L/2R \rfloor$. The parameters are the same as in Prop. 1 and Lemma 2.*

*Proof.* Let $\mathbf{x}_0 = \mathbf{x}(0) \subseteq \mathbb{Z} \cap [1, L]$ be an arbitrary configuration of length $L$ and $\mathbf{y}_0 = \mathbf{y}(0) \subset \mathbb{Z} \cap [L+1, \infty)$ another arbitrary, half-infinite configuration. Denote by $\mathbf{x}_0 \mathbf{y}_0 = \mathbf{x}_0 \cup \mathbf{y}_0$ the extension of the finite chain $\mathbf{x}_0$ by $\mathbf{y}_0$ to a half-infinite chain. If we write $\overline{\mathbf{x}}(t) = \overline{\mathrm{TLV}}_{[1,L]}^t(\mathbf{x}_0)$ and $\overrightarrow{\mathbf{x}}(t)\mathbf{y}(t) = \overrightarrow{\mathrm{TLV}}_{\mathbb{N}}^t(\mathbf{x}_0\mathbf{y}_0)$, it is clear that ($L$ even)

$$\overline{x}_i(t) = \overrightarrow{x}_i(t) \quad \text{for} \quad 1 \le i \le \frac{L}{2}, \, 0 \le t \le t_L^*, \tag{156}$$

with $t_L^* = \lfloor L/2R \rfloor$ due to the finite speed $R$ of information transfer. The put it in a nutshell: the leftmost half of a finite chain evolves exactly like the corresponding section of a half-infinite chain adjacent to the mirrored boundary for $t \le t_L^*$. This is obvious because these sites cannot be influenced by the existence/non-existence of the rightmost boundary as long as it does not enter their past light cone (which happens at $t \sim L/2R$ or later). If we combine this with the fact that, for Bernoulli distributed initial states, $\mathbf{x}_0$ and $\mathbf{y}_0$ are uncorrelated, it follows immediately that all results on $\overrightarrow{\mathrm{TLV}}$ hold also for $\overline{\mathrm{TLV}}$ as long as only times $t \le t_L^*$ and sites in $[1, L/2]$ are concerned. In particular, Lemma 2 tells us that

$$\Pr\left(\overline{\mathbf{x}}(t) \cap [1, L/2] \ne \emptyset\right) \le (2tR + L/2) \exp\left(-\gamma \lfloor t/m \rfloor^\beta\right) \tag{157}$$

for $t \le t_L^*$. On account of the reflection symmetry of TLV, all statements hold also for the rightmost half $[L/2 + 1, L]$ with a mirrored boundary to the right (then with a reflected, half-infinite set of rules $\overleftarrow{\mathrm{TLV}}$). Therefore subadditivity yields

$$\Pr\left(\overline{\mathbf{x}}(t) \ne \emptyset\right) \le (4tR + L) \exp\left(-\gamma \lfloor t/m \rfloor^\beta\right) \tag{158}$$

for $t \le t_L^*$.

Here comes the crucial step: Since the chain is *finite* and $\overline{\mathbf{x}} = \emptyset$ is a fixed point of $\overline{\mathrm{TLV}}$, it is $\overline{\mathbf{x}}(t_L^*) = \emptyset \Rightarrow \overline{\mathbf{x}}(t) = \emptyset$ for all $t > t_L^*$. It follows that

$$\Pr\left(\overline{\mathbf{x}}(t) \ne \emptyset\right) \le \Pr\left(\overline{\mathbf{x}}(t_L^*) \ne \emptyset\right) \tag{159}$$

for $t \ge t_L^*$. This leads to

$$\Pr\left(\overline{\mathbf{x}}(t) \ne \emptyset\right) \le (4R\{t\} + L) \exp\left(-\gamma \lfloor \{t\}/m \rfloor^\beta\right), \tag{160}$$

with $\{t\} \equiv \min\{t, t_L^*\}$ for all $t \ge 0$. ∎

Note that the lower-bounded decay of the probability is to be expected for a *finite* system: Due to the finite state space, there is an upper bound for $t$ (depending on $L$) such that the system either (1) relaxed to the clean state, (2) to a non-clean fixed point, or (3) entered a non-trivial cycle. In the first case, it is clean forever, whereas in the latter two cases, it can never become clean. Therefore the probability to be not clean cannot decrease arbitrarily and must be bounded from below for fixed $L$ and $t \to \infty$.

However, if we are interested in the thermodynamic limit, $L \to \infty$, we can ask how long one has to wait for $\overline{\mathrm{TLV}}$ to clean the system almost surely. This leads us to our main result:

**Corollary 1.** *Consider a **finite** chain of length $L$ on $\mathscr{L} = \{1, \ldots, L\}$ governed by $\overline{\text{TLV}}$ with mirrored boundaries and initial configurations $\mathbf{x}(0) \subseteq \mathscr{L}$ drawn from a Bernoulli distribution with parameter $p_0^{\mathsf{X}}$.*

*For $\kappa \in \mathbb{R}$ with $0 < \kappa < 1$, the probability of $\mathbf{x}(t) = \overline{\text{TLV}}_{\mathscr{L}}^{t}(\mathbf{x}(0))$ to be non-empty after*

$$t_{\max}(L) \equiv \lfloor L^{\kappa} \rfloor \tag{161}$$

*time steps is upper-bounded by*

$$\Pr(\mathbf{x}(t_{\max}) \neq \emptyset) \leq (4R + 1) L \exp\left(-\gamma \lfloor L^{\kappa}/m \rfloor^{\beta}\right) \tag{162}$$

*for $L \geq L_R$ with $0 < L_R < \infty$ a $R$-dependent constant. For $p_0^{\mathsf{X}} < \tilde{p}_c^{\mathsf{X}}$ it follows that*

$$\Pr(\mathbf{x}(t_{\max}(L)) \neq \emptyset) \to 0 \quad \text{for} \quad L \to \infty \tag{163}$$

*exponentially fast. The parameters are the same as in Prop. 1 and Lemma 2.*

*Proof.* Use the result of Lemma 3 with $t = t_{\max}(L) < t_L^*$ for $L > L_R$ where $L_R$ is a finite $R$-dependent constant. Then $\{t_{\max}(L)\} = \min\{t_{\max}(L), t_L^*\} = t_{\max}(L)$ and we have

$$\Pr(\mathbf{x}(t_{\max}) \neq \emptyset) \leq (4R\, t_{\max} + L) \exp\left(-\gamma \lfloor t_{\max}/m \rfloor^{\beta}\right). \tag{164}$$

If we use that $\lfloor \lfloor L^{\kappa} \rfloor/m \rfloor = \lfloor L^{\kappa}/m \rfloor$ (for $m \in \mathbb{N}$) this yields the final result

$$\Pr(\mathbf{x}(t_{\max}) \neq \emptyset)$$
$$\leq (4R\lfloor L^{\kappa} \rfloor + L) \exp\left(-\gamma \lfloor L^{\kappa}/m \rfloor^{\beta}\right) \tag{165a}$$
$$\leq (4R + 1) L \exp\left(-\gamma \lfloor L^{\kappa}/m \rfloor^{\beta}\right), \tag{165b}$$

which vanishes for $L \to \infty$ for $\kappa, \beta > 0$ and $\gamma > 0$, i.e., if $p_0^{\mathsf{X}} < \tilde{p}_c^{\mathsf{X}}$. ∎

Note that one could get rid of the remaining floor function via

$$\lfloor L^{\kappa}/m \rfloor \geq L^{\kappa}/m - 1 \geq (1 - \varepsilon) L^{\kappa}/m, \tag{166}$$

where the last lower bound requires

$$\varepsilon L^{\kappa}/m \geq 1, \tag{167}$$

which holds for all $\varepsilon > 0$ if $L > L_{\varepsilon}$ is large enough. Then, for $L > \max\{L_R, L_{\varepsilon}\}$, one finds

$$\Pr(\mathbf{x}(t_{\max}) \neq \emptyset) \leq (4R + 1) L \exp\left(-\gamma [(1 - \varepsilon)/m]^{\beta} L^{\kappa\beta}\right), \tag{168}$$

which clearly vanishes exponentially fast for $L \to \infty$ if $0 < \varepsilon < 1$ and $\gamma, \kappa, \beta > 0$.

As a concluding remark, we note that the growth of $t_{\max}$ with $L$ can be much slower, namely (poly-)logarithmic,

$$t_{\max}(L) = \lfloor (\ln L)^{\kappa} \rfloor \tag{169}$$

for large enough $\kappa > 0$. Indeed, the probability still vanishes for $L \to \infty$ (but now sub-exponentially),

$$\Pr(\mathbf{x}(t_{\max}) \neq \emptyset) \lesssim L \exp\left[-\tilde{\gamma}(\ln L)^{\kappa\beta}\right] \tag{170a}$$
$$= L^{1 - \tilde{\gamma}/\ln(L)^{1-\kappa\beta}} \longrightarrow 0 \quad \text{for} \quad L \to \infty \tag{170b}$$

if $\kappa\beta > 1 \Leftrightarrow \kappa > \beta^{-1} = \ln(4k + 3)/\ln(2)$ and for some $\tilde{\gamma} > 0$.

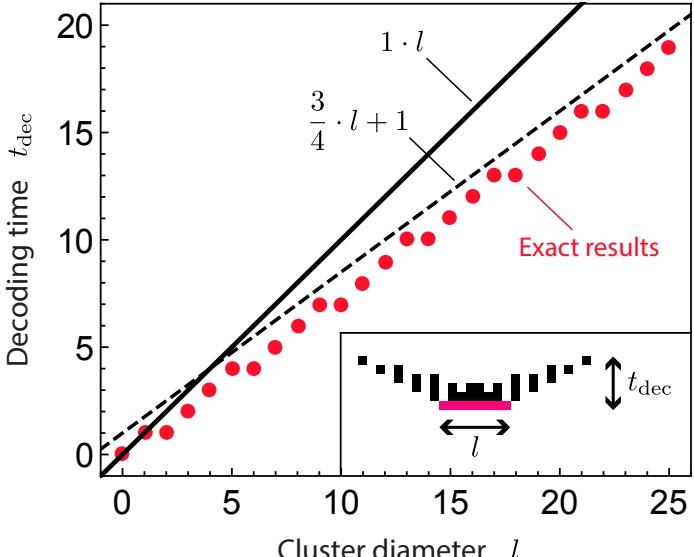

Figure 14: *Eroder parameter.* Time $t_{\text{dec}}$ needed by TLV to erase a homogeneous cluster of diameter $l$ completely. The red bullets mark exact results from simulations, featuring a 4-periodic structure that derives from the rules of radius $R = 4$. The most stringent upper bound is given by $t_{\text{dec}} \leq \frac{3}{4} \cdot l + 1$ (dashed line) but we use $t_{\text{dec}} \leq 1 \cdot l$ (solid line) for the sake of simplicity (i.e., $m = 1$). These bounds are also valid for non-homogeneous clusters due to the monotonicity of TLV.

## F    Parameters

TLV is a linear eroder, i.e., clusters on a background of zeros/ones with diameter $l$ are erased after at most $ml$ time steps, where $m \in \mathbb{R}^+$ is a rule-specific constant:

$$t_{\text{dec}} \leq m\|I\| \tag{171}$$

for arbitrary (independent) clusters $I$. To determine $m$, it is easiest to simulate the evolution of homogeneous clusters of ones on a background of zeros for increasing diameter $l$. The monotonicity of TLV (holes in the initial cluster entail holes in the spacetime diagram) makes the inferred upper bound valid for arbitrary clusters.

In Fig. 14 we show results for $0 \leq l \leq 25$. The exact results feature a 4-periodic structure which is related to the rules of radius $R = 4$. The most stringent upper bound reads

$$t_{\text{dec}} \leq \frac{3}{4} \cdot l + 1, \tag{172}$$

which does not exactly fit our needs of linearity (due to the affine offset, it can be recast into a purely linear upper bound for large enough $l$ with slightly increased prefactor $3/4 + \varepsilon$). For the sake of simplicity, we choose

$$t_{\text{dec}} \leq 1 \cdot l, \tag{173}$$

such that the eroder parameter becomes $m = 1$. Fig. 14 confirms that this upper bound on $t_{\text{dec}}$ is valid for all $l \geq 0$.

With Lemma 2, Prop. 1 and the radius $R = 4$ we find the (non-optimal) sparseness parameter $k = 2Rm = 8$, and the lower bound on $p_c^{\mathsf{X}}$ evaluates to

$$\tilde{p}_c^{\mathsf{X}} = \frac{1}{[(2k)(4k+3)]^2} \approx 3.2 \times 10^{-6}. \tag{174}$$

Note that this result does not convey any information on the true critical value $p_c^{\chi}$ (below which successful decoding is possible) but that it is larger than the above value and therefore non-zero. In fact, simulations suggest that $p_c^{\chi} = \frac{1}{2}$ and $\tilde{p}_c^{\chi}$ is only a weak lower bound (which explains why there is no much sense in optimizing $m$ slightly from $m = 1$ to $m = 3/4 + \varepsilon$).

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
