# Peer review of "Strictly local one-dimensional topological quantum error correction with symmetry-constrained cellular automata"

_SciPost Physics, doi:SciPost Phys. 4, 007 (2018)_

## Round 1 · Referee Report · Anonymous (Referee 1) · 2017-12-27

Strengths

1- Self-contained 2- Concise 3- Accessible 4- Provides intuition 5- Considers practicalities of implementation

Weaknesses

1- Proof of Lemma 1 is a bit hard to follow. In particular, fact (39) might not be obvious to most non-experts in error correction. 2- Section 5.2 slightly dense and hard to follow

Report

In this work, the authors address the important issue of error correcting the Majorana wire, which in this context serves the purpose of a quantum memory of a logical qubit.
The Majorana chain at its sweet spot (topological fixed-point) is viewed in terms of stabilisers, and consequently as a 1D limit of the Toric Code.
Motivating the need for error correcting against noise threatening the logical information's safety, MWPM reduces to majority voting under the 1D limit.

1)Here I kindly ask the authors to possibly comment on the result of arXiv:1701.00797. In this paper, it is
shown that the information stored at the edges of the Majorana wire survives at finite temperature for long times,
but they do not mention the need for error correction. How do these results and the results of the current work interplay?

The authors continue to explain the physical constraints of global majority voting in a clear way
and motivate the use of local decoders. Here another question arises and I would like to see it commented upon in one or two sentences:
2)Do the same constraints apply to MWPM when decoding the Toric Code or other higher-than-1D codes?

A CA decoder, called TLV is introduced as a candidate for a local decoder and the generic properties of CA are nicely presented.
In particular, the authors prove the equivalence between state-state and syndrome-delta representations which allows them to later design a deep decoder made of stacked TLVs in order to fight against continuous noise.
3)When the concept of a deep decoder is introduced, I would request the authors to very briefly comment on a possible connection with the holographic decoder discussed in arXiv:1503.06237.

In the following sections, the authors argue that a local decoder that approximates global majority voting is performing 1D density classification. With analytical (especially section 4.2) and numerical results, they show that the modified TLV which takes care of finite chain lengths is able to decode random error patterns corresponding to a low enough error rates with decoding time scaling sublinearly,
as well as Corollary 1, among others, which are important results.

Finally, in the presence of continuous noise, the deconfinement of error defects is shown. This results in the failure of TLV to decode efficiently as the proposed CA is argued to be ergodic.
However, the introduction of 2D-evolved TLV recovers the desired scaling of decoding time with chain length.
Importantly, the depth of the decoder scales favourably, which allows for a shallow decoding circuit.
The attention to what is implementable experimentally is also something to be commended.

In conclusion, I find this work important and suitable for publication.

Requested changes

I leave it up to the authors to decide whether they believe it is relevant to comment on points 1) 2) 3) in the Report.

  • validity: top
  • significance: high
  • originality: high
  • clarity: high
  • formatting: perfect
  • grammar: excellent

Author:  Nicolai Lang  on 2018-01-12  [id 199]

(in reply to Report 1 on 2017-12-27)
Category:
remark
answer to question

We would like to thank the referee for carefully reading our manuscript and his/her constructive questions and comments. We took care of all raised issues in a revised version of the manuscript. It the following, we comment in detail on these issues:

1) Here I kindly ask the authors to possibly comment on the result of arXiv:1701.00797. In this paper, it is shown that the information stored at the edges of the Majorana wire survives at finite temperature for long times, but they do not mention the need for error correction. How do these results and the results of the current work interplay?

The authors of arXiv:1701.00797 study the transverse field Ising model, which is indeed mathematically equivalent to the fermionic Majorana chain. The focus is on the correlators $A_s(t)=\langle s|\sigma^z_1(t)\sigma^z_1(0)|s\rangle$ for eigenstates $|s\rangle$ of the Hamiltonian which probe how much the edge spin remembers of its initial polarization. Indeed they find that surprisingly the system, even for a thermal initial state, remembers its initial state.

However, there is a very crucial difference in their setting compared to setups relevant for quantum error correction: the authors study a closed system and analyze the unitary time evolution for a mixed initial state. Such scenarios are closely related to questions on thermal equilibration of a subsystem under unitary dynamics and its failure under many-body localization.

In contrast, the relevant physical setting for quantum error correction is completely different: there, one is interested in a system coupled to an environment where noise and fluctuations in the environment continuously perturb the quantum code, i.e., one is interested in the stability of the memory under arbitrary time-dependent perturbations. In a open-system setting with a thermal bath (which is where our motivation stems from), the correlators $A_s(t)$ in the above reference would decay exponentially. Just think of $\sigma^x_1$ (spin flips) operating with a size independent rate on the edge spin due to thermal fluctuations. Such operations make $A_s(t)$ decay quickly in time.

To avoid confusion between these physically completely different questions, we avoided to discuss the above manuscript as well as questions on many-body localization.

The authors continue to explain the physical constraints of global majority voting in a clear way and motivate the use of local decoders. Here another question arises and I would like to see it commented upon in one or two sentences:

2) Do the same constraints apply to MWPM when decoding the Toric Code or other higher-than-1D codes?

In principle yes, because MWPM, just as majority voting, is a function of the global syndrome. Therefore performing exact MWPM on an extensive system requires an extensive time (at least scaling with the linear dimension of the system, the time complexity of evaluating the functions comes on top). Whether this is of practical importance is another question and certainly depends on the size of the code and its particular realization.

Regarding the scaling of the logical error rate for a linearly decreasing correction frequency, we showed only that it is not tolerable for the Majorana chain. We cannot say anything about the 2D toric code (or any other 2D topological code) because an analytical expression for the logical failure rate in dependence of system size and microscopic error probability is required to evaluate the limit $L\rightarrow\infty$ of Eq. (25) (which we are not aware of for the toric code).

But in principle it is possible that 2D codes are more resilient because their physical substrate scales as $L^2$ whereas the constraint due to finite communication speed still scales as $L$. However (at least for the toric code) we would be surprised if this is the case because the second dimension is already "used" to protect from bit-flip errors without requiring symmetry protection (as compared to the 1D Majorana chain).

3) When the concept of a deep decoder is introduced, I would request the authors to very briefly comment on a possible connection with the holographic decoder discussed in arXiv:1503.06237.

I was not aware of the mentioned paper (it is an interesting piece of work; thank you for pointing me to it). After reading most of it, I do not think that suggesting a link between their setting and ours helps the reader to understand our setup. Let me detail on this:

The authors of arXiv:1503.06237 study a specific class of tensor networks that are most conveniently embedded in hyperbolic space. Dividing uncontracted indices of the network into two classes (on the boundary and in the bulk) allows an interpretation of the network as an isometric mapping that encodes few bulk qubits into many boundary qubits (this is where the hyperbolic geometry is important). The graphs describing these networks are typically infinite-dimensional in our Euclidean space (like the Bethe lattice). Because this is tensor network language, physical reality is located on the boundary only (the "physical" qubits). Thus the "depth" of the network (the radius from the central logical qubit to the peripheral physical ones) is purely abstract and has no geometrical meaning in real space (where the physical qubits live): It is the property of a mathematical map (the encoding map) which describes a linear subspace parametrized by the bulk qubits (the codespace) in a much larger physical Hilbert space (spanned by the boundary qubits). This is in contrast to the notion of depth in our case where it refers to the physical dimension of classical circuitry perpendicular to a one-dimensional quantum substrate (which hosts the Majorana chain quantum code). In addition, the tensor network of arXiv:1503.06237 describes the encoding map and thereby the quantum code itself. In contrast, our classical circuit realizes a decoding algorithm. The authors of arXiv:1503.06237 do not discuss decoding in detail because their focus is on AdS/CFT, but when they discuss stabilizer realizations of their holographic codes it becomes clear that measuring the syndrome (i.e., the stabilizers) is a non-local task on the boundary which would be completely incompatible with our approach (local stabilizers measurements are clearly essential for local decoding). So neither interpretation (virtual tensors vs. actual classical gates) nor purpose (encoding vs. decoding/stabilization) is comparable between arXiv:1503.06237 and our paper.

Weaknesses:

1- Proof of Lemma 1 is a bit hard to follow. In particular, fact (39) might not be obvious to most non-experts in error correction.

  • We added an new Eq. (39) to make the validity of Eq. (40) (formerly 39) clear.

  • We agree that the proof is rather technical (and consequently hard to follow) although its mathematical content is quite basic. One could defer it to the appendix but then Lemma 1 becomes more mysterious than it really is. Therefore we added a preliminary sentence to allow readers to skip the proof if they are willing to believe Lemma 1 (or just "see" it).

2- Section 5.2 slightly dense and hard to follow

  • We considered expanding this part. However, we noticed that this section then becomes too long while still being rather technical. Since the proposed implementation is only meant as a proof-of-principle implementation (it certainly deserves optimization), dwelling too long on its details eventually distracts from the main message of the manuscript (viz, that going into the second dimension irons out the insufficiencies of strictly 1D decoders, irrespective of the technical details). Therefore we would like to ask the referee whether we can keep this section in its current (terse) version. We added a short note at the beginning of section 5.2. to make the "proof-of-principle"-nature of what follows clear to the reader.

---

## Round 2 · Author Response

See our reply to the first referee report for detailed comments.

---

## Round 2 · List of Changes

- Added Eq. (39) to derive the validity of Eq. (40)
- Added a preliminary statement on p. 17 before the proof to allow readers to skip the latter without loosing the guiding thread of the paper.
- Added a preliminary note on p. 33 at the beginning of section 5.2 to highlight the proof-of-concept character of our proposed setup.

You are currently on this page

Resubmission 1711.08196v2 on 15 January 2018

---

## Editorial Decision

published